# Controlling the Complexity and Lipschitz Constant improves Polynomial Nets

**Zhenyu Zhu,    Fabian Latorre,    Grigorios G Chrysos,    Volkan Cevher**

EPFL, Switzerland
{[first name].[surname]}@epfl.ch

## Abstract

While the class of Polynomial Nets demonstrates comparable performance to neural networks (NN), it currently has neither theoretical generalization characterization nor robustness guarantees. To this end, we derive new complexity bounds for the set of Coupled CP-Decomposition (CCP) and Nested Coupled CP-decomposition (NCP) models of Polynomial Nets in terms of the $\ell_\infty$-operator-norm and the $\ell_2$-operator norm. In addition, we derive bounds on the Lipschitz constant for both models to establish a theoretical certificate for their robustness. The theoretical results enable us to propose a principled regularization scheme that we also evaluate experimentally in six datasets and show that it improves the accuracy as well as the robustness of the models to adversarial perturbations. We showcase how this regularization can be combined with adversarial training, resulting in further improvements.

## 1 Introduction

Recently, high-degree Polynomial Nets (PNs) have been demonstrating state-of-the-art performance in a range of challenging tasks like image generation (Karras et al., 2019; Chrysos and Panagakis, 2020), image classification (Wang et al., 2018), reinforcement learning (Jayakumar et al., 2020), non-euclidean representation learning (Chrysos et al., 2020) and sequence models (Su et al., 2020). In particular, in public benchmarks like the *Face verification on MegaFace* task[1] (Kemelmacher-Shlizerman et al., 2016), Polynomial Nets are currently the top performing model.

A major advantage of Polynomial Nets over traditional Neural Networks[2] is that they are compatible with efficient *Leveled Fully Homomorphic Encryption (LFHE)* protocols (Brakerski et al., 2014). Such protocols allow efficient computation on encrypted data, but they only support addition or multiplication operations i.e., polynomials. This has prompted an effort to adapt neural networks by replacing typical activation functions with polynomial approximations (Gilad-Bachrach et al., 2016; Hesamifard et al., 2018). Polynomial Nets do not need any adaptation to work with LFHE.

Without doubt, these arguments motivate further investigation about the inner-workings of Polynomial Nets. Surprisingly, little is known about the theoretical properties of such high-degree polynomial expansions, despite their success. Previous work on PNs (Chrysos et al., 2020; Chrysos and Panagakis, 2020) have focused on developing the foundational structure of the models as well as their training, but do not provide an analysis of their generalization ability or robustness to adversarial perturbations.

In contrast, such type of results are readily available for traditional feed-forward Deep Neural Networks, in the form of high-probability generalization error bounds (Neyshabur et al., 2015; Bartlett et al., 2017; Neyshabur et al., 2017; Golowich et al., 2018) or upper bounds on their Lipschitz constant (Scaman and Virmaux, 2018; Fazlyab et al., 2019; Latorre et al., 2020). Despite their similarity in the compositional structure, theoretical results for Deep Neural Networks[2] do not apply to Polynomial Nets, as they are essentialy two non-overlapping classes of functions.

Why are such results important? First, they provide key theoretical quantities like the sample complexity of a hypothesis class: how many samples are required to succeed at learning in the PAC-framework. Second, they provide certified performance guarantees to adversarial perturbations

---

[1] https://paperswithcode.com/sota/face-verification-on-megaface
[2] with non-polynomial activation functions.

(Szegedy et al., 2014; Goodfellow et al., 2015) via a worst-case analysis c.f. Scaman and Virmaux (2018). Most importantly, the bounds themselves provide a principled way to regularize the hypothesis class and improve their accuracy or robustness.

For example, Generalization and Lipschitz constant bounds of Deep Neural Networks that depend on the operator-norm of their weight matrices (Bartlett et al., 2017; Neyshabur et al., 2017) have laid out the path for regularization schemes like spectral regularization (Yoshida and Miyato, 2017; Miyato et al., 2018), Lipschitz-margin training (Tsuzuku et al., 2018) and Parseval Networks (Cisse et al., 2017), to name a few.

Indeed, such schemes have been observed in practice to improve the performance of Deep Neural Networks. Unfortunately, similar regularization schemes for Polynomial Nets do not exist due to the lack of analogous bounds. Hence, it is possible that PNs are not yet being used to their fullest potential. We believe that theoretical advances in their understanding might lead to more resilient and accurate models. In this work, we aim to fill the gap in the theoretical understanding of PNs. We summarize our main contributions as follows:

**Rademacher Complexity Bounds.** We derive bounds on the Rademacher Complexity of the Coupled CP-decomposition model (CCP) and Nested Coupled CP-decomposition model (NCP) of PNs, under the assumption of a unit $\ell_\infty$-norm bound on the input (Theorems 1 and 3), a natural assumption in image-based applications. Analogous bounds for the $\ell_2$-norm are also provided (Appendices E.3 and F.3). Such bounds lead to the first known generalization error bounds for this class of models.

**Lipschitz constant Bounds.** To complement our understanding of the CCP and NCP models, we derive upper bounds on their $\ell_\infty$-Lipschitz constants (Theorems 2 and 4), which are directly related to their robustness against $\ell_\infty$-bounded adversarial perturbations, and provide formal guarantees. Analogous results hold for any $\ell_p$-norm (Appendices E.4 and F.4).

**Regularization schemes.** We identify key quantities that simultaneously control both Rademacher Complexity and Lipschitz constant bounds that we previously derived, i.e., the operator norms of the weight matrices in the Polynomial Nets. Hence, we propose to regularize the CCP and NCP models by constraining such operator norms. In doing so, our theoretical results indicate that both the generalization and the robustness to adversarial perturbations should improve. We propose a Projected Stochastic Gradient Descent scheme (Algorithm 1), enjoying the same per-iteration complexity as vanilla back-propagation in the $\ell_\infty$-norm case, and a variant that augments the base algorithm with adversarial traning (Algorithm 2).

**Experiments.** We conduct experimentation in *five* widely-used datasets on image recognition and on dataset in audio recognition. The experimentation illustrates how the aforementioned regularization schemes impact the accuracy (and the robust accuracy) of both CCP and NCP models, outperforming alternative schemes such as Jacobian regularization and the $L_2$ weight decay. Indeed, for a grid of regularization parameters we observe that there exists a sweet-spot for the regularization parameter which not only increases the test-accuracy of the model, but also its resilience to adversarial perturbations. Larger values of the regularization parameter also allow a trade-off between accuracy and robustness. The observation is consistent across all datasets and all adversarial attacks demonstrating the efficacy of the proposed regularization scheme.

## 2 RADEMACHER COMPLEXITY AND LIPSCHITZ CONSTANT BOUNDS FOR POLYNOMIAL NETS

**Notation.** The symbol $\circ$ denotes the Hadamard (element-wise) product, the symbol $\bullet$ is the face-splitting product, while the symbol $\star$ denotes a convolutional operator. Matrices are denoted by uppercase letters e.g., $V$. Due to the space constraints, a detailed notation is deferred to Appendix C.

**Assumption on the input distribution.** Unless explicitly mentioned otherwise, we assume an $\ell_\infty$-norm unit bound on the input data i.e., $\|z\|_\infty \le 1$ for any input $z$. This is the most common assumption in image-domain applications in contemporary deep learning, i.e., each pixel takes values in $[-1, 1]$ interval. Nevertheless, analogous results for $\ell_2$-norm unit bound assumptions are presented in Appendices E.3, E.4, F.3 and F.4.

We now introduce the basic concepts that will be developed throughout the paper i.e., the Rademacher Complexity of a class of functions (Bartlett and Mendelson, 2002) and the Lipschitz constant.

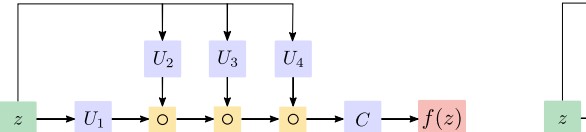 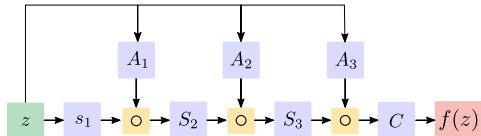

Figure 1: Schematic of CCP model (left) and NCP model (right), where $\circ$ denotes the Hadamard product. Blue boxes correspond to learnable parameters. Green and red boxes denote input and output, respectively. Yellow boxes denote operations.

**Definition 1** (Empirical Rademacher Complexity). *Let $Z = \{z_1, \ldots, z_n\} \subseteq \mathbb{R}^d$ and let $\{\sigma_j : j = 1, \ldots, n\}$ be independent Rademacher random variables i.e., taking values uniformly in $\{-1, +1\}$. Let $\mathcal{F}$ be a class of real-valued functions over $\mathbb{R}^d$. The Empirical Rademacher complexity of $\mathcal{F}$ with respect to $Z$ is defined as follows:*

$$\mathcal{R}_Z(\mathcal{F}) \coloneqq \mathbb{E}_\sigma\left[\sup_{f \in \mathcal{F}} \frac{1}{n}\sum_{j=1}^n \sigma_j f(z_j)\right].$$

**Definition 2** (Lipschitz constant). *Given two normed spaces $(\mathcal{X}, \|\cdot\|_\mathcal{X})$ and $(\mathcal{Y}, \|\cdot\|_\mathcal{Y})$, a function $f : \mathcal{X} \to \mathcal{Y}$ is called Lipschitz continuous with Lipschitz constant $K \geq 0$ if for all $x_1, x_2$ in $\mathcal{X}$:*

$$\|f(x_1) - f(x_2)\|_\mathcal{Y} \leq K\|x_1 - x_2\|_\mathcal{X}.$$

## 2.1 Coupled CP-Decomposition of Polynomial Nets (CCP model)

The Coupled CP-Decomposition (CCP) model of PNs (Chrysos et al., 2020) leverages a coupled CP Tensor decomposition (Kolda and Bader, 2009) to vastly reduce the parameters required to describe a high-degree polynomial, and allows its computation in a compositional fashion, much similar to a feed-forward pass through a traditional neural network. The CCP model was used in Chrysos and Panagakis (2020) to construct a generative model. CCP can be succintly defined as follows:

$$f(z) = C \circ_{i=1}^k U_i z, \tag{CCP}$$

where $z \in \mathbb{R}^d$ is the input data, $f(z) \in \mathbb{R}^o$ is the output of the model and $U_n \in \mathbb{R}^{m \times d}, C \in \mathbb{R}^{o \times m}$ are the learnable parameters, where $m \in \mathbb{N}$ is the hidden rank. In Fig. 1 we provide a schematic of the architecture, while in Appendix D.1 we include further details on the original CCP formulation (and how to obtain our equivalent re-parametrization) for the interested reader.

In Theorem 1 we derive an upper bound on the complexity of CCP models with bounded $\ell_\infty$-operator-norms of the face-splitting product of the weight matrices. Its proof can be found in Appendix E.1. For a given CCP model, we derive an upper bound on its $\ell_\infty$-Lipschitz constant in Theorem 2 and its proof is given in Appendix E.4.1.

**Theorem 1.** *Let $Z = \{z_1, \ldots, z_n\} \subseteq \mathbb{R}^d$ and suppose that $\|z_j\|_\infty \leq 1$ for all $j = 1, \ldots, n$. Let*

$$\mathcal{F}_{CCP}^k \coloneqq \left\{f(z) = \left\langle c, \circ_{i=1}^k U_i z\right\rangle : \|c\|_1 \leq \mu, \left\|\bullet_{i=1}^k U_i\right\|_\infty \leq \lambda\right\}.$$

*The Empirical Rademacher Complexity of $CCP_k$ ($k$-degree CCP polynomials) with respect to $Z$ is bounded as:*

$$\mathcal{R}_Z(\mathcal{F}_{CCP}^k) \leq 2\mu\lambda\sqrt{\frac{2k\log(d)}{n}}.$$

**Proof sketch of Theorem 1** We now describe the core steps of the proof. For the interested reader, the complete and detailed proof steps are presented in Appendix E.1. First, Hölder's inequality is used to bound the Rademacher complexity as:

$$\mathcal{R}_Z(\mathcal{F}_{\text{CCP}}^k) = \mathbb{E}\sup_{f \in \mathcal{F}_{\text{CCP}}^k} \frac{1}{n}\left\langle c, \sum_{j=1}^n [\sigma_j \circ_{i=1}^k (U_i z_j)]\right\rangle \leq \mathbb{E}\sup_{f \in \mathcal{F}_{\text{CCP}}^k} \frac{1}{n}\|c\|_1 \left\|\sum_{j=1}^n [\sigma_j \circ_{i=1}^k (U_i z_j)]\right\|_\infty. \tag{1}$$

This shows why the factor $\|c\|_1 \leq \mu$ appears in the final bound. Then, using the mixed product property (Slyusar, 1999) and its extension to repeated Hadamard products (Lemma 7 in Appendix C.3), we can rewrite the summation in the right-hand-side of (1) as follows:

$$\sum_{j=1}^{n} \sigma_j \circ_{i=1}^{k} (U_i z_j) = \sum_{j=1}^{n} \sigma_j \bullet_{i=1}^{k} (U_i) *_{i=1}^{k} (z_j) = \bullet_{i=1}^{k} (U_i) \sum_{j=1}^{n} \sigma_j *_{i=1}^{k} (z_j).$$

This step can be seen as a linearization of the polynomial by lifting the problem to a higher dimensional space. We use this fact and the definition of the operator norm to further bound the term inside the $\ell_\infty$-norm in the right-hand-side of (1). Such term is bounded as the product of the $\ell_\infty$-operator norm of $\bullet_{i=1}^{k}(U_i)$, and the $\ell_\infty$-norm of an expression involving the Rademacher variables $\sigma_j$ and the vectors $*_{i=1}^{k}(z_j)$. Finally, an application of Massart's Lemma (Shalev-Shwartz and Ben-David (2014), Lemma 26.8) leads to the final result.

**Theorem 2.** *The Lipschitz constant (with respect to the $\ell_\infty$-norm) of the function defined in Eq. (CCP), restricted to the set $\{z \in \mathbb{R}^d : \|z\|_\infty \leq 1\}$ is bounded as:*

$$Lip_\infty(f) \leq k\|C\|_\infty \prod_{i=1}^{k} \|U_i\|_\infty.$$

## 2.2 NESTED COUPLED CP-DECOMPOSITION (NCP MODEL)

The Nested Coupled CP-Decomposition (NCP) model leverages a joint hierarchical decomposition, which provided strong results in both generative and discriminative tasks in Chrysos et al. (2020). A slight re-parametrization of the model (Appendix D.2) can be expressed with the following recursive relation:

$$x_1 = (A_1 z) \circ (s_1), \qquad x_n = (A_n z) \circ (S_n x_{n-1}), \qquad f(z) = C x_k, \qquad \text{(NCP)}$$

where $z \in \mathbb{R}^d$ is the input vector and $C \in \mathbb{R}^{o \times m}$, $A_n \in \mathbb{R}^{m \times d}$, $S_n \in \mathbb{R}^{m \times m}$ and $s_1 \in \mathbb{R}^m$ are the learnable parameters. In Fig. 1 we provide a schematic of the architecture.

In Theorem 3 we derive an upper bound on the complexity of NCP models with bounded $\ell_\infty$-operator-norm of a matrix function of its parameters. Its proof can be found in Appendix F.1. For a given NCP model, we derive an upper bound on its $\ell_\infty$-Lipschitz constant in Theorem 4 and its proof is given in Appendix F.4.1.

**Theorem 3.** *Let $Z = \{z_1, \ldots, z_n\} \subseteq \mathbb{R}^d$ and suppose that $\|z_j\|_\infty \leq 1$ for all $j = 1, \ldots, n$. Define the matrix $\Phi(A_1, S_1, \ldots, A_n, S_n) := (A_k \bullet S_k) \prod_{i=1}^{k-1} I \otimes A_i \bullet S_i$. Consider the class of functions:*

$$\mathcal{F}_{NCP}^k := \{f(z) \text{ as in (NCP)} : \|C\|_\infty \leq \mu, \|\Phi(A_1, S_1, \ldots, A_k, S_k)\|_\infty \leq \lambda\},$$

*where $C \in \mathbb{R}^{1 \times m}$ (single output), thus, we will write it as $c$, and the corresponding bound also becomes $\|c\|_1 \leq \mu$. The Empirical Rademacher Complexity of $NCP_k$ (k-degree NCP polynomials) with respect to $Z$ is bounded as:*

$$\mathcal{R}_Z(\mathcal{F}_{NCP}^k) \leq 2\mu\lambda \sqrt{\frac{2k \log(d)}{n}}.$$

**Theorem 4.** *The Lipschitz constant (with respect to the $\ell_\infty$-norm) of the function defined in Eq. (NCP), restricted to the set $\{z \in \mathbb{R}^d : \|z\|_\infty \leq 1\}$ is bounded as:*

$$Lip_\infty(f) \leq k\|C\|_\infty \prod_{i=1}^{k} (\|A_i\|_\infty \|S_i\|_\infty).$$

## 3 ALGORITHMS

By constraining the quantities in the upper bounds on the Rademacher complexity (Theorems 1 and 3), we can regularize the empirical loss minimization objective (Mohri et al., 2018, Theorem 3.3). Such method would prevent overfitting and can lead to an improved accuracy. However, one issue with the quantities involved in Theorems 1 and 3, namely

$$\left\|\bullet_{i=1}^{k} U_i\right\|_\infty, \qquad \left\|(A_k \bullet S_k) \prod_{i=1}^{k-1} I \otimes A_i \bullet S_i\right\|_\infty,$$

is that projecting onto their level sets correspond to a difficult non-convex problem. Nevertheless, we can control an upper bound that depends on the $\ell_\infty$-operator norm of each weight matrix:

**Lemma 1.** *It holds that* $\left\| \bullet_{i=1}^k \boldsymbol{U}_i \right\|_\infty \leq \prod_{i=1}^k \|\boldsymbol{U}_i\|_\infty.$

**Lemma 2.** *It holds that* $\left\| (\boldsymbol{A}_k \bullet \boldsymbol{S}_k) \prod_{i=1}^{k-1} \boldsymbol{I} \otimes \boldsymbol{A}_i \bullet \boldsymbol{S}_i \right\|_\infty \leq \prod_{i=1}^k \|\boldsymbol{A}_i\|_\infty \|\boldsymbol{S}_i\|_\infty.$

The proofs of Lemmas 1 and 2 can be found in Appendix E.2 and Appendix F.2. These results mean that by constraining the operator norms of each weight matrix, we can control the overall complexity of the CCP and NCP models.

Projecting a matrix onto an $\ell_\infty$-operator norm ball is a simple task that can be achieved by projecting each row of the matrix onto an $\ell_1$-norm ball, for example, using the well-known algorithm from Duchi et al. (2008). The final optimization objective for training a regularized CCP is the following:

$$\min_{\boldsymbol{C},\boldsymbol{U}_1,\ldots,\boldsymbol{U}_k} \frac{1}{n} \sum_{i=1}^n L(\boldsymbol{C},\boldsymbol{U}_1,\ldots,\boldsymbol{U}_k;\boldsymbol{x}_i,y_i) \qquad \text{subject to } \|\boldsymbol{C}\|_\infty \leq \mu, \|\boldsymbol{U}_i\|_\infty \leq \lambda, \qquad (2)$$

where $(\boldsymbol{x}_i,y_i)_{i=1}^n$ is the training dataset, $L$ is the loss function (e.g., cross-entropy) and $\mu, \lambda$ are the regularization parameters. We notice that the constraints on the learnable parameters $\boldsymbol{U}_i$ and $\boldsymbol{C}$ have the effect of simultaneously controlling the Rademacher Complexity and the Lipschitz constant of the CCP model. For the NCP model, an analogous objective function is used.

To solve the optimization problem in Eq. (2) we will use a Projected Stochastic Gradient Descent method Algorithm 1. We also propose a variant that combines Adversarial Training with the projection step (Algorithm 2) with the goal of increasing robustness to adversarial examples.

---

**Algorithm 1:** Projected SGD

**Input:** dataset $Z$, learning rate $\{\gamma_t > 0\}_{t=0}^{T-1}$, iterations $T$, hyper-parameters $R$, $f$ , Loss $L$.
**Output:** model with parameters $\boldsymbol{\theta}$.
  Initialize $\boldsymbol{\theta}$.
  **for** $t = 0$ to $T - 1$ **do**
    Sample $(\boldsymbol{x}, y)$ from $Z$
    $\boldsymbol{\theta} = \boldsymbol{\theta} - \gamma_t \nabla_{\boldsymbol{\theta}} L(\boldsymbol{\theta}; \boldsymbol{x}, y).$
    **if** $t \mod f = 0$ **then**
      $\boldsymbol{\theta} = \prod_{\{\boldsymbol{\theta}:\|\boldsymbol{\theta}\|_\infty \leq R\}}(\boldsymbol{\theta})$

---

**Algorithm 2:** Projected SGD + Adversarial Training

**Input:** dataset $Z$, learning rate $\{\gamma_t > 0\}_{t=0}^{T-1}$, iterations $T$ and $n$, hyper-parameters $R$, $f$ , $\epsilon$ and $\alpha$, Loss $L$.
**Output:** model with parameters $\boldsymbol{\theta}$.
  Initialize $\boldsymbol{\theta}$.
  **for** $t = 0$ to $T - 1$ **do**
    Sample $(\boldsymbol{x}, y)$ from $Z$
    **for** $i = 0$ to $n - 1$ **do**
      $\boldsymbol{x}^{\text{adv}} = \prod_{\{\boldsymbol{x}':\|\boldsymbol{x}'-\boldsymbol{x}\|_\infty \leq \epsilon\}} \{\boldsymbol{x} + \alpha \nabla_{\boldsymbol{x}} L(\boldsymbol{\theta}; \boldsymbol{x}, y)\}$
    $\boldsymbol{\theta} = \boldsymbol{\theta} - \gamma_t \nabla_{\boldsymbol{\theta}} L(\boldsymbol{\theta}; \boldsymbol{x}^{\text{adv}}, y)$
    **if** $t \mod f = 0$ **then**
      $\boldsymbol{\theta} = \prod_{\{\boldsymbol{\theta}:\|\boldsymbol{\theta}\|_\infty \leq R\}}(\boldsymbol{\theta})$

---

In Algorithms 1 and 2 the parameter $f$ is set in practice to a positive value, so that the projection (denoted by $\Pi$) is made only every few iterations. The variable $\boldsymbol{\theta}$ represents the weight matrices of the model, and the projection in the last line should be understood as applied independently for every weight matrix. The regularization parameter $R$ corresponds to the variables $\mu, \lambda$ in Eq. (2).

**Convolutional layers** Frequently, convolutions are employed in the literature, especially in the image-domain. It is important to understand how our previous results extend to this case, and how the proposed algorithms work in that case. Below, we show that the $\ell_\infty$-operator norm of the convolutional layer (as a linear operator) is related to the $\ell_\infty$-operator norm of the kernel after a reshaping operation. For simplicity, we consider only convolutions with zero padding.

We study the cases of 1D, 2D and 3D convolutions. For clarity, we mention below the result for the 3D convolution, since this is relevant to our experimental validation, and we defer the other two cases to Appendix G.

**Theorem 5.** *Let $\boldsymbol{A} \in \mathbb{R}^{n \times m \times r}$ be an input image and let $\boldsymbol{K} \in \mathbb{R}^{h \times h \times r \times o}$ be a convolutional kernel with $o$ output channels. For simplicity assume that $k \leq \min(n, m)$ is odd. Denote by $\boldsymbol{B} = \boldsymbol{K} \star \boldsymbol{A}$ the output of the convolutional layer. Let $\boldsymbol{U} \in \mathbb{R}^{nmo \times nmr}$ be the matrix such that $vec(\boldsymbol{B}) = \boldsymbol{U} vec(\boldsymbol{A})$ i.e., $\boldsymbol{U}$ is the matrix representation of the convolution. Let $M(\boldsymbol{K}) \in \mathbb{R}^{o \times hhr}$ be the matricization of*

$K$, *where each row contains the parameters of a single output channel of the convolution. It holds that:* $\|U\|_{\infty} = \|M(K)\|_{\infty}$.

Thus, we can control the $\ell_{\infty}$-operator-norm of a convolutional layer during training by controlling that of the reshaping of the kernel, which is done with the same code as for fully connected layers. It can be seen that when the padding is non-zero, the result still holds.

## 4 NUMERICAL EVIDENCE

The generalization properties and the robustness of PNs are numerically verified in this section. We evaluate the robustness to three widely-used adversarial attacks in sec. 4.2. We assess whether the compared regularization schemes can also help in the case of adversarial training in sec. 4.3. Experiments with additional datasets, models (NCP models), adversarial attacks (APGDT, PGDT) and layer-wise bound (instead of a single bound for all matrices) are conducted in Appendix H due to the restricted space. The results exhibit a **consistent** behavior across different adversarial attacks, different datasets and different models. Whenever the results differ, we *explicitly* mention the differences in the main body below.

### 4.1 EXPERIMENTAL SETUP

The accuracy is reported as as the evaluation metric for every experiment, where a higher accuracy translates to better performance.

**Datasets and Benchmark Models:** We conduct experiments on the popular datasets of Fashion-MNIST (Xiao et al., 2017), E-MNIST (Cohen et al., 2017) and CIFAR-10 (Krizhevsky et al., 2014). The first two datasets include grayscale images of resolution $28 \times 28$, while CIFAR-10 includes $60,000$ RGB images of resolution $32 \times 32$. Each image is annotated with one out of the ten categories. We use two popular regularization methods from the literature for comparison, i.e., Jacobian regularization (Hoffman et al., 2019) and $L_2$ regularization (weight decay).

**Models:** We report results using the following three models: 1) a $4^{\text{th}}$-degree CCP model named "PN-4", 2) a $10^{\text{th}}$-degree CCP model referenced as "PN-10" and 3) a $4^{\text{th}}$-degree Convolutional CCP model called "PN-Conv". In the PN-Conv, we have replaced all the $U_i$ matrices with convolutional kernels. None of the variants contains any activation functions.

**Hyper-parameters:** Unless mentioned otherwise, all models are trained for 100 epochs with a batch size of 64. The initial value of the learning rate is $0.001$. After the first 25 epochs, the learning rate is multiplied by a factor of $0.2$ every 50 epochs. The SGD is used to optimize all the models, while the cross-entropy loss is used. In the experiments that include projection or adversarial training, the first 50 epochs are pre-training, i.e., training only with the cross-entropy loss. The projection is performed every ten iterations.

**Adversarial Attack Settings:** We utilize two widely used attacks: a) Fast Gradient Sign Method (FGSM) and b) Projected Gradient Descent (PGD). In FGSM the hyper-parameter $\epsilon$ represents the step size of the adversarial attack. In PGD there is a triple of parameters ($\epsilon_{\text{total}}$, $n_{\text{iters}}$, $\epsilon_{\text{iter}}$), which represent the maximum step size of the total adversarial attack, the number of steps to perform for a single attack, and the step size of each adversarial attack step respectively. We consider the following hyper-parameters for the attacks: a) FGSM with $\epsilon = 0.1$, b) PGD with parameters $(0.1, 20, 0.01)$, c) PGD with parameters $(0.3, 20, 0.03)$.

### 4.2 EVALUATION OF THE ROBUSTNESS OF PNs

In the next experiment, we assess the robustness of PNs under adversarial noise. That is, the method is trained on the train set of the respective dataset and the evaluation is performed on the test set perturbed by additive adversarial noise. That is, each image is individually perturbed based on the respective adversarial attack. The proposed method implements Algorithm 1.

The quantitative results in both Fashion-MNIST and E-MNIST using PN-4, PN-10 and PN-Conv under the three attacks are reported in Table 1. The column 'No-proj' exhibits the plain SGD training (i.e., without regularization), while the remaining columns include the proposed regularization, Jacobian and the $L_2$ regularization respectively. The results without regularization exhibit a substantial decrease in accuracy for stronger adversarial attacks. The proposed regularization outperforms all methods consistently across different adversarial attacks. Interestingly, the stronger the adversarial

| | Method | No proj. | Our method | Jacobian | $L_2$ |
|---|---|---|---|---|---|
| | | | *Fashion-MNIST* | | |
| PN-4 | Clean | $87.28 \pm 0.18\%$ | $\mathbf{87.32 \pm 0.14}\%$ | $86.24 \pm 0.14\%$ | $87.31 \pm 0.13\%$ |
| | FGSM-0.1 | $12.92 \pm 2.74\%$ | $\mathbf{46.43 \pm 0.95}\%$ | $17.90 \pm 6.51\%$ | $13.80 \pm 3.65\%$ |
| | PGD-(0.1, 20, 0.01) | $5.64 \pm 1.76\%$ | $\mathbf{49.58 \pm 0.59}\%$ | $12.23 \pm 5.63\%$ | $5.01 \pm 2.44\%$ |
| | PGD-(0.3, 20, 0.03) | $0.18 \pm 0.16\%$ | $\mathbf{28.96 \pm 2.31}\%$ | $1.27 \pm 1.29\%$ | $0.28 \pm 0.18\%$ |
| PN-10 | Clean | $88.48 \pm 0.17\%$ | $\mathbf{88.72 \pm 0.12}\%$ | $88.12 \pm 0.11\%$ | $88.46 \pm 0.19\%$ |
| | FGSM-0.1 | $15.96 \pm 1.00\%$ | $\mathbf{44.71 \pm 1.24}\%$ | $19.52 \pm 1.14\%$ | $16.51 \pm 2.33\%$ |
| | PGD-(0.1, 20, 0.01) | $1.94 \pm 0.82\%$ | $\mathbf{47.94 \pm 2.29}\%$ | $5.44 \pm 0.81\%$ | $2.16 \pm 0.95\%$ |
| | PGD-(0.3, 20, 0.03) | $0.02 \pm 0.03\%$ | $\mathbf{30.51 \pm 1.22}\%$ | $0.05 \pm 0.02\%$ | $0.01 \pm 0.02\%$ |
| PN-Conv | Clean | $86.36 \pm 0.21\%$ | $86.38 \pm 0.26\%$ | $84.69 \pm 0.44\%$ | $\mathbf{86.45 \pm 0.21}\%$ |
| | FGSM-0.1 | $10.80 \pm 1.82\%$ | $\mathbf{48.15 \pm 1.23}\%$ | $10.62 \pm 0.77\%$ | $10.73 \pm 1.58\%$ |
| | PGD-(0.1, 20, 0.01) | $9.37 \pm 1.00\%$ | $\mathbf{46.63 \pm 3.68}\%$ | $10.20 \pm 0.32\%$ | $8.96 \pm 0.83\%$ |
| | PGD-(0.3, 20, 0.03) | $1.75 \pm 0.83\%$ | $\mathbf{28.94 \pm 1.20}\%$ | $8.26 \pm 1.05\%$ | $2.03 \pm 0.99\%$ |
| | | | *E-MNIST* | | |
| PN-4 | Clean | $84.27 \pm 0.26\%$ | $\mathbf{84.34 \pm 0.31}\%$ | $81.99 \pm 0.33\%$ | $84.22 \pm 0.33\%$ |
| | FGSM-0.1 | $8.92 \pm 1.99\%$ | $\mathbf{27.56 \pm 3.32}\%$ | $14.96 \pm 1.32\%$ | $8.18 \pm 3.48\%$ |
| | PGD-(0.1, 20, 0.01) | $6.24 \pm 1.43\%$ | $\mathbf{29.46 \pm 2.73}\%$ | $6.75 \pm 2.92\%$ | $5.93 \pm 1.97\%$ |
| | PGD-(0.3, 20, 0.03) | $1.22 \pm 0.85\%$ | $\mathbf{19.07 \pm 0.98}\%$ | $3.06 \pm 0.53\%$ | $1.00 \pm 0.76\%$ |
| PN-10 | Clean | $89.31 \pm 0.09\%$ | $\mathbf{90.56 \pm 0.10}\%$ | $89.19 \pm 0.07\%$ | $89.23 \pm 0.13\%$ |
| | FGSM-0.1 | $15.56 \pm 1.16\%$ | $\mathbf{37.11 \pm 2.81}\%$ | $24.21 \pm 1.89\%$ | $16.30 \pm 1.82\%$ |
| | PGD-(0.1, 20, 0.01) | $2.63 \pm 0.65\%$ | $\mathbf{37.89 \pm 2.91}\%$ | $9.18 \pm 1.09\%$ | $2.33 \pm 0.43\%$ |
| | PGD-(0.3, 20, 0.03) | $0.00 \pm 0.00\%$ | $\mathbf{20.47 \pm 0.96}\%$ | $0.11 \pm 0.08\%$ | $0.02 \pm 0.03\%$ |
| PN-Conv | Clean | $91.49 \pm 0.29\%$ | $\mathbf{91.57 \pm 0.19}\%$ | $90.38 \pm 0.13\%$ | $91.41 \pm 0.18\%$ |
| | FGSM-0.1 | $4.28 \pm 0.55\%$ | $\mathbf{35.39 \pm 7.51}\%$ | $3.88 \pm 0.04\%$ | $4.13 \pm 0.41\%$ |
| | PGD-(0.1, 20, 0.01) | $3.98 \pm 0.82\%$ | $\mathbf{33.75 \pm 7.17}\%$ | $3.86 \pm 0.01\%$ | $4.83 \pm 0.87\%$ |
| | PGD-(0.3, 20, 0.03) | $3.24 \pm 0.76\%$ | $\mathbf{28.10 \pm 3.27}\%$ | $3.84 \pm 0.01\%$ | $2.76 \pm 0.65\%$ |

Table 1: Comparison of regularization techniques on Fashion-MNIST (top) and E-MNIST (bottom). In each dataset, the base networks are PN-4, i.e., a 4[th] degree polynomial, on the top four rows, PN-10, i.e., a 10[th] degree polynomial, on the middle four rows and PN-Conv, i.e., a 4[th] degree polynomial with convolutions, on the bottom four rows. Our projection method exhibits the best performance in all three attacks, with the difference on accuracy to stronger attacks being substantial.

attack, the bigger the difference of the proposed regularization scheme with the alternatives of Jacobian and $L_2$ regularizations.

Next, we learn the networks with varying projection bounds. The results on Fashion-MNIST and E-MNIST are visualized in Fig. 2, where the x-axis is plotted in log-scale. As a reference point, we include the clean accuracy curves, i.e., when there is no adversarial noise. Projection bounds larger than 2 (in the log-axis) leave the accuracy unchanged. As the bounds decrease, the results gradually improve. This can be attributed to the constraints the projection bounds impose into the $U_i$ matrices.

Similar observations can be made when evaluating the clean accuracy (i.e., no adversarial noise in the test set). However, in the case of adversarial attacks a tighter bound performs better, i.e., the best accuracy is exhibited in the region of 0 in the log-axis. The projection bounds can have a substantial improvement on the performance, especially in the case of stronger adversarial attacks, i.e., PGD. Notice that all in the aforementioned cases, the intermediate values of the projection bounds yield an increased performance in terms of the test-accuracy and the adversarial perturbations.

Beyond the aforementioned datasets, we also validate the proposed method on CIFAR-10 dataset. The results in Fig. 3 and Table 2 exhibit similar patterns as the aforementioned experiments. Although the improvement is smaller than the case of Fashion-MNIST and E-MNIST, we can still obtain about 10% accuracy improvement under three different adversarial attacks.

## 4.3 ADVERSARIAL TRAINING (AT) ON PNS

Adversarial training has been used as a strong defence against adversarial attacks. In this experiment we evaluate whether different regularization methods can work in conjunction with adversarial training that is widely used as a defence method. Since multi-step adversarial attacks are computationally intensive, we utilize the FGSM attack during training, while we evaluate the trained model in all three

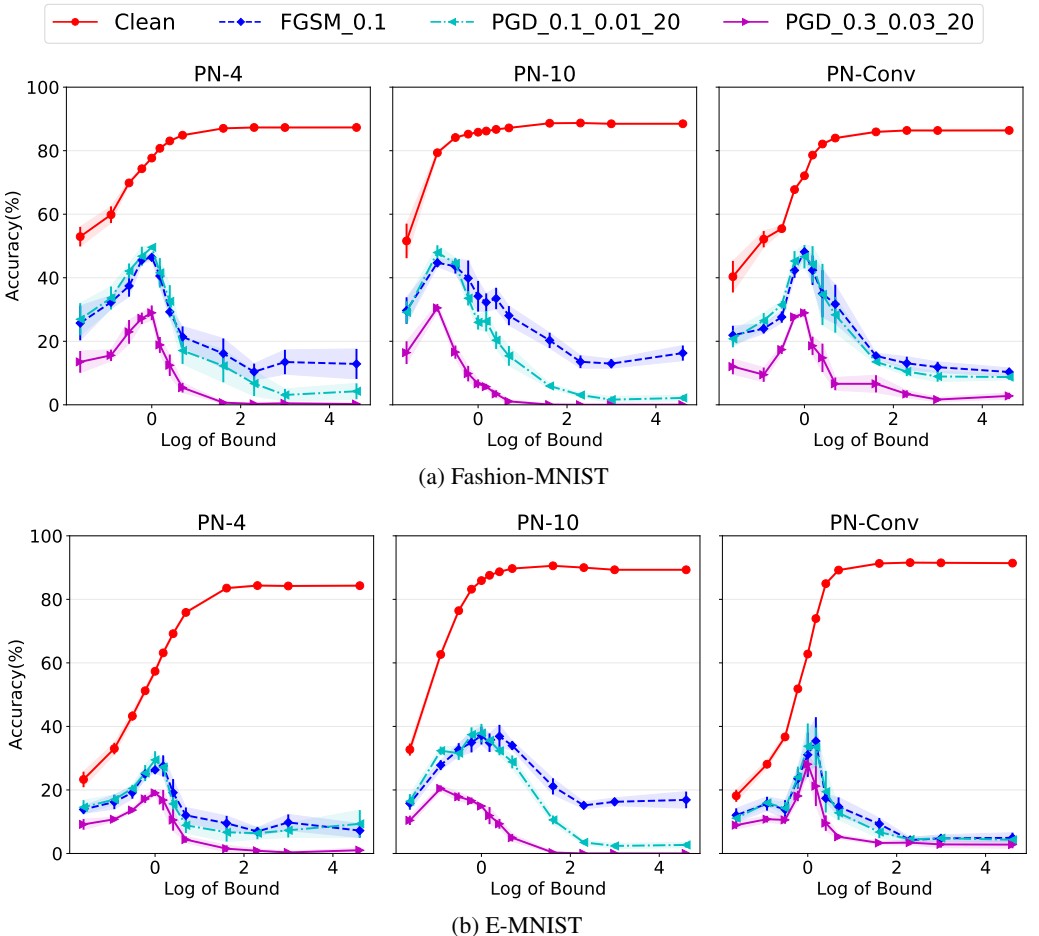

Figure 2: Adversarial attacks during testing on (a) Fashion-MNIST (top), (b) E-MNIST (bottom) with the x-axis is plotted in log-scale. Note that intermediate values of projection bounds yield the highest accuracy. The patterns are consistent in both datasets and across adversarial attacks.

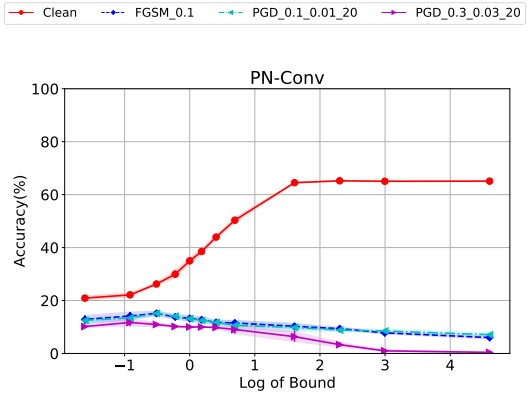

Figure 3: Adversarial attacks during testing on CIFAR-10.

| Model | PN-Conv | | | |
|---|---|---|---|---|
| Projection | No-proj | Our method | Jacobian | $L_2$ |
| Clean accuracy | $65.09 \pm 0.14\%$ | $\mathbf{65.22 \pm 0.13}\%$ | $64.43 \pm 0.19\%$ | $65.11 \pm 0.08\%$ |
| FGSM-0.1 | $6.00 \pm 0.53\%$ | $\mathbf{15.13 \pm 0.81}\%$ | $3.34 \pm 0.40\%$ | $1.27 \pm 0.10\%$ |
| PGD-(0.1, 20, 0.01) | $7.08 \pm 0.68\%$ | $\mathbf{15.17 \pm 0.88}\%$ | $1.74 \pm 0.14\%$ | $1.05 \pm 0.05\%$ |
| PGD-(0.3, 20, 0.03) | $0.41 \pm 0.09\%$ | $\mathbf{11.71 \pm 1.11}\%$ | $0.04 \pm 0.02\%$ | $0.51 \pm 0.04\%$ |

Table 2: Evaluation of the robustness of PN models on CIFAR-10. Each line refers to a different adversarial attack. The projection offers an improvement in the accuracy in each case; in PGD attacks the projection improves the accuracy by a significant margin.

adversarial attacks. For this experiment we select PN-10 as the base model. The proposed model implements Algorithm 2.

The accuracy is reported in Table 3 with Fashion-MNIST on the top and E-MNIST on the bottom. In the FGSM attack, the difference of the compared methods is smaller, which is expected since similar attack is used for the training. However, for stronger attacks the difference becomes pronounced with the proposed regularization method outperforming both the Jacobian and the $L_2$ regularization methods.

| Method | AT | Our method + AT | Jacobian + AT | $L_2$ + AT |
|---|---|---|---|---|
| | Adversarial training (AT) with PN-10 on *Fashion-MNIST* | | | |
| FGSM-0.1 | $65.33 \pm 0.46\%$ | $\mathbf{65.64 \pm 0.35}\%$ | $62.04 \pm 0.22\%$ | $65.62 \pm 0.15\%$ |
| PGD-(0.1, 20, 0.01) | $57.45 \pm 0.35\%$ | $\mathbf{59.89 \pm 0.22}\%$ | $57.42 \pm 0.24\%$ | $57.40 \pm 0.36\%$ |
| PGD-(0.3, 20, 0.03) | $24.46 \pm 0.45\%$ | $\mathbf{39.79 \pm 1.40}\%$ | $25.59 \pm 0.20\%$ | $24.99 \pm 0.57\%$ |
| | Adversarial training (AT) with PN-10 on *E-MNIST* | | | |
| FGSM-0.1 | $78.30 \pm 0.18\%$ | $\mathbf{78.61 \pm 0.58}\%$ | $70.11 \pm 0.18\%$ | $78.31 \pm 0.32\%$ |
| PGD-(0.1, 20, 0.01) | $68.40 \pm 0.32\%$ | $\mathbf{68.51 \pm 0.19}\%$ | $64.61 \pm 0.16\%$ | $68.41 \pm 0.37\%$ |
| PGD-(0.3, 20, 0.03) | $35.58 \pm 0.33\%$ | $\mathbf{42.22 \pm 0.60}\%$ | $39.83 \pm 0.24\%$ | $35.17 \pm 0.46\%$ |

Table 3: Comparison of regularization techniques on (a) Fashion-MNIST (top) and (b) E-MNIST (bottom) along with adversarial training (AT). The base network is a PN-10, i.e., $10^{\text{th}}$ degree polynomial. Our projection method exhibits the best performance in all three attacks.

The limitations of the proposed work are threefold. Firstly, Theorem 1 relies on the $\ell_\infty$-operator norm of the face-splitting product of the weight matrices, which in practice we relax in Lemma 1 for performing the projection. In the future, we aim to study if it is feasible to compute the non-convex projection onto the set of PNs with bounded $\ell_\infty$-norm of the face-splitting product of the weight matrices. This would allow us to let go off the relaxation argument and directly optimize the original tighter Rademacher Complexity bound (Theorem 1).

Secondly, the regularization effect of the projection differs across datasets and adversarial attacks, a topic that is worth investigating in the future.

Thirdly, our bounds do not take into account the algorithm used, which corresponds to a variant of the Stochastic Projected Gradient Descent, and hence any improved generalization properties due to possible uniform stability (Bousquet and Elisseeff, 2002) of the algorithm or implicit regularization properties (Neyshabur, 2017), do not play a role in our analysis.

## 5 CONCLUSION

In this work, we explore the generalization properties of the Coupled CP-decomposition (CCP) and nested coupled CP-decomposition (NCP) models that belong in the class of Polynomial Nets (PNs). We derive bounds for the Rademacher complexity and the Lipschitz constant of the CCP and the NCP models. We utilize the computed bounds as a regularization during training. The regularization terms have also a substantial effect on the robustness of the model, i.e., when adversarial noise is added to the test set. A future direction of research is to obtain generalization bounds for this class of functions using stability notions. Along with the recent empirical results on PNs, our derived bounds can further explain the benefits and drawbacks of using PNs.

ACKNOWLEDGEMENTS

We are thankful to Igor Krawczuk and Andreas Loukas for their comments on the paper. We are also thankful to the reviewers for providing constructive feedback. Research was sponsored by the Army Research Office and was accomplished under Grant Number W911NF-19-1-0404. This work is funded (in part) through a PhD fellowship of the Swiss Data Science Center, a joint venture between EPFL and ETH Zurich. This project has received funding from the European Research Council (ERC) under the European Union's Horizon 2020 research and innovation programme (grant agreement number 725594 - time-data).

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

## A    APPENDIX INTRODUCTION

The Appendix is organized as follows:

- The related work is summarized in Appendix B.
- In Appendix C the notation and the core Lemmas from the literature are described.
- Further details on the Polynomial Nets are provided in Appendix D.
- The proofs on the CCP and the NCP models are added in Appendix E and Appendix F respectively.
- The extensions of the theorems for convolutional layers and their proofs are detailed in Appendix G.
- Additional experiments are included in Appendix H.

## B    RELATED WORK

**Rademacher Complexity:** Known bounds for the class of polynomials are a consequence of more general result for kernel methods (Mohri et al., 2018, Theorem 6.12). Support Vector Machines (SVMs) with a polynomial kernel of degree $k$ effectively correspond to a general polynomial with the same degree. In contrast, our bound is tailored to the parametric definition of the CCP and the NCP models, which are a subset of the class of all polynomials. Hence, they are tighter than the general kernel complexity bounds.

Bounds for the class of neural networks were stablished in (Bartlett et al., 2017; Neyshabur et al., 2017), but they require a long and technical proof, and in particular it assumes an $\ell_2$-bound on the input, which is incompatible with image-based applications. This bound also depend on the product of spectral norms of each layer. In contrast, our bounds are more similar in spirit to the path-norm-based complexity bounds (Neyshabur et al., 2015), as they depend on interactions between neurons at different layers. This interaction precisely corresponds to the face-splitting product between weight matrices that appears in Theorem 1.

**Lipschitz constant:** A variety of methods have been proposed for estimating the Lipschitz constant of neural networks. For example, Scaman and Virmaux (2018) (SVD), Fazlyab et al. (2019) (Semidefinite programming) and Latorre et al. (2020) (Polynomial Optimization) are expensive optimization methods to compute tighter bounds on such constant. These methods are unusable in our case as they would require a non-trivial adaptation to work with Polynomial Nets. In contrast we find an upper bound that applies to such family of models, and it can be controlled with efficient $\ell_\infty$-operator-norm projections. However, our bounds might not be the tightest. Developing tighter methods to bound and control the Lipschitz constant for Polynomial Nets is a promising direction of future research.

## C    BACKGROUND

Below, we develop a detailed notation in Appendix C.1, we include related definitions in Appendix C.2 and Lemmas required for our proofs in Appendix C.3. The goal of this section is to cover many of the required information for following the proofs and the notation we follow in this work. Readers familiar with the different matrix/vector products, e.g., Khatri-Rao or face-splitting product, and with basic inequalities, e.g., Hölder's inequality, can skip to the next section.

### C.1    NOTATION

Different matrix products and their associated symbols are referenced in Table 4, while matrix operations on a matrix $\boldsymbol{A}$ are defined on Table 5. Every matrix product, e.g., Hadamard product, can be used in two ways: a) $\boldsymbol{A} \circ \boldsymbol{B}$, which translates to Hadamard product of matrices $\boldsymbol{A}$ and $\boldsymbol{B}$, b) $\circ_{i=1}^{N} \boldsymbol{A}_i$ abbreviates the Hadamard products $\underbrace{\boldsymbol{A}_1 \circ \boldsymbol{A}_2 \circ \ldots \boldsymbol{A}_N}_{N \text{ products}}$.

The symbol $x_i^j$ refers to $j^{\text{th}}$ element of vector $\boldsymbol{x}_i$.

Table 4: Symbols for various matrix products. The precise definitions of the products are included in Appendix C.2 for completion.

| Symbol | Definition |
|:---:|:---:|
| $\circ$ | Hadamard (element-wise) product. |
| $*$ | Column-wise Khatri–Rao product. |
| $\bullet$ | Face-splitting product. |
| $\otimes$ | Kronecker product. |
| $\star$ | Convolution. |

Table 5: Operations and symbols on a matrix $\boldsymbol{A}$.

| Symbol | Definition |
|:---:|:---:|
| $\|\boldsymbol{A}\|_\infty$ | $\ell_\infty$-operator-norm; corresponds to the maximum $\ell_1$-norm of its rows. |
| $\boldsymbol{A}^i$ | $i^{\text{th}}$ row of $\boldsymbol{A}$. |
| $\boldsymbol{a}_{i,j}$ | $(i,j)^{\text{th}}$ element of $\boldsymbol{A}$. |
| $\boldsymbol{A}_{i,j}$ | $(i,j)^{\text{th}}$ block of a block-matrix $\boldsymbol{A}$ |
| $\boldsymbol{A}_i$ | The i-th matrix in a set of matrices $\{\boldsymbol{A}_1, \cdots, \boldsymbol{A}_N\}$. |

### C.2 DEFINITIONS

For thoroughness, we include the definitions of the core products that we use in this work. Specifically, the definitions of the Hadamard product (Definition 3), Kronecker product (Definition 4), the Khatri-Rao product (Definition 5), column-wise Khatri-Rao product (Definition 6) and the face-splitting product (Definition 7) are included.

**Definition 3** (Hadamard product). *For two matrices $\boldsymbol{A}$ and $\boldsymbol{B}$ of the same dimension $m \times n$, the Hadamard product $\boldsymbol{A} \circ \boldsymbol{B}$ is a matrix of the same dimension as the operands, with elements given by*

$$(a \circ b)_{i,j} = a_{i,j} b_{i,j}.$$

**Definition 4** (Kronecker product). *If $\boldsymbol{A}$ is an $m \times n$ matrix and $\boldsymbol{B}$ is a $p \times q$ matrix, then the Kronecker product $\boldsymbol{A} \otimes \boldsymbol{B}$ is the $pm \times qn$ block matrix, given as follows:*

$$\boldsymbol{A} \otimes \boldsymbol{B} = \begin{bmatrix} a_{1,1}\boldsymbol{B} & \cdots & a_{1,n}\boldsymbol{B} \\ \vdots & \ddots & \vdots \\ a_{m,1}\boldsymbol{B} & \cdots & a_{m,n}\boldsymbol{B} \end{bmatrix}.$$

Example: the Kronecker product of the matrices $\boldsymbol{A} \in \mathbb{R}^{2\times2}$ and $\boldsymbol{B} \in \mathbb{R}^{2\times2}$ is computed below:

$$\underbrace{\begin{bmatrix} a_{1,1} & a_{1,2} \\ a_{2,1} & a_{2,2} \end{bmatrix}}_{\boldsymbol{A}} \otimes \underbrace{\begin{bmatrix} b_{1,1} & b_{1,2} \\ b_{2,1} & b_{2,2} \end{bmatrix}}_{\boldsymbol{B}} = \underbrace{\begin{bmatrix} a_{1,1}b_{1,1} & a_{1,1}b_{1,2} & a_{1,2}b_{1,1} & a_{1,2}b_{1,2} \\ a_{1,1}b_{2,1} & a_{1,1}b_{2,2} & a_{1,2}b_{2,1} & a_{1,2}b_{2,2} \\ a_{2,1}b_{1,1} & a_{2,1}b_{1,2} & a_{2,2}b_{1,1} & a_{2,2}b_{1,2} \\ a_{2,1}b_{2,1} & a_{2,1}b_{2,2} & a_{2,2}b_{2,1} & a_{2,2}b_{2,2} \end{bmatrix}}_{\boldsymbol{A} \otimes \boldsymbol{B}}.$$

**Definition 5** (Khatri–Rao product). *The Khatri–Rao product is defined as:*

$$\boldsymbol{A} * \boldsymbol{B} = (A_{i,j} \otimes B_{i,j})_{i,j},$$

*in which the $(i,j)$-th block is the $m_i p_i \times n_j q_j$ sized Kronecker product of the corresponding blocks of $\boldsymbol{A}$ and $\boldsymbol{B}$, assuming the number of row and column partitions of both matrices is equal. The size of the product is then $(\sum_i m_i p_i) \times (\sum_i n_j q_j)$.*

Example: if $\boldsymbol{A}$ and $\boldsymbol{B}$ both are $2 \times 2$ partitioned matrices e.g.:

$$\boldsymbol{A} = \left[\begin{array}{c|c} \boldsymbol{A}_{1,1} & \boldsymbol{A}_{1,2} \\ \hline \boldsymbol{A}_{2,1} & \boldsymbol{A}_{2,2} \end{array}\right] = \left[\begin{array}{cc|c} a_{1,1} & a_{1,2} & a_{1,3} \\ a_{2,1} & a_{2,2} & a_{2,3} \\ \hline a_{3,1} & a_{3,2} & a_{3,3} \end{array}\right],$$

$$
\boldsymbol{B} = \left[ \begin{array}{c|c} \boldsymbol{B}_{1,1} & \boldsymbol{B}_{1,2} \\ \hline \boldsymbol{B}_{2,1} & \boldsymbol{B}_{2,2} \end{array} \right] = \left[ \begin{array}{c|cc} b_{1,1} & b_{1,2} & b_{1,3} \\ b_{2,1} & b_{2,2} & b_{2,3} \\ \hline b_{3,1} & b_{3,2} & b_{3,3} \end{array} \right] ,
$$

then we obtain the following:

$$
\boldsymbol{A} * \boldsymbol{B} = \left[ \begin{array}{c|c} \boldsymbol{A}_{1,1} \otimes \boldsymbol{B}_{1,1} & \boldsymbol{A}_{1,2} \otimes \boldsymbol{B}_{1,2} \\ \hline \boldsymbol{A}_{2,1} \otimes \boldsymbol{B}_{2,1} & \boldsymbol{A}_{2,2} \otimes \boldsymbol{B}_{2,2} \end{array} \right] = \left[ \begin{array}{cc|cc} a_{1,1}b_{1,1} & a_{1,2}b_{1,1} & a_{1,3}b_{1,2} & a_{1,3}b_{1,3} \\ a_{2,1}b_{1,1} & a_{2,2}b_{1,1} & a_{2,3}b_{1,2} & a_{2,3}b_{1,3} \\ \hline a_{3,1}b_{2,1} & a_{3,2}b_{2,1} & a_{3,3}b_{2,2} & a_{3,3}b_{2,3} \\ a_{3,1}b_{3,1} & a_{3,2}b_{3,1} & a_{3,3}b_{3,2} & a_{3,3}b_{3,3} \end{array} \right] .
$$

**Definition 6** (Column-wise Khatri–Rao product). *A column-wise Kronecker product of two matrices may also be called the Khatri–Rao product. This product assumes the partitions of the matrices are their columns. In this case $m_1 = m$, $p_1 = p$, $n = q$ and for each j: $n_j = p_j = 1$. The resulting product is a $mp \times n$ matrix of which each column is the Kronecker product of the corresponding columns of $\boldsymbol{A}$ and $\boldsymbol{B}$.*

Example: the Column-wise Khatri–Rao product of the matrices $\boldsymbol{A} \in \mathbb{R}^{2 \times 3}$ and $\boldsymbol{B} \in \mathbb{R}^{3 \times 3}$ is computed below:

$$
\underbrace{\begin{bmatrix} a_{1,1} & a_{1,2} & a_{1,3} \\ a_{2,1} & a_{2,2} & a_{2,3} \end{bmatrix}}_{\boldsymbol{A}} * \underbrace{\begin{bmatrix} b_{1,1} & b_{1,2} & b_{1,3} \\ b_{2,1} & b_{2,2} & b_{2,3} \\ b_{3,1} & b_{3,2} & b_{3,3} \end{bmatrix}}_{\boldsymbol{B}} = \underbrace{\begin{bmatrix} a_{1,1}b_{1,1} & a_{1,2}b_{1,2} & a_{1,3}b_{1,3} \\ a_{1,1}b_{2,1} & a_{1,2}b_{2,2} & a_{1,3}b_{2,3} \\ a_{1,1}b_{3,1} & a_{1,2}b_{3,2} & a_{1,3}b_{3,3} \\ a_{2,1}b_{1,1} & a_{2,2}b_{1,2} & a_{2,3}b_{1,3} \\ a_{2,1}b_{2,1} & a_{2,2}b_{2,2} & a_{2,3}b_{2,3} \\ a_{2,1}b_{3,1} & a_{2,2}b_{3,2} & a_{2,3}b_{3,3} \end{bmatrix}}_{\boldsymbol{A}*\boldsymbol{B}} .
$$

From here on, all $*$ refer to Column-wise Khatri–Rao product.

**Definition 7** (Face-splitting product). *The alternative concept of the matrix product, which uses row-wise splitting of matrices with a given quantity of rows. This matrix operation was named the* face-splitting product *of matrices or the* transposed Khatri–Rao product*. This type of operation is based on row-by-row Kronecker products of two matrices.*

Example: the Face-splitting product of the matrices $\boldsymbol{A} \in \mathbb{R}^{3 \times 2}$ and $\boldsymbol{B} \in \mathbb{R}^{3 \times 2}$ is computed below:

$$
\underbrace{\begin{bmatrix} a_{1,1} & a_{1,2} \\ a_{2,1} & a_{3,2} \\ a_{3,1} & a_{3,2} \end{bmatrix}}_{\boldsymbol{A}} \bullet \underbrace{\begin{bmatrix} b_{1,1} & b_{1,2} \\ b_{2,1} & b_{2,2} \\ b_{3,1} & b_{3,2} \end{bmatrix}}_{\boldsymbol{B}} = \underbrace{\begin{bmatrix} a_{1,1}b_{1,1} & a_{1,2}b_{1,1} & a_{1,1}b_{1,2} & a_{1,2}b_{1,2} \\ a_{2,1}b_{2,1} & a_{2,2}b_{2,1} & a_{2,1}b_{2,2} & a_{2,2}b_{2,2} \\ a_{3,1}b_{3,1} & a_{3,2}b_{3,1} & a_{3,1}b_{3,2} & a_{3,2}b_{3,2} \end{bmatrix}}_{\boldsymbol{A}\bullet\boldsymbol{B}} .
$$

## C.3 Well-known Lemmas

In this section, we provide the details on the Lemmas required for our proofs along with their proofs or the corresponding citations where the Lemmas can be found as well.

**Lemma 3.** *(Federer, 2014) Let $g$, $h$ be two composable Lipschitz functions. Then $g \circ h$ is also Lipschitz with $Lip(g \circ h) \leq Lip(g)Lip(h)$. Here and only here $\circ$ represents function composition.*

**Lemma 4.** *(Federer, 2014) Let $f : \mathcal{X} \subseteq \mathbb{R}^n \to \mathbb{R}^m$ be differentiable and Lipschitz continuous. Let $J_f(\boldsymbol{x})$ denote its total derivative (Jacobian) at $\boldsymbol{x}$. Then $Lip_p(f) = \sup_{\boldsymbol{x} \in \mathcal{X}} \|J_f(\boldsymbol{x})\|_p$ where $\|J_f(\boldsymbol{x})\|_p$ is the induced operator norm on $J_f(\boldsymbol{x})$.*

**Lemma 5** (Hölder's inequality). *(Cvetkovski, 2012) Let $(S, \Sigma, \mu)$ be a measure space and let $p, q \in [1, \infty]$ with $\frac{1}{p} + \frac{1}{q} = 1$. Then, for all measurable real-valued functions $f$ and $g$ on S, it holds that:*

$$
\|\boldsymbol{f}\boldsymbol{g}\|_1 \leq \|\boldsymbol{f}\|_p \|\boldsymbol{g}\|_q .
$$

**Lemma 6** (Mixed Product Property 1). *(Slyusar, 1999) The following holds:*

$$(A_1 B_1) \circ (A_2 B_2) = (A_1 \bullet A_2)(B_1 * B_2).$$

**Lemma 7** (Mixed Product Property 2). *The following holds:*

$$\circ_{i=1}^N (A_i B_i) = \bullet_{i=1}^N (A_i) *_{i=1}^N (B_i).$$

*Proof.* We prove this lemma by induction on $N$.

Base case ($N = 1$): $A_1 B_1 = A_1 B_1$.

Inductive step: Assume that the induction hypothesis holds for a particular $k$, i.e., the case $N = k$ holds. That can be expressed as:

$$\circ_{i=1}^k (A_i B_i) = \bullet_{i=1}^k (A_i) *_{i=1}^k (B_i). \tag{3}$$

Then we will prove that it holds for $N = k + 1$:

$$
\begin{aligned}
\circ_{i=1}^{k+1} (A_i B_i) \\
&= [\circ_{i=1}^k (A_i B_i)] \circ (A_{k+1} B_{k+1}) \\
&= [\bullet_{i=1}^k (A_i) *_{i=1}^k (B_i)] \circ (A_{k+1} B_{k+1}) \quad \text{use inductive hypothesis (Eq. (3))} \\
&= [\bullet_{i=1}^k (A_i) \bullet A_{k+1}][*_{i=1}^k (B_i) * B_{k+1}] \quad \text{Lemma 6 [Mixed product property 1]} \\
&= \bullet_{i=1}^{k+1} (A_i) *_{i=1}^{k+1} (B_i).
\end{aligned}
$$

That is, the case $N = k + 1$ also holds true, establishing the inductive step. □

**Lemma 8** (Massart Lemma. Lemma 26.8 in Shalev-Shwartz and Ben-David (2014)). *Let $\mathcal{A}$ $= \{a_1, \cdots, a_N\}$ be a finite set of vectors in $\mathbb{R}^m$. Define $\bar{a} = \frac{1}{N} \sum_{i=1}^N a_i$. Then:*

$$\mathcal{R}(\mathcal{A}) \leq \max_{a \in A} \|a - \bar{a}\| \frac{\sqrt{2 \log N}}{m}.$$

**Definition 8** (Consistency of a matrix norm). *A matrix norm is called consistent on $\mathbb{C}^{n,n}$, if*

$$\|AB\| \leq \|A\| \|B\|.$$

*holds for $A, B \in \mathbb{C}^{n,n}$.*

**Lemma 9** (Consistency of the operator norm). *(Lyche, 2020) The operator norm is consistent if the vector norm $\|\cdot\|_\alpha$ is defined for all $m \in \mathbb{N}$ and $\|\cdot\|_\beta = \|\cdot\|_\alpha$*

*Proof.*

$$
\begin{aligned}
\|AB\| &= \max_{Bx \neq 0} \frac{\|ABx\|_\alpha}{\|x\|_\alpha} = \max_{Bx \neq 0} \frac{\|ABx\|_\alpha}{\|Bx\|_\alpha} \frac{\|Bx\|_\alpha}{\|x\|_\alpha} \\
&\leq \max_{y \neq 0} \frac{\|Ay\|_\alpha}{\|y\|_\alpha} \max_{x \neq 0} \frac{\|Bx\|_\alpha}{\|x\|_\alpha} = \|A\| \|B\|.
\end{aligned}
$$

□

**Lemma 10.** *(Rao, 1970)*

$$(AC) * (BD) = (A \otimes B)(C * D).$$

## D DETAILS ON POLYNOMIAL NETWORKS

In this section, we provide further details on the two most prominent parametrizations proposed in Chrysos et al. (2020). This re-parametrization creates equivalent models, but enables us to absorb the bias terms into the input terms. Firstly, we provide the re-parametrization of the CCP model in Appendix D.1 and then we create the re-parametrization of the NCP model in Appendix D.2.

## D.1 RE-PARAMETRIZATION OF CCP MODEL

The Coupled CP-Decomposition (CCP) model of PNs (Chrysos et al., 2020) leverages a coupled CP Tensor decomposition. A $k$-degree CCP model $f(\zeta)$ can be succinctly described by the following recursive relations:

$$\boldsymbol{y}_1 = \boldsymbol{V}_1 \zeta, \qquad \boldsymbol{y}_n = (\boldsymbol{V}_n \zeta) \circ \boldsymbol{y}_{n-1} + \boldsymbol{y}_{n-1}, \qquad f(\zeta) = \boldsymbol{Q}\boldsymbol{y}_k + \boldsymbol{\beta}, \qquad (4)$$

where $\zeta \in \mathbb{R}^\delta$ is the input data with $\delta \in \mathbb{N}$, $f(\zeta) \in \mathbb{R}^o$ is the output of the model and $\boldsymbol{V}_n \in \mathbb{R}^{\mu \times \delta}$, $\boldsymbol{Q} \in \mathbb{R}^{o \times \mu}$ and $\boldsymbol{\beta} \in \mathbb{R}^o$ are the learnable parameters, where $\mu \in \mathbb{N}$ is the hidden rank. In order to simplify the bias terms in the model, we will introduce a minor re-parametrization in Lemma 11 that we will use to present our results in the subsequent sections.

**Lemma 11.** *Let* $\boldsymbol{z} = [\zeta^\top, 1]^\top \in \mathbb{R}^d$, $\boldsymbol{x}_n = [\boldsymbol{y}_n^\top, 1]^\top \in \mathbb{R}^m$, $\boldsymbol{C} = [\boldsymbol{Q}, \boldsymbol{\beta}] \in \mathbb{R}^{o \times m}$, $d = \delta + 1$, $m = \mu + 1$. *Define:*

$$\boldsymbol{U}_1 = \begin{bmatrix} \boldsymbol{V}_1 & \boldsymbol{0} \\ \boldsymbol{0}^\top & 1 \end{bmatrix} \in \mathbb{R}^{m \times d}, \qquad \boldsymbol{U}_i = \begin{bmatrix} \boldsymbol{V}_i & \boldsymbol{1} \\ \boldsymbol{0}^\top & 1 \end{bmatrix} \in \mathbb{R}^{m \times d} \qquad (i > 1).$$

*where the boldface numbers* $\boldsymbol{0}$ *and* $\boldsymbol{1}$ *denote all-zeros and all-ones column vectors of appropriate size, respectively. The CCP model in Eq. (4) can be rewritten as* $f(\boldsymbol{z}) = \boldsymbol{C} \circ_{i=1}^k \boldsymbol{U}_i \boldsymbol{z}$, *which is the one used in Eq. (CCP).*

As a reminder before providing the proof, the core symbols for this proof are summarized in Table 6.

Table 6: Core symbols in the proof of Lemma 11.

| Symbol | Dimensions | Definition |
|:---:|:---:|:---:|
| $\circ$ | - | Hadamard (element-wise) product. |
| $\zeta$ | $\mathbb{R}^\delta$ | Input of the polynomial expansion. |
| $f(\zeta)$ | $\mathbb{R}^o$ | Output of the polynomial expansion. |
| $k$ | $\mathbb{N}$ | Degree of polynomial expansion. |
| $m$ | $\mathbb{N}$ | Hidden rank of the expansion. |
| $\boldsymbol{V}_n$ | $\mathbb{R}^{\mu \times \delta}$ | Learnable matrices of the expansion. |
| $\boldsymbol{Q}$ | $\mathbb{R}^{o \times \mu}$ | Learnable matrix of the expansion. |
| $\boldsymbol{\beta}$ | $\mathbb{R}^o$ | Bias of the expansion. |
| $\boldsymbol{z}$ | $\mathbb{R}^d$ | Re-parametrization of the input. |
| $\boldsymbol{C}$ | $\mathbb{R}^{o \times m}$ | $\boldsymbol{C} = (\boldsymbol{Q}, \boldsymbol{\beta})$. |

*Proof.* By definition, we have:

$$\boldsymbol{x}_1 = [\boldsymbol{y}_1^\top, 1]^\top = \begin{bmatrix} \boldsymbol{y}_1 \\ 1 \end{bmatrix} = \begin{bmatrix} \boldsymbol{V}_1 \zeta \\ 1 \end{bmatrix} = \begin{bmatrix} \boldsymbol{V}_1 & \boldsymbol{0} \\ \boldsymbol{0}^\top & 1 \end{bmatrix} \begin{bmatrix} \zeta \\ 1 \end{bmatrix} = \begin{bmatrix} \boldsymbol{V}_1 & \boldsymbol{0} \\ \boldsymbol{0}^\top & 1 \end{bmatrix} [\zeta^\top, 1]^\top = \boldsymbol{U}_1 \boldsymbol{z}.$$

$$\boldsymbol{x}_n = [\boldsymbol{y}_n^\top, 1]^\top = \begin{bmatrix} \boldsymbol{y}_n \\ 1 \end{bmatrix} = \begin{bmatrix} (\boldsymbol{V}_n \zeta) \circ \boldsymbol{y}_{n-1} + \boldsymbol{y}_{n-1} \\ 1 \end{bmatrix} = \begin{bmatrix} \boldsymbol{V}_n \zeta + \boldsymbol{1} \\ 1 \end{bmatrix} \circ \begin{bmatrix} \boldsymbol{y}_{n-1} \\ 1 \end{bmatrix}$$

$$= \begin{bmatrix} \boldsymbol{V}_n & \boldsymbol{1} \\ \boldsymbol{0}^\top & 1 \end{bmatrix} \begin{bmatrix} \zeta \\ 1 \end{bmatrix} \circ \begin{bmatrix} \boldsymbol{y}_{n-1} \\ 1 \end{bmatrix} = \begin{bmatrix} \boldsymbol{V}_n & \boldsymbol{1} \\ \boldsymbol{0}^\top & 1 \end{bmatrix} [\zeta^\top, 1]^\top \circ [\boldsymbol{y}_{n-1}^\top, 1]^\top = \boldsymbol{U}_n \boldsymbol{z} \circ \boldsymbol{x}_{n-1}.$$

Hence, it holds that:

$$f(\boldsymbol{z}) = \boldsymbol{Q}\boldsymbol{y}_k + \boldsymbol{\beta} = (\boldsymbol{Q}, \boldsymbol{\beta}) \begin{bmatrix} \boldsymbol{y}_k \\ 1 \end{bmatrix} = \boldsymbol{C}\boldsymbol{x}_k$$

$$= \boldsymbol{C}\,\boldsymbol{U}_k \boldsymbol{z} \circ \boldsymbol{x}_{k-1}$$
$$= \boldsymbol{C}\,\boldsymbol{U}_k \boldsymbol{z} \circ (\boldsymbol{U}_{k-1} \boldsymbol{z}) \circ \boldsymbol{x}_{k-2}$$
$$= \cdots$$
$$= \boldsymbol{C}\,\boldsymbol{U}_k \boldsymbol{z} \circ (\boldsymbol{U}_{k-1} \boldsymbol{z}) \circ \cdots \circ (\boldsymbol{U}_2 \boldsymbol{z}) \circ \boldsymbol{x}_1$$
$$= \boldsymbol{C}\,\boldsymbol{U}_k \boldsymbol{z} \circ (\boldsymbol{U}_{k-1} \boldsymbol{z}) \circ \cdots \circ (\boldsymbol{U}_2 \boldsymbol{z}) \circ (\boldsymbol{U}_1 \boldsymbol{z})$$
$$= \boldsymbol{C} \circ_{i=1}^k (\boldsymbol{U}_i \boldsymbol{z}).$$

$\square$

## D.2 REPARAMETRIZATION OF THE NCP MODEL

The nested coupled CP decomposition (NCP) model of PNs (Chrysos et al., 2020) leverages a joint hierarchical decomposition. A *k-degree NCP model* $f(\boldsymbol{\zeta})$ is expressed with the following recursive relations:

$$\boldsymbol{y}_1 = (\boldsymbol{V}_1\boldsymbol{\zeta}) \circ (\boldsymbol{b}_1), \qquad \boldsymbol{y}_n = (\boldsymbol{V}_n\boldsymbol{\zeta}) \circ (\boldsymbol{U}_n\boldsymbol{y}_{n-1} + \boldsymbol{b}_n), \qquad f(\boldsymbol{\zeta}) = \boldsymbol{Q}\boldsymbol{y}_k + \boldsymbol{\beta}. \tag{5}$$

where $\boldsymbol{\zeta} \in \mathbb{R}^\delta$ is the input data with $\delta \in \mathbb{N}$, $f(\boldsymbol{\zeta}) \in \mathbb{R}^o$ is the output of the model and $\boldsymbol{V}_n \in \mathbb{R}^{\mu \times \delta}$, $\boldsymbol{b}_n \in \mathbb{R}^\mu$, $\boldsymbol{U}_n \in \mathbb{R}^{\mu \times \mu}$, $\boldsymbol{Q} \in \mathbb{R}^{o \times \mu}$ and $\boldsymbol{\beta} \in \mathbb{R}^o$ are the learnable parameters, where $\mu \in \mathbb{N}$ is the hidden rank. In order to simplify the bias terms in the model, we will introduce a minor re-parametrization in Lemma 12 that we will use to present our results in the subsequent sections.

**Lemma 12.** *Let* $z = [\boldsymbol{\zeta}^\top, 1]^\top \in \mathbb{R}^d$, $\boldsymbol{x}_n = [\boldsymbol{y}_n^\top, 1]^\top \in \mathbb{R}^m$, $\boldsymbol{C} = [\boldsymbol{Q}, \boldsymbol{\beta}] \in \mathbb{R}^{o \times m}$, $d = \delta + 1, m = \mu + 1$. *Let:*

$$\boldsymbol{s}_1 = [\boldsymbol{b}_1^\top, 1]^\top \in \mathbb{R}^m, \quad \boldsymbol{S}_i = \begin{bmatrix} \boldsymbol{U}_i & \boldsymbol{b}_i \\ \boldsymbol{0}^\top & 1 \end{bmatrix} \in \mathbb{R}^{m \times m}(i > 1), \quad \boldsymbol{A}_i = \begin{bmatrix} \boldsymbol{V}_i & \boldsymbol{0} \\ \boldsymbol{0}^\top & 1 \end{bmatrix} \in \mathbb{R}^{m \times d}.$$

*where the boldface numbers $\boldsymbol{0}$ and $\boldsymbol{1}$ denote all-zeros and all-ones column vectors of appropriate size, respectively. The NCP model in Eq. (5) can be rewritten as*

$$\boldsymbol{x}_1 = (\boldsymbol{A}_1\boldsymbol{z}) \circ (\boldsymbol{s}_1), \qquad \boldsymbol{x}_n = (\boldsymbol{A}_n\boldsymbol{z}) \circ (\boldsymbol{S}_n\boldsymbol{x}_{n-1}), \qquad f(\boldsymbol{z}) = \boldsymbol{C}\boldsymbol{x}_k. \tag{6}$$

*In the aforementioned Eq. (6), we have written $\boldsymbol{S}_n$ even for $n = 1$, when $s_1$ is technically a vector, but this is done for convenience only and does not change the end result.*

## E RESULT OF THE CCP MODEL

### E.1 PROOF OF THEOREM 1: RADEMACHER COMPLEXITY BOUND OF CCP UNDER $\ell_\infty$ NORM

To facilitate the proof below, we include the related symbols in Table 7. Below, to avoid cluttering the notation, we consider that the expectation is over $\sigma$ and omit the brackets as well.

Table 7: Core symbols for proof of Theorem 1.

| Symbol | Dimensions | Definition |
|:---:|:---:|:---:|
| $\circ$ | - | Hadamard (element-wise) product. |
| $\bullet$ | - | Face-splitting product. |
| $*$ | - | Column-wise Khatri–Rao product. |
| $\boldsymbol{z}$ | $\mathbb{R}^d$ | Input of the polynomial expansion. |
| $f(\boldsymbol{z})$ | $\mathbb{R}$ | Output of the polynomial expansion. |
| $k$ | $\mathbb{N}$ | Degree of polynomial expansion. |
| $m$ | $\mathbb{N}$ | Hidden rank of the expansion. |
| $\boldsymbol{U}_i$ | $\mathbb{R}^{m \times d}$ | Learnable matrices. |
| $\boldsymbol{c}$ | $\mathbb{R}^{1 \times m}$ | Learnable matrix. |
| $\mu$ | $\mathbb{R}$ | $\|\boldsymbol{c}\|_1 \leq \mu.$ |
| $\lambda$ | $\mathbb{R}$ | $\left\|\bullet_{i=1}^k (\boldsymbol{U}_i)\right\|_\infty \leq \lambda.$ |

*Proof.*

$$
\begin{aligned}
\mathcal{R}_Z(\mathcal{F}_{\text{CCP}}^k) &= \mathbb{E} \sup_{f \in \mathcal{F}_{\text{CCP}}^k} \frac{1}{n} \sum_{j=1}^n \sigma_j f(\boldsymbol{z}_j) \\
&= \mathbb{E} \sup_{f \in \mathcal{F}_{\text{CCP}}^k} \frac{1}{n} \sum_{j=1}^n \left( \sigma_j \left\langle \boldsymbol{c}, \circ_{i=1}^k (\boldsymbol{U}_i \boldsymbol{z}_j) \right\rangle \right) \\
&= \mathbb{E} \sup_{f \in \mathcal{F}_{\text{CCP}}^k} \frac{1}{n} \left\langle \boldsymbol{c}, \sum_{j=1}^n [\sigma_j \circ_{i=1}^k (\boldsymbol{U}_i \boldsymbol{z}_j)] \right\rangle \\
&\leq \mathbb{E} \sup_{f \in \mathcal{F}_{\text{CCP}}^k} \frac{1}{n} \|\boldsymbol{c}\|_1 \left\| \sum_{j=1}^n [\sigma_j \circ_{i=1}^k (\boldsymbol{U}_i \boldsymbol{z}_j)] \right\|_\infty && \text{Lemma 5 [Hölder's inequality]} \\
&= \mathbb{E} \sup_{f \in \mathcal{F}_{\text{CCP}}^k} \frac{1}{n} \|\boldsymbol{c}\|_1 \left\| \sum_{j=1}^n [\sigma_j \bullet_{i=1}^k (\boldsymbol{U}_i) *_{i=1}^k (\boldsymbol{z}_j)] \right\|_\infty && \text{Lemma 7 [Mixed product property]} \\
&= \mathbb{E} \sup_{f \in \mathcal{F}_{\text{CCP}}^k} \frac{1}{n} \|\boldsymbol{c}\|_1 \left\| \bullet_{i=1}^k (\boldsymbol{U}_i) \sum_{j=1}^n [\sigma_j *_{i=1}^k (\boldsymbol{z}_j)] \right\|_\infty \\
&\leq \mathbb{E} \sup_{f \in \mathcal{F}_{\text{CCP}}^k} \frac{1}{n} \|\boldsymbol{c}\|_1 \left\| \sum_{j=1}^n [\sigma_j *_{i=1}^k (\boldsymbol{z}_j)] \right\|_\infty \left\| \bullet_{i=1}^k (\boldsymbol{U}_i) \right\|_\infty \\
&\leq \frac{\mu \lambda}{n} \mathbb{E} \left\| \sum_{j=1}^n [\sigma_j *_{i=1}^k (\boldsymbol{z}_j)] \right\|_\infty .
\end{aligned}
$$

$$(7)$$

Next, we compute the bound of $\mathbb{E} \left\| \sum_{j=1}^n [\sigma_j *_{i=1}^k (\boldsymbol{z}_j)] \right\|_\infty$.

Let $\boldsymbol{Z}_j = *_{i=1}^k (\boldsymbol{z}_j) \in \mathbb{R}^{d^k}$. For each $l \in [d^k]$, let $\boldsymbol{v}_l = (\boldsymbol{Z}_1^l, \ldots, \boldsymbol{Z}_n^l) \in \mathbb{R}^n$. Note that $\|\boldsymbol{v}_l\|_2 \leq \sqrt{n} \, \max_j \|\boldsymbol{Z}_j\|_\infty$. Let $V = \{\boldsymbol{v}_1, \ldots, \boldsymbol{v}_{d^k}\}$. Then, it is true that:

$$
\mathbb{E} \left\| \sum_{j=1}^n [\sigma_j *_{i=1}^k (\boldsymbol{z}_j)] \right\|_\infty = \mathbb{E} \left\| \sum_{j=1}^n \sigma_j \boldsymbol{Z}_j \right\|_\infty = \mathbb{E} \max_{l=1}^{d^k} \left| \sum_{j=1}^n \sigma_j (\boldsymbol{v}_l)_j \right| = n \mathcal{R}(V). \tag{8}
$$

Using Lemma 8 [Massart Lemma] we have that:

$$
\mathcal{R}(V) \leq 2 \max_j \|\boldsymbol{Z}_j\|_\infty \sqrt{2 \log(d^k)/n}. \tag{9}
$$

Then, it holds that:

$$\mathcal{R}_Z(\mathcal{F}_{\text{CCP}}^k) = \mathbb{E} \sup_{f \in \mathcal{F}_{\text{CCP}}^k} \frac{1}{n} \sum_{j=1}^n \sigma_j f(\boldsymbol{z}_j)$$

$$\leq \frac{\mu\lambda}{n} \mathbb{E} \left\| \sum_{j=1}^n [\sigma_j *_{i=1}^k (\boldsymbol{z}_j)] \right\|_\infty \qquad \text{Eq. (7)}$$

$$= \frac{\mu\lambda}{n} n\mathcal{R}(V) \qquad \text{Eq. (8)} \qquad\qquad (10)$$

$$\leq 2\mu\lambda \max_j \|\boldsymbol{Z}_j\|_\infty \sqrt{2\log(d^k)/n} \qquad \text{Eq. (9)}$$

$$= 2\mu\lambda \max_j \left\| *_{i=1}^k (\boldsymbol{z}_j) \right\|_\infty \sqrt{2\log(d^k)/n}$$

$$\leq 2\mu\lambda (\max_j \|\boldsymbol{z}_j\|_\infty)^k \sqrt{2\log(d^k)/n}$$

$$\leq 2\mu\lambda \sqrt{2k\log(d)/n} \, .$$

$\square$

## E.2 Proof of Lemma 1

Table 8: Core symbols in the proof of Lemma 1.

| Symbol | Dimensions | Definition |
|:---:|:---:|:---:|
| $\otimes$ | - | Kronecker product. |
| $\bullet$ | - | Face-splitting product. |
| $\boldsymbol{z}$ | $\mathbb{R}^d$ | Input of the polynomial expansion. |
| $f(\boldsymbol{z})$ | $\mathbb{R}$ | Output of the polynomial expansion. |
| $k$ | $\mathbb{N}$ | Degree of polynomial expansion. |
| $m$ | $\mathbb{N}$ | Hidden rank of the expansion. |
| $\boldsymbol{U}_i$ | $\mathbb{R}^{m \times d}$ | Learnable matrices. |
| $\boldsymbol{U}_i^j$ | $\mathbb{R}^d$ | $j^{\text{th}}$ row of $\boldsymbol{U}_i$. |
| $\lambda_i$ | $\mathbb{R}$ | $\|\boldsymbol{U}_i\|_\infty \leq \lambda_i$ for $i = 1, 2, \ldots, k$. |

*Proof.*

$$\left\| \bullet_{i=1}^k (\boldsymbol{U}_i) \right\|_\infty = \max_{j=1}^m \left\| [\bullet_{i=1}^k (\boldsymbol{U}_i)]^j \right\|_1$$

$$= \max_{j=1}^m \left\| \otimes_{i=1}^k [\boldsymbol{U}_i^j] \right\|_1 \qquad \text{Definition of Face-splitting product}$$

$$= \max_{j=1}^m \left[ \prod_{i=1}^k \left\| \boldsymbol{U}_i^j \right\|_1 \right] \qquad \text{Multiplicativity of absolute value}$$

$$\leq \prod_{i=1}^k \left[ \max_{j=1}^m \left\| \boldsymbol{U}_i^j \right\|_1 \right]$$

$$= \prod_{i=1}^k \|\boldsymbol{U}_i\|_\infty \, .$$

$\square$

## E.3 Rademacher Complexity bound under $\ell_2$ norm

**Theorem 6.** *Let $Z = \{\boldsymbol{z}_1, \ldots, \boldsymbol{z}_n\} \subseteq \mathbb{R}^d$ and suppose that $\|\boldsymbol{z}_j\|_\infty \leq 1$ for all $j = 1, \ldots, n$. Let*
$$\mathcal{F}_{CCP}^k := \left\{ f(\boldsymbol{z}) = \langle \boldsymbol{c}, \circ_{i=1}^k \boldsymbol{U}_i \boldsymbol{z} \rangle : \|\boldsymbol{c}\|_2 \leq \mu, \left\| \bullet_{i=1}^k \boldsymbol{U}_i \right\|_2 \leq \lambda \right\} \, .$$

*The Empirical Rademacher Complexity of $CCP_k$ ($k$-degree CCP polynomials) with respect to $\mathbf{Z}$ is bounded as:*

$$\mathcal{R}_Z(\mathcal{F}_{CCP}^k) \leq \frac{\mu\lambda}{\sqrt{n}} \,.$$

To facilitate the proof below, we include the related symbols in Table 9. Below, to avoid cluttering the notation, we consider that the expectation is over $\sigma$ and omit the brackets as well.

Table 9: Core symbols for proof of Theorem 6.

| Symbol | Dimensions | Definition |
|--------|------------|------------|
| $\circ$ | - | Hadamard (element-wise) product. |
| $\bullet$ | - | Face-splitting product. |
| $*$ | - | Column-wise Khatri–Rao product. |
| $\mathbf{z}$ | $\mathbb{R}^d$ | Input of the polynomial expansion. |
| $f(\mathbf{z})$ | $\mathbb{R}$ | Output of the polynomial expansion. |
| $k$ | $\mathbb{N}$ | Degree of polynomial expansion. |
| $m$ | $\mathbb{N}$ | Hidden rank of the expansion. |
| $\mathbf{U}_i$ | $\mathbb{R}^{m \times d}$ | Learnable matrices. |
| $\mathbf{c}$ | $\mathbb{R}^{1 \times m}$ | Learnable matrix. |
| $\mu$ | $\mathbb{R}$ | $\|\mathbf{c}\|_2 \leq \mu$. |
| $\lambda$ | $\mathbb{R}$ | $\left\|\bullet_{i=1}^k (\mathbf{U}_i)\right\|_2 \leq \lambda$. |

*Proof.*

$$\mathcal{R}_Z(\mathcal{F}_{\text{CCP}}^k) = \mathbb{E} \sup_{f \in \mathcal{F}_{\text{CCP}}^k} \frac{1}{n} \sum_{j=1}^n \sigma_j f(\mathbf{z}_j)$$

$$= \mathbb{E} \sup_{f \in \mathcal{F}_{\text{CCP}}^k} \frac{1}{n} \sum_{j=1}^n \left( \sigma_j \left\langle \mathbf{c}, \circ_{i=1}^k (\mathbf{U}_i \mathbf{z}_j) \right\rangle \right)$$

$$= \mathbb{E} \sup_{f \in \mathcal{F}_{\text{CCP}}^k} \frac{1}{n} \left\langle \mathbf{c}, \sum_{j=1}^n [\sigma_j \circ_{i=1}^k (\mathbf{U}_i \mathbf{z}_j)] \right\rangle$$

$$\leq \mathbb{E} \sup_{f \in \mathcal{F}_{\text{CCP}}^k} \frac{1}{n} \|\mathbf{c}\|_2 \left\| \sum_{j=1}^n [\sigma_j \circ_{i=1}^k (\mathbf{U}_i \mathbf{z}_j)] \right\|_2 \qquad \text{Lemma 5 [Hölder's inequality]}$$

$$= \mathbb{E} \sup_{f \in \mathcal{F}_{\text{CCP}}^k} \frac{1}{n} \|\mathbf{c}\|_2 \left\| \sum_{j=1}^n [\sigma_j \bullet_{i=1}^k (\mathbf{U}_i) *_{i=1}^k (\mathbf{z}_j)] \right\|_2 \qquad \text{Lemma 7 [Mixed product property]}$$

$$= \mathbb{E} \sup_{f \in \mathcal{F}_{\text{CCP}}^k} \frac{1}{n} \|\mathbf{c}\|_2 \left\| \bullet_{i=1}^k (\mathbf{U}_i) \sum_{j=1}^n [\sigma_j *_{i=1}^k (\mathbf{z}_j)] \right\|_2$$

$$\leq \mathbb{E} \sup_{f \in \mathcal{F}_{\text{CCP}}^k} \frac{1}{n} \|\mathbf{c}\|_2 \left\| \sum_{j=1}^n [\sigma_j *_{i=1}^k (\mathbf{z}_j)] \right\|_2 \left\| \bullet_{i=1}^k (\mathbf{U}_i) \right\|_2 \,.$$

$$\mathbb{E}\left\|\sum_{j=1}^{n}[\sigma_j *_{i=1}^{k}(\boldsymbol{z}_j)]\right\|_2 = \mathbb{E}\sqrt{\left\|\sum_{j=1}^{n}[\sigma_j *_{i=1}^{k}(\boldsymbol{z}_j)]\right\|_2^2}$$

$$\leq \sqrt{\mathbb{E}\left\|\sum_{j=1}^{n}[\sigma_j *_{i=1}^{k}(\boldsymbol{z}_j)]\right\|_2^2} \qquad \text{Jensen's inequality}$$

$$= \sqrt{\mathbb{E}\sum_{s,j}^{n}[\sigma_s\sigma_j\left\langle *_{i=1}^{k}(\boldsymbol{z}_s), *_{i=1}^{k}(\boldsymbol{z}_j)\right\rangle]} \qquad (11)$$

$$= \sqrt{\sum_{j=1}^{n}[\left\| *_{i=1}^{k}(\boldsymbol{z}_j)\right\|_2^2]}$$

$$= \sqrt{\sum_{j=1}^{n}(\prod_{i=1}^{k}\|\boldsymbol{z}_j\|_2^2)}$$

$$\leq \sqrt{n}.$$

So:

$$\mathcal{R}_Z(\mathcal{F}_{\text{CCP}}^k) \leq \mathbb{E}\sup_{f\in\mathcal{F}_{\text{CCP}}^k}\frac{1}{n}\|\boldsymbol{c}\|_2\left\|\sum_{j=1}^{n}[\sigma_j *_{i=1}^{k}(\boldsymbol{z}_j)]\right\|_2\left\|\bullet_{i=1}^{k}(\boldsymbol{U}_i)\right\|_2$$

$$\leq \frac{\sup_{f\in\mathcal{F}_{\text{CCP}}^k}\|\boldsymbol{c}\|_2\left\|\bullet_{i=1}^{k}(\boldsymbol{U}_i)\right\|_2}{\sqrt{n}}$$

$$\leq \frac{\mu\lambda}{\sqrt{n}}.$$

$\square$

### E.4 LIPSCHITZ CONSTANT BOUND OF THE CCP MODEL

We will first prove a more general result about the $\ell_p$-Lipschitz constant of the CCP model.

**Theorem 7.** *The Lipschitz constant (with respect to the $\ell_p$-norm) of the function defined in Eq. (CCP), restricted to the set $\{\boldsymbol{z}\in\mathbb{R}^d : \|\boldsymbol{z}\|_p \leq 1\}$ is bounded as:*

$$Lip_p(f) \leq k\|\boldsymbol{C}\|_p\prod_{i=1}^{k}\|\boldsymbol{U}_i\|_p.$$

*Proof.* Let $g(\boldsymbol{x}) = \boldsymbol{C}\boldsymbol{x}$ and $h(\boldsymbol{z}) = \circ_{i=1}^{k}(\boldsymbol{U}_i\boldsymbol{z})$. Then it holds that $f(\boldsymbol{z}) = g(h(\boldsymbol{z}))$. By Lemma 3, we have: $\text{Lip}_p(f) \leq \text{Lip}_p(g)\text{Lip}_p(h)$. We will compute an upper bound of each function individually.

Let us first consider the function $g(\boldsymbol{x}) = \boldsymbol{C}\boldsymbol{x}$. By Lemma 4, because g is a linear map represented by a matrix $\boldsymbol{C}$, its Jacobian is $J_g(\boldsymbol{x}) = \boldsymbol{C}$. So:

$$\text{Lip}_p(g) = \|\boldsymbol{C}\|_p := \sup_{\|\boldsymbol{x}\|_p=1}\|\boldsymbol{C}\boldsymbol{x}\|_p.$$

where $\|\boldsymbol{C}\|_p$ is the operator norm on matrices induced by the vector $p$-norm.

Now, let us consider the function $h(\boldsymbol{z}) = \circ_{i=1}^{k}\boldsymbol{U}_i\boldsymbol{z}$. Its Jacobian is given by:

$$\frac{dh}{d\boldsymbol{z}} = \sum_{i=1}^{k}\text{diag}(\circ_{j\neq i}\boldsymbol{U}_j\boldsymbol{z})\boldsymbol{U}_i.$$

Using Lemma 4 we have:

$$
\begin{aligned}
\mathrm{Lip}_p(h) &\leq \sup_{\boldsymbol{z}:\|\boldsymbol{z}\|_p \leq 1} \left\| \sum_{i=1}^{k} [\mathrm{diag}(\circ_{j\neq i}(\boldsymbol{U}_j \boldsymbol{z})) \boldsymbol{U}_i] \right\|_p \\
&\leq \sup_{\boldsymbol{z}:\|\boldsymbol{z}\|_p \leq 1} \sum_{i=1}^{k} \left\| \mathrm{diag}(\circ_{j\neq i}(\boldsymbol{U}_j \boldsymbol{z})) \boldsymbol{U}_i \right\|_p && \text{Triangle inequality} \\
&\leq \sup_{\boldsymbol{z}:\|\boldsymbol{z}\|_p \leq 1} \sum_{i=1}^{k} \left\| \mathrm{diag}(\circ_{j\neq i}(\boldsymbol{U}_j \boldsymbol{z})) \right\|_p \|\boldsymbol{U}_i\|_p && \text{Lemma 9 [consistency]} \\
&\leq \sup_{\boldsymbol{z}:\|\boldsymbol{z}\|_p \leq 1} \sum_{i=1}^{k} \left\| \circ_{j\neq i}(\boldsymbol{U}_j \boldsymbol{z}) \right\|_p \|\boldsymbol{U}_i\|_p \\
&\leq \sup_{\boldsymbol{z}:\|\boldsymbol{z}\|_p \leq 1} \sum_{i=1}^{k} \prod_{j\neq i} (\|\boldsymbol{U}_j \boldsymbol{z}\|_p) \|\boldsymbol{U}_i\|_p \\
&\leq \sup_{\boldsymbol{z}:\|\boldsymbol{z}\|_p \leq 1} \sum_{i=1}^{k} \prod_{j\neq i} (\|\boldsymbol{U}_j\|_p \|\boldsymbol{z}\|_p) \|\boldsymbol{U}_i\|_p \\
&\leq \sup_{\boldsymbol{z}:\|\boldsymbol{z}\|_p \leq 1} \sum_{i=1}^{k} \prod_{j=1}^{k} (\|\boldsymbol{U}_j\|_p) \\
&= k \prod_{j=1}^{k} \|\boldsymbol{U}_j\|_p \ .
\end{aligned}
$$

So:

$$
\begin{aligned}
Lip_p(\mathcal{F}_L) &\leq Lip_p(g) Lip_p(h) \\
&\leq k \|\boldsymbol{C}\|_p \prod_{i=1}^{k} \|\boldsymbol{U}_i\|_p \ .
\end{aligned}
$$

$\square$

### E.4.1 PROOF OF THEOREM 2

*Proof.* This is a particular case of Theorem 7 when $p = \infty$. $\square$

# F  RESULT OF THE NCP MODEL

## F.1  PROOF OF THEOREM 3: RADEMACHER COMPLEXITY OF NCP UNDER $\ell_\infty$ NORM

*Proof.*

$$
\begin{aligned}
\mathcal{R}_Z(\mathcal{F}^k_{\text{NCP}}) &= \mathbb{E} \sup_{f \in \mathcal{F}^k_{\text{NCP}}} \frac{1}{n} \sum_{j=1}^n \sigma_j f(\boldsymbol{z}_j) \\
&= \mathbb{E} \sup_{f \in \mathcal{F}^k_{\text{NCP}}} \frac{1}{n} \sum_{j=1}^n (\sigma_j \langle \boldsymbol{c}, \boldsymbol{x}_k(\boldsymbol{z}_j) \rangle) \\
&= \mathbb{E} \sup_{f \in \mathcal{F}^k_{\text{NCP}}} \frac{1}{n} \left\langle \boldsymbol{c}, \sum_{j=1}^n [\sigma_j \boldsymbol{x}_k(\boldsymbol{z}_j)] \right\rangle \\
&\le \mathbb{E} \sup_{f \in \mathcal{F}^k_{\text{NCP}}} \frac{1}{n} \|\boldsymbol{c}\|_1 \left\| \sum_{j=1}^n [\sigma_j \boldsymbol{x}_k(\boldsymbol{z}_j)] \right\|_\infty \qquad\qquad \text{Lemma 5 [Hölder's inequality]} \\
&= \mathbb{E} \sup_{f \in \mathcal{F}^k_{\text{NCP}}} \frac{1}{n} \|\boldsymbol{c}\|_1 \left\| \sum_{j=1}^n [\sigma_j ((\boldsymbol{A}_k \boldsymbol{z}_j) \circ (\boldsymbol{S}_k \boldsymbol{x}_{k-1}(\boldsymbol{z}_j)))] \right\|_\infty \\
&= \mathbb{E} \sup_{f \in \mathcal{F}^k_{\text{NCP}}} \frac{1}{n} \|\boldsymbol{c}\|_1 \left\| \sum_{j=1}^n [\sigma_j ((\boldsymbol{A}_k \bullet \boldsymbol{S}_k)(\boldsymbol{z}_j * \boldsymbol{x}_{k-1}(\boldsymbol{z}_j)))] \right\|_\infty \qquad \text{Lemma 7 [Mixed product property]} \\
&= \mathbb{E} \sup_{f \in \mathcal{F}^k_{\text{NCP}}} \frac{1}{n} \|\boldsymbol{c}\|_1 \left\| (\boldsymbol{A}_k \bullet \boldsymbol{S}_k) \sum_{j=1}^n [\sigma_j (\boldsymbol{z}_j * \boldsymbol{x}_{k-1}(\boldsymbol{z}_j))] \right\|_\infty.
\end{aligned}
\tag{12}
$$

Now, because of the recursive definition of the Eq. (NCP), we obtain:

$$
\begin{aligned}
\sum_{j=1}^n \sigma_j (\boldsymbol{z}_j * \boldsymbol{x}_{k-1}(\boldsymbol{z}_j)) &= \sum_{j=1}^n \sigma_j (\boldsymbol{z}_j * (\boldsymbol{A}_{k-1} \boldsymbol{z}_j) \circ (\boldsymbol{S}_{k-1} \boldsymbol{x}_{k-2}(\boldsymbol{z}_j))) \\
&= \sum_{j=1}^n \sigma_j (\boldsymbol{z}_j * ((\boldsymbol{A}_{k-1} \bullet \boldsymbol{S}_{k-1})(\boldsymbol{z}_j * \boldsymbol{x}_{k-2}(\boldsymbol{z}_j)))) \qquad \text{Lemma 7} \\
&= \sum_{j=1}^n \sigma_j (\boldsymbol{I} \otimes (\boldsymbol{A}_{k-1} \bullet \boldsymbol{S}_{k-1}))(\boldsymbol{z}_j * (\boldsymbol{z}_j * \boldsymbol{x}_{k-2}(\boldsymbol{z}_j))) \qquad \text{Lemma 10} \\
&= \boldsymbol{I} \otimes (\boldsymbol{A}_{k-1} \bullet \boldsymbol{S}_{k-1}) \sum_{j=1}^n [\sigma_j (\boldsymbol{z}_j * (\boldsymbol{z}_j * \boldsymbol{x}_{k-2}(\boldsymbol{z}_j)))].
\end{aligned}
\tag{13}
$$

recursively applying this argument we have:

$$
\sum_{j=1}^n \sigma_j (\boldsymbol{z}_j * \boldsymbol{x}_{k-1}(\boldsymbol{z}_j)) = \left( \prod_{i=1}^{k-1} \boldsymbol{I} \otimes \boldsymbol{A}_i \bullet \boldsymbol{S}_i \right) \sum_{j=1}^n \sigma_j *_{i=1}^k (\boldsymbol{z}_j).
\tag{14}
$$

Combining the two previous equations (Eqs. (13) and (14)) inside Eq. (12) we finally obtain

$$\mathcal{R}_Z(\mathcal{F}_{\text{NCP}}^k) \leq \mathbb{E} \sup_{f \in \mathcal{F}_{\text{NCP}}^k} \frac{1}{n} \|\boldsymbol{c}\|_1 \left\| (\boldsymbol{A}_k \bullet \boldsymbol{S}_k) \left( \prod_{i=1}^{k-1} \boldsymbol{I} \otimes \boldsymbol{A}_i \bullet \boldsymbol{S}_i \right) \sum_{j=1}^{n} \sigma_j *_{i=1}^{k} (\boldsymbol{z}_j) \right\|_\infty$$

$$\leq \mathbb{E} \sup_{f \in \mathcal{F}_{\text{NCP}}^k} \frac{1}{n} \|\boldsymbol{c}\|_1 \left\| (\boldsymbol{A}_k \bullet \boldsymbol{S}_k) \left( \prod_{i=1}^{k-1} \boldsymbol{I} \otimes \boldsymbol{A}_i \bullet \boldsymbol{S}_i \right) \right\|_\infty \left\| \sum_{j=1}^{n} \sigma_j *_{i=1}^{k} (\boldsymbol{z}_j) \right\|_\infty$$

$$\leq \frac{\mu\lambda}{n} \mathbb{E} \left\| \sum_{j=1}^{n} \sigma_j *_{i=1}^{k} (\boldsymbol{z}_j) \right\|_\infty$$

$$= \frac{\mu\lambda}{n} n \mathcal{R}(V) = \mu\lambda\mathcal{R}(V). \qquad \text{Eq. (8)}.$$

following the same arguments as in Eq. (10) it follows that:

$$\mathcal{R}_Z(\mathcal{F}_{\text{NCP}}^k) \leq 2\mu\lambda \sqrt{\frac{2k \log(d)}{n}}.$$

$\square$

## F.2 Proof of Lemma 2

*Proof.*

$$\left\| (\boldsymbol{A}_k \bullet \boldsymbol{S}_k) \prod_{i=1}^{k-1} \boldsymbol{I} \otimes \boldsymbol{A}_i \bullet \boldsymbol{S}_i \right\|_\infty \leq \|\boldsymbol{A}_k \bullet \boldsymbol{S}_k\|_\infty \prod_{i=1}^{k-1} \|\boldsymbol{I} \otimes \boldsymbol{A}_i \bullet \boldsymbol{S}_i\|_\infty \qquad \text{Lemma 9 [consistent]}$$

$$= \prod_{i=1}^{k} \|\boldsymbol{A}_i \bullet \boldsymbol{S}_i\|_\infty$$

$$\leq \prod_{i=1}^{k} \|\boldsymbol{A}_i\|_\infty \|\boldsymbol{S}_i\|_\infty \qquad \text{Lemma 1}.$$

$\square$

## F.3 Rademacher Complexity under $\ell_2$ norm

**Theorem 8.** *Let $Z = \{\boldsymbol{z}_1, \ldots, \boldsymbol{z}_n\} \subseteq \mathbb{R}^d$ and suppose that $\|\boldsymbol{z}_j\|_\infty \leq 1$ for all $j = 1, \ldots, n$. Define the matrix $\Phi(\boldsymbol{A}_1, \boldsymbol{S}_1, \ldots, \boldsymbol{A}_n, \boldsymbol{S}_n) \coloneqq (\boldsymbol{A}_k \bullet \boldsymbol{S}_k) \prod_{i=1}^{k-1} \boldsymbol{I} \otimes \boldsymbol{A}_i \bullet \boldsymbol{S}_i$. Consider the class of functions:*

$$\mathcal{F}_{NCP}^k \coloneqq \{f(\boldsymbol{z}) \text{ as in (NCP)} : \|\boldsymbol{C}\|_2 \leq \mu, \|\Phi(\boldsymbol{A}_1, \boldsymbol{S}_1, \ldots, \boldsymbol{A}_k, \boldsymbol{S}_k)\|_2 \leq \lambda\},$$

*where $\boldsymbol{C} \in \mathbb{R}^{1 \times m}$ (single output case). The Empirical Rademacher Complexity of $NCP_k$ (k-degree NCP polynomials) with respect to Z is bounded as:*

$$\mathcal{R}_Z(\mathcal{F}_{NCP}^k) \leq \frac{\mu\lambda}{\sqrt{n}}.$$

*Proof.*

$$\mathcal{R}_Z(\mathcal{F}_{\text{NCP}}^k) = \mathbb{E} \sup_{f \in \mathcal{F}_{\text{NCP}}^k} \frac{1}{n} \sum_{j=1}^{n} \sigma_j f(\boldsymbol{z}_j)$$

$$= \mathbb{E} \sup_{f \in \mathcal{F}_{\text{NCP}}^k} \frac{1}{n} \sum_{j=1}^{n} (\sigma_j \langle \boldsymbol{c}, \boldsymbol{x}_k(\boldsymbol{z}_j) \rangle)$$

$$= \mathbb{E} \sup_{f \in \mathcal{F}_{\text{NCP}}^k} \frac{1}{n} \left\langle \boldsymbol{c}, \sum_{j=1}^{n} [\sigma_j \boldsymbol{x}_k(\boldsymbol{z}_j)] \right\rangle$$

$$\leq \mathbb{E} \sup_{f \in \mathcal{F}_{\text{NCP}}^k} \frac{1}{n} \|\boldsymbol{c}\|_2 \left\| \sum_{j=1}^{n} [\sigma_j \boldsymbol{x}_k(\boldsymbol{z}_j)] \right\|_2 \qquad \text{Lemma 5 [Hölder's inequality]}$$

$$= \mathbb{E} \sup_{f \in \mathcal{F}_{\text{NCP}}^k} \frac{1}{n} \|\boldsymbol{c}\|_2 \left\| \sum_{j=1}^{n} [\sigma_j((\boldsymbol{A}_k \boldsymbol{z}_j) \circ (\boldsymbol{S}_k \boldsymbol{x}_{k-1}(\boldsymbol{z}_j)))] \right\|_2$$

$$= \mathbb{E} \sup_{f \in \mathcal{F}_{\text{NCP}}^k} \frac{1}{n} \|\boldsymbol{c}\|_2 \left\| \sum_{j=1}^{n} [\sigma_j((\boldsymbol{A}_k \bullet \boldsymbol{S}_k)(\boldsymbol{z}_j * \boldsymbol{x}_{k-1}(\boldsymbol{z}_j)))] \right\|_2 \qquad \text{Lemma 7 [Mixed product property]}$$

$$= \mathbb{E} \sup_{f \in \mathcal{F}_{\text{NCP}}^k} \frac{1}{n} \|\boldsymbol{c}\|_2 \left\| (\boldsymbol{A}_k \bullet \boldsymbol{S}_k) \sum_{j=1}^{n} [\sigma_j(\boldsymbol{z}_j * \boldsymbol{x}_{k-1}(\boldsymbol{z}_j))] \right\|_2$$

$$= \mathbb{E} \sup_{f \in \mathcal{F}_{\text{NCP}}^k} \frac{1}{n} \|\boldsymbol{c}\|_2 \left\| (\boldsymbol{A}_k \bullet \boldsymbol{S}_k) \left( \prod_{i=1}^{k-1} \boldsymbol{I} \otimes \boldsymbol{A}_i \bullet \boldsymbol{S}_i \right) \sum_{j=1}^{n} \sigma_j *_{i=1}^{k} (\boldsymbol{z}_j)] \right\|_2 \qquad \text{Eq. (14)}$$

$$\leq \mathbb{E} \sup_{f \in \mathcal{F}_{\text{NCP}}^k} \frac{1}{n} \|\boldsymbol{c}\|_2 \left\| (\boldsymbol{A}_k \bullet \boldsymbol{S}_k) \left( \prod_{i=1}^{k-1} \boldsymbol{I} \otimes \boldsymbol{A}_i \bullet \boldsymbol{S}_i \right) \right\|_2 \left\| \sum_{j=1}^{n} \sigma_j *_{i=1}^{k} (\boldsymbol{z}_j)] \right\|_2$$

$$\leq \frac{\mu\lambda}{n} \mathbb{E} \left\| \sum_{j=1}^{n} \sigma_j *_{i=1}^{k} (\boldsymbol{z}_j) \right\|_2$$

$$\leq \frac{\mu\lambda}{\sqrt{n}} . \qquad \text{Eq. (11)}.$$

$\square$

## F.4 LIPSCHITZ CONSTANT BOUND OF THE NCP MODEL

**Theorem 9.** *Let $\mathcal{F}_L$ be the class of functions defined as*

$$\mathcal{F}_L := \left\{ \boldsymbol{x}_1 = (\boldsymbol{A}_1 \boldsymbol{z}) \circ (\boldsymbol{S}_1), \boldsymbol{x}_n = (\boldsymbol{A}_n \boldsymbol{z}) \circ (\boldsymbol{S}_n \boldsymbol{x}_{n-1}), f(\boldsymbol{z}) = \boldsymbol{C} \boldsymbol{x}_k : \right.$$

$$\left. \|\boldsymbol{C}\|_p \leq \mu, \|\boldsymbol{A}_i\|_p \leq \lambda_i, \|\boldsymbol{S}_i\|_p \leq \rho_i, \|\boldsymbol{z}\|_p \leq 1 \right\}.$$

*The Lipschitz Constant of $\mathcal{F}_\mathcal{L}$ (k-degree NCP polynomial) under $\ell_p$ norm restrictions is bounded as:*

$$Lip_p(\mathcal{F}_L) \leq k\mu \prod_{i=1}^{k} (\lambda_i \rho_i) .$$

*Proof.* Let $g(\boldsymbol{x}) = \boldsymbol{C}\boldsymbol{x}$, $h(\boldsymbol{z}) = (\boldsymbol{A}_n \boldsymbol{z}) \circ (\boldsymbol{S}_n \boldsymbol{x}_{n-1}(\boldsymbol{z}))$. Then it holds that $f(\boldsymbol{z}) = g(h(\boldsymbol{z}))$.

By Lemma 3, we have: $\mathrm{Lip}(f) \leq \mathrm{Lip}(g)\mathrm{Lip}(h)$. This enables us to compute an upper bound of each function (i.e., $g, h$) individually.

Let us first consider the function $g(\boldsymbol{x}) = \boldsymbol{C}\boldsymbol{x}$. By Lemma 4, because g is a linear map represented by a matrix $\boldsymbol{C}$, its Jacobian is $J_g(\boldsymbol{x}) = \boldsymbol{C}$. So:

$$\mathrm{Lip}_p(g) = \|\boldsymbol{C}\|_p := \sup_{\|\boldsymbol{x}\|_p = 1} \|\boldsymbol{C}\boldsymbol{x}\|_p = \begin{cases} \sigma_{\max}(\boldsymbol{C}) & \text{if } p = 2 \\ \max_i \sum_j |\boldsymbol{C}_{(i,j)}| & \text{if } p = \infty. \end{cases}$$

where $\|\boldsymbol{C}\|_p$ is the operator norm on matrices induced by the vector p-norm, and $\sigma_{\max}(\boldsymbol{C})$ is the largest singular value of $\boldsymbol{C}$.

Now, let us consider the function $\boldsymbol{x}_n(\boldsymbol{z}) = h(\boldsymbol{z}) = (\boldsymbol{A}_n\boldsymbol{z}) \circ (\boldsymbol{S}_n\boldsymbol{x}_{n-1}(\boldsymbol{z}))$. Its Jacobian is given by:

$$J_{\boldsymbol{x}_n} = \mathrm{diag}(\boldsymbol{A}_n\boldsymbol{z})\boldsymbol{S}_n J_{\boldsymbol{x}_{n-1}} + \mathrm{diag}(\boldsymbol{S}_n\boldsymbol{x}_{n-1})\boldsymbol{A}_n, \qquad J_{\boldsymbol{x}_1} = \mathrm{diag}(\boldsymbol{S}_1)\boldsymbol{A}_1.$$

$$
\begin{aligned}
Lip_p(h) &= \sup_{\boldsymbol{z}:\|\boldsymbol{z}\|_p \leq 1} \|J_{\boldsymbol{x}_n}\|_p \\
&= \sup_{\boldsymbol{z}:\|\boldsymbol{z}\|_p \leq 1} \|\mathrm{diag}(\boldsymbol{A}_n\boldsymbol{z})\boldsymbol{S}_n J_{\boldsymbol{x}_{n-1}} + \mathrm{diag}(\boldsymbol{S}_n\boldsymbol{x}_{n-1})\boldsymbol{A}_n\|_p \\
&\leq \sup_{\boldsymbol{z}:\|\boldsymbol{z}\|_p \leq 1} \|\mathrm{diag}(\boldsymbol{A}_n\boldsymbol{z})\boldsymbol{S}_n J_{\boldsymbol{x}_{n-1}}\|_p + \|\mathrm{diag}(\boldsymbol{S}_n\boldsymbol{x}_{n-1})\boldsymbol{A}_n\|_p \qquad \text{Triangle inequality} \\
&\leq \sup_{\boldsymbol{z}:\|\boldsymbol{z}\|_p \leq 1} \|\mathrm{diag}(\boldsymbol{A}_n\boldsymbol{z})\|_p\|\boldsymbol{S}_n\|_p\|J_{\boldsymbol{x}_{n-1}}\|_p + \|\mathrm{diag}(\boldsymbol{S}_n\boldsymbol{x}_{n-1})\|_p\|\boldsymbol{A}_n\|_p \qquad \text{Lemma 9 [consistent]} \\
&\leq \sup_{\boldsymbol{z}:\|\boldsymbol{z}\|_p \leq 1} \|\boldsymbol{A}_n\boldsymbol{z}\|_p\|\boldsymbol{S}_n\|_p\|J_{\boldsymbol{x}_{n-1}}\|_p + \|\boldsymbol{S}_n\boldsymbol{x}_{n-1}\|_p\|\boldsymbol{A}_n\|_p \\
&\leq \sup_{\boldsymbol{z}:\|\boldsymbol{z}\|_p \leq 1} \|\boldsymbol{A}_n\|_p\|\boldsymbol{z}\|_p\|\boldsymbol{S}_n\|_p\|J_{\boldsymbol{x}_{n-1}}\|_p + \|\boldsymbol{S}_n\|_p\|\boldsymbol{x}_{n-1}\|_p\|\boldsymbol{A}_n\|_p \\
&= \sup_{\boldsymbol{z}:\|\boldsymbol{z}\|_p \leq 1} \|\boldsymbol{A}_n\|_p\|\boldsymbol{z}\|_p\|\boldsymbol{S}_n\|_p\|J_{\boldsymbol{x}_{n-1}}\|_p + \|\boldsymbol{S}_n\|_p\|(\boldsymbol{A}_{n-1}\boldsymbol{z}) \circ (\boldsymbol{S}_{n-1}\boldsymbol{x}_{n-2})\|_p\|\boldsymbol{A}_n\|_p \\
&\leq \sup_{\boldsymbol{z}:\|\boldsymbol{z}\|_p \leq 1} \|\boldsymbol{A}_n\|_p\|\boldsymbol{z}\|_p\|\boldsymbol{S}_n\|_p\|J_{\boldsymbol{x}_{n-1}}\|_p + \|\boldsymbol{S}_n\|_p\|\boldsymbol{A}_{n-1}\boldsymbol{z}\|_p\|\boldsymbol{S}_{n-1}\boldsymbol{x}_{n-2}\|_p\|\boldsymbol{A}_n\|_p \\
&\leq \sup_{\boldsymbol{z}:\|\boldsymbol{z}\|_p \leq 1} \|\boldsymbol{A}_n\|_p\|\boldsymbol{z}\|_p\|\boldsymbol{S}_n\|_p\|J_{\boldsymbol{x}_{n-1}}\|_p + \|\boldsymbol{S}_n\|_p\|\boldsymbol{A}_{n-1}\|_p\|\boldsymbol{z}\|_p\|\boldsymbol{S}_{n-1}\|_p\|\boldsymbol{x}_{n-2}\|_p\|\boldsymbol{A}_n\|_p \\
&= \sup_{\boldsymbol{z}:\|\boldsymbol{z}\|_p \leq 1} \|\boldsymbol{A}_n\|_p\|\boldsymbol{z}\|_p\|\boldsymbol{S}_n\|_p(\|J_{\boldsymbol{x}_{n-1}}\|_p + \|\boldsymbol{A}_{n-1}\|_p\|\boldsymbol{S}_{n-1}\|_p\|\boldsymbol{x}_{n-2}\|_p) \\
&\leq \sup_{\boldsymbol{z}:\|\boldsymbol{z}\|_p \leq 1} \|\boldsymbol{A}_n\|_p\|\boldsymbol{z}\|_p\|\boldsymbol{S}_n\|_p(\|J_{\boldsymbol{x}_{n-1}}\|_p + \prod_{i=1}^{n-1}(\|\boldsymbol{S}_i\|_p\|\boldsymbol{A}_i\|_p)\|\boldsymbol{z}\|_p^{n-2}).
\end{aligned}
$$

Then we proof the result by induction.

Inductive hypothesis:

$$\sup_{\boldsymbol{z}:\|\boldsymbol{z}\|_p \leq 1} \|J_{\boldsymbol{x}_n}\|_p \leq n \prod_{i=1}^{n}(\|\boldsymbol{S}_i\|_p\|\boldsymbol{A}_i\|_p).$$

Case $k = 1$:

$$
\begin{aligned}
Lip_p(h) &= \sup_{\boldsymbol{z}:\|\boldsymbol{z}\|_p \leq 1} \|J_{\boldsymbol{x}_1}\|_p \\
&= \|\mathrm{diag}(\boldsymbol{S}_1)\boldsymbol{A}_1\|_p \\
&\leq \|\mathrm{diag}(\boldsymbol{S}_1)\|_p\|\boldsymbol{A}_1\|_p \\
&\leq \|\boldsymbol{S}_1\|_p\|\boldsymbol{A}_1\|_p.
\end{aligned}
$$

Case $k = n$:

$$Lip_p(h) = \sup_{\boldsymbol{z}:\|\boldsymbol{z}\|_p \le 1} \|J_{\boldsymbol{x}_n}\|_p$$

$$\le \sup_{\boldsymbol{z}:\|\boldsymbol{z}\|_p \le 1} \|\boldsymbol{A}_n\|_p \|\boldsymbol{z}\|_p \|\boldsymbol{S}_n\|_p (\|J_{\boldsymbol{x}_{n-1}}\|_p + \prod_{i=1}^{n-1}(\|\boldsymbol{S}_i\|_p \|\boldsymbol{A}_i\|_p)\|\boldsymbol{z}\|_p^{n-2})$$

$$\le \sup_{\boldsymbol{z}:\|\boldsymbol{z}\|_p \le 1} \|\boldsymbol{A}_n\|_p \|\boldsymbol{z}\|_p \|\boldsymbol{S}_n\|_p ((n-1)\prod_{i=1}^{n-1}(\|\boldsymbol{S}_i\|_p \|\boldsymbol{A}_i\|_p) + \prod_{i=1}^{n-1}(\|\boldsymbol{S}_i\|_p \|\boldsymbol{A}_i\|_p)\|\boldsymbol{z}\|_p^{n-2})$$

$$\le n\prod_{i=1}^{n}(\|\boldsymbol{S}_i\|_p \|\boldsymbol{A}_i\|_p).$$

So:

$$Lip_p(\mathcal{F}_L) \le Lip_p(g)Lip_p(h)$$

$$\le k\|\boldsymbol{C}\|_p \prod_{i=1}^{k}(\|\boldsymbol{S}_i\|_p \|\boldsymbol{A}_i\|_p).$$

□

### F.4.1 PROOF OF THEOREM 4

*Proof.* This is particular case of Theorem 9 with $p = \infty$. □

## G RELATIONSHIP BETWEEN A CONVOLUTIONAL LAYER AND A FULLY CONNECTED LAYER

In this section we discuss various cases of input/output types depending on the dimensionality of the input tensor and the output tensor. We also provide the proof of Theorem 5.

**Theorem 10.** *Let $\boldsymbol{A} \in \mathbb{R}^n$. Let $\boldsymbol{K} \in \mathbb{R}^h$ be a 1-D convolutional kernel. For simplicity, we assume $h$ is odd and $h \le n$. Let $\boldsymbol{B} \in \mathbb{R}^n$, $\boldsymbol{B} = \boldsymbol{K} \star \boldsymbol{A}$ be the output of the convolution. Let $\boldsymbol{U}$ be the convolutional operator i.e., the linear operator (matrix) $\boldsymbol{U} \in \mathbb{R}^{n\times n}$ such that $\boldsymbol{B} = \boldsymbol{K} \star \boldsymbol{A} = \boldsymbol{U}\boldsymbol{A}$. It holds that $\|\boldsymbol{U}\|_\infty = \|\boldsymbol{K}\|_1$.*

**Theorem 11.** *Let $\boldsymbol{A} \in \mathbb{R}^{n\times m}$, and let $\boldsymbol{K} \in \mathbb{R}^{h\times h}$ be a 2-D convolutional kernel. For simplicity assume $h$ is odd number and $h \le \min(n, m)$. Let $\boldsymbol{B} \in \mathbb{R}^{n\times m}$, $\boldsymbol{B} = \boldsymbol{K} \star \boldsymbol{A}$ be the output of the convolution. Let $\boldsymbol{U}$ be the convolutional operator i.e., the linear operator (matrix) $\boldsymbol{U} \in \mathbb{R}^{nm\times nm}$ such that $vec(\boldsymbol{B}) = \boldsymbol{U}vec(\boldsymbol{A})$. It holds that $\|\boldsymbol{U}\|_\infty = \|vec(\boldsymbol{K})\|_1$.*

### G.1 PROOF OF THEOREM 10

*Proof.* From, $\boldsymbol{B} = \boldsymbol{K} \star \boldsymbol{A} = \boldsymbol{U}\boldsymbol{A}$ we can obtain the following:

$$\begin{pmatrix} u_{1,1} & u_{1,2} & \cdots & u_{1,n} \\ u_{2,1} & u_{2,2} & \cdots & u_{2,n} \\ \vdots & \vdots & \ddots & \vdots \\ u_{n,1} & u_{n,2} & \cdots & u_{n,n} \end{pmatrix} (\boldsymbol{A}^1, \cdots, \boldsymbol{A}^n)^\top = (\boldsymbol{K}^1, \cdots, \boldsymbol{K}^h)^\top \star (\boldsymbol{A}^1, \cdots, \boldsymbol{A}^n)^\top.$$

We observe that:

$$u_{i,j} = \begin{cases} \boldsymbol{K}^{\frac{h+1}{2}+j-i} & \text{if } |i-j| \le \frac{h-1}{2} \, ; \\ 0 & \text{if } |i-j| > \frac{h-1}{2} \, . \end{cases}$$

Then, it holds that:

$$\|\boldsymbol{U}\|_\infty = \max_{i=1}^n \sum_{j=1}^n |u_{i,j}| \le \max_{i=1}^n \sum_{j=1}^h |\boldsymbol{K}^j| = \|\boldsymbol{K}\|_1 \; .$$

$\square$

### G.2 PROOF OF THEOREM 11

*Proof.* We partition $\boldsymbol{U}$ into $n \times n$ partition matrices of shape $m \times m$. Then the $(i,j)^{\text{th}}$ partition matrix $\boldsymbol{U}_{(i,j)}$ describes the relationship between the $\boldsymbol{B}^i$ and the $\boldsymbol{A}^j$. So, $\boldsymbol{U}_{(i,j)}$ is also similar to the Toeplitz matrix in the previous result.

$$\boldsymbol{U}_{(i,j)} = \begin{cases} \boldsymbol{M}_{\frac{h+1}{2}+j-i} & \text{if } |i-j| \le \frac{h-1}{2} \; ; \\ \boldsymbol{0} & \text{if } |i-j| > \frac{h-1}{2} \; . \end{cases}$$

Meanwhile, the matrix $M$ satisfies:

$$m_{i(s,l)} = \begin{cases} k_{(i,\frac{h+1}{2}+l-s)} & \text{if } |s-l| \le \frac{h-1}{2} \; ; \\ 0 & \text{if } |s-l| > \frac{h-1}{2} \; . \end{cases}$$

Then, we have the following:

$$\|\boldsymbol{U}\|_\infty = \max_{i=1}^{n\times m} \sum_{j=1}^{n\times m} |u_{i,j}| \le \max_{i=1}^n \sum_{j=1}^n \|\boldsymbol{U}_{i,j}\|_\infty \le \max_{i=1}^n \sum_{j=1}^h \|\boldsymbol{K}^j\|_1 = \sum_{i=1}^h \|\boldsymbol{K}^i\|_1 = \|\text{vec}(\boldsymbol{K})\|_1 \, .$$

In addition, by $h \le n$, $h \le m$ we have:

$$\left\|\boldsymbol{U}^{\frac{h+1}{2}+\boldsymbol{M}(\frac{h-1}{2})}\right\|_1 = \sum_{i=1}^h \|\boldsymbol{K}^i\|_1 \; . \tag{15}$$

Then, it holds that $\|\boldsymbol{U}\|_\infty = \sum_{i=1}^h \|\boldsymbol{K}^i\|_1$. $\square$

### G.3 PROOF OF THEOREM 5

*Proof.* We partition $\boldsymbol{U}$ into $o \times r$ partition matrices of shape $nm \times nm$. Then the $(i,j)^{\text{th}}$ partition of the matrix $\boldsymbol{U}_{(i,j)}$ describes the relationship between the $i^{\text{th}}$ channel of $\boldsymbol{B}$ and the $j^{\text{th}}$ channel of $\boldsymbol{A}$. Then, the following holds: $\|\boldsymbol{U}_{(i,j)}\|_\infty = \sum_{i=1}^h \|\boldsymbol{K}_{ij}^i\|_1$, where $\boldsymbol{K}_{ij}$ means the two-dimensional tensor obtained by the third dimension of $\boldsymbol{K}$ takes $j$ and the fourth dimension of $\boldsymbol{K}$ takes $i$.

$$\begin{aligned}
\|\boldsymbol{U}\|_\infty &= max_{i=1}^{n\times m\times o} \sum_{j=1}^{n\times m\times r} |u_{(i,j)}| \\
&= \max_{l=0}^{o-1} \max_{i=1}^{n\times m} \sum_{s=0}^{r-1} \sum_{j=1}^{n\times m} |u_{(i+nml,j+nms)}| \\
&\le \max_{l=0}^{o-1} \sum_{s=0}^{r-1} (\max_{i=1}^{n\times m} \sum_{j=1}^{n\times m} |u_{(i+nml,j+nms)}|) \\
&= \max_{l=0}^{o-1} \sum_{s=0}^{r-1} \|\boldsymbol{U}_{(l+1,s+1)}\|_\infty \\
&= \max_{l=1}^o \sum_{s=1}^r \|\boldsymbol{U}_{(l,s)}\|_\infty \\
&= \max_{l=1}^o \sum_{s=1}^r \sum_{i=1}^h \|\boldsymbol{K}_{ls}^i\|_1 \\
&\le \max_{l=1}^o \|\hat{\boldsymbol{K}}^l\|_1 \\
&= \|\hat{\boldsymbol{K}}\|_\infty \; .
\end{aligned}$$

Similar to Eq. (15): for every $nm$ rows, we choose $\frac{k+1}{2}^{\text{th}}$ row. Then its 1-norm is equal to this $nm$ rows of the $\hat{K}$'s $\infty$-norm. So the equation holds. $\qquad\square$

## H  AUXILIARY NUMERICAL EVIDENCE

A number of additional experiments are conducted in this section. Unless explicitly mentioned otherwise, the experimental setup remains similar to the one in the main paper. The following experiments are conducted below:

1. The difference between the theoretical and the algorithmic bound and their evolution during training is studied in Appendix H.1.

2. An ablation study on the hidden size is conducted in Appendix H.2.

3. An ablation study is conducted on the effect of adversarial steps in Appendix H.3.

4. We evaluate the effect of the proposed projection into the testset performance in Appendix H.4.

5. We conduct experiments on four new datasets, i.e., MNIST, K-MNIST, E-MNIST-BY, NSYNTH in Appendix H.5. These experiments are conducted in addition to the datasets already presented in the main paper.

6. In Appendix H.6 experiments on three additional adversarial attacks, i.e., FGSM-0.01, APGDT and TPGD, are performed.

7. We conduct an experiment using the NCP model in Appendix H.7.

8. The layer-wise bound (instead of a single bound for all matrices) is explored in Appendix H.8.

9. The comparison with adversarial defense methods is conducted in Appendix H.9.

### H.1  THEORETICAL AND ALGORITHMIC BOUND

As mentioned in Section 3, projecting the quantity $\theta = \left\|\bullet_{i=1}^{k}\boldsymbol{U}_i\right\|_{\infty}$ onto their level set corresponds to a difficult non-convex problem. Given that we have an upper bound

$$\theta = \left\|\bullet_{i=1}^{k}\boldsymbol{U}_i\right\|_{\infty} \leq \Pi_{i=1}^{k}\|\boldsymbol{U}_i\|_{\infty} =: \gamma\,.$$

we want to understand in practice how tight is this bound. In Fig. 4 we compute the ratio $\frac{\gamma}{\theta}$ for PN-4. In Fig. 5 the ratio is illustrated for randomly initialized matrices (i.e., untrained networks).

### H.2  ABLATION STUDY ON THE HIDDEN SIZE

Initially, we explore the effect of the hidden rank of PN-4 and PN-10 on Fashion-MNIST. Fig. 6 exhibits the accuracy on both the training and the test-set for both models. We observe that PN-10 has a better accuracy on the training set, however the accuracy on the test set is the same in the two models. We also note that increasing the hidden rank improves the accuracy on the training set, but not on the test set.

### H.3  ABLATION STUDY ON THE EFFECT OF ADVERSARIAL STEPS

Our next experiment scrutinizes the effect of the number of adversarial steps on the robust accuracy. We consider in all cases a projection bound of 1, which provides the best empirical results. We vary the number of adversarial steps and report the accuracy in Fig. 7. The results exhibit a similar performance both in terms of the dataset (i.e., Fashion-MNIST and K-MNIST) and in terms of the network (PN-4 and PN-Conv). Notice that when the adversarial attack has more than 10 steps the performance does not vary significantly from the performance at 10 steps, indicating that the projection bound is effective for stronger adversarial attacks.

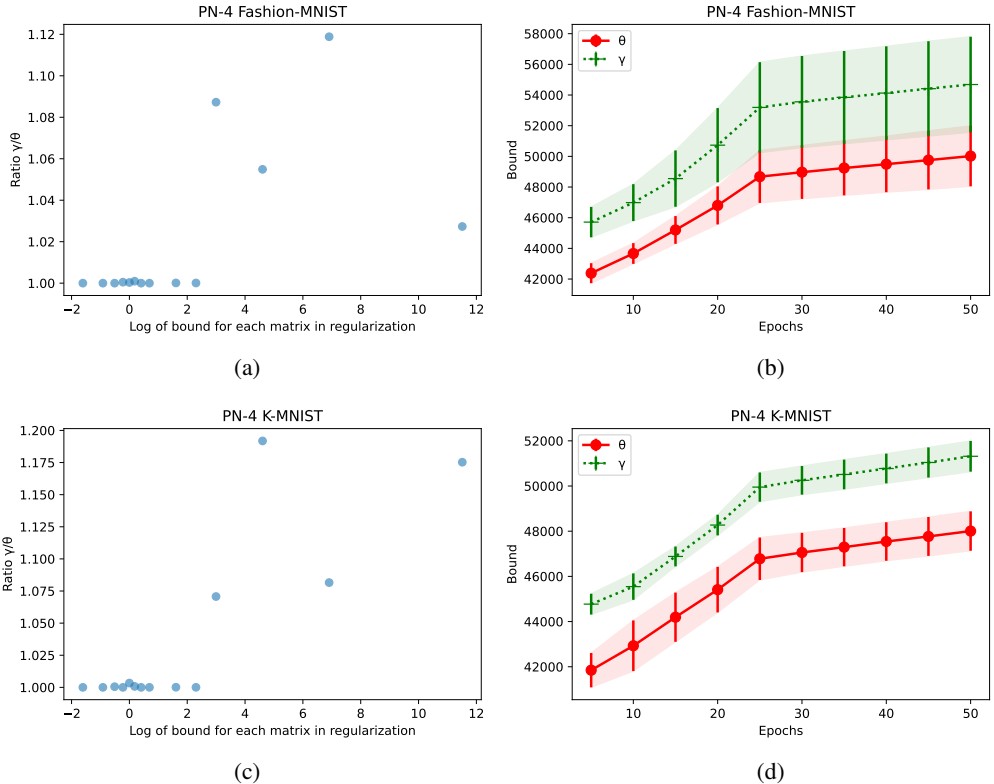

Figure 4: Visualization of the difference between the bound results on Fashion-MNIST (top row) and on K-MNIST (bottom row). Specifically, in (a) and (c) we visualize the ratio $\frac{\gamma}{\theta} = \frac{\prod_{i=1}^{k} \|U_i\|_\infty}{\left\| \bullet_{i=1}^{k} U_i \right\|_\infty}$ for different log bound values for PN-4. In (b), (d) the exact values of the two bounds are computed over the course of the unregularized training. Notice that there is a gap between the two bounds, however importantly the two bounds are increasing at the same rate, while their ratio is close to 1.

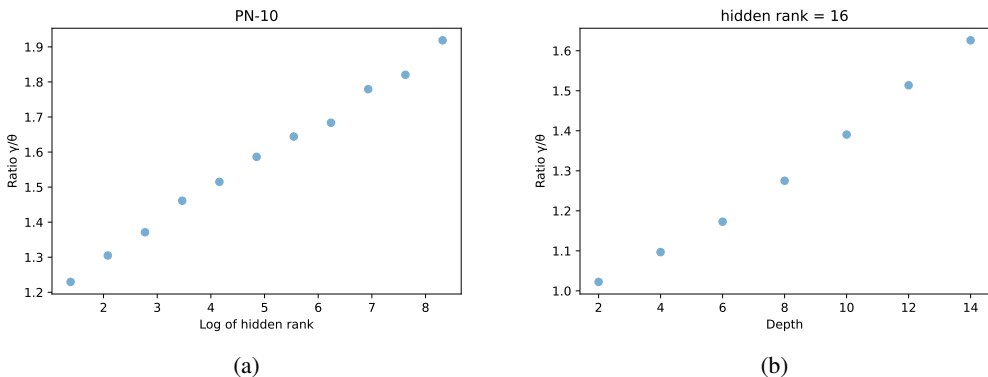

Figure 5: Visualization of the ratio $\frac{\prod_{i=1}^{k} \|U_i\|_\infty}{\left\| \bullet_{i=1}^{k} U_i \right\|_\infty}$ in a randomly initialized network (i.e., using normal distribution random matrices). Specifically, in (a) we visualize the ratio for different log hidden rank values for PN-10. In (b) we visualize the ratio for different depth values for hidden rank = 16. Neither of two plots contain any regularization.

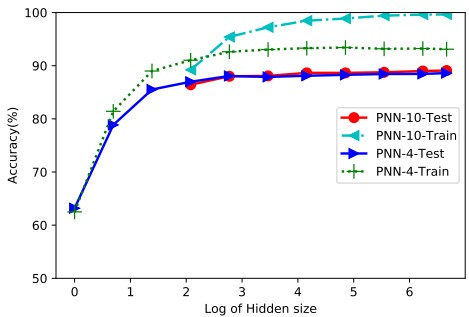

Figure 6: Accuracy of PN-4 and PN-10 when the hidden rank varies (plotted in log-scale).

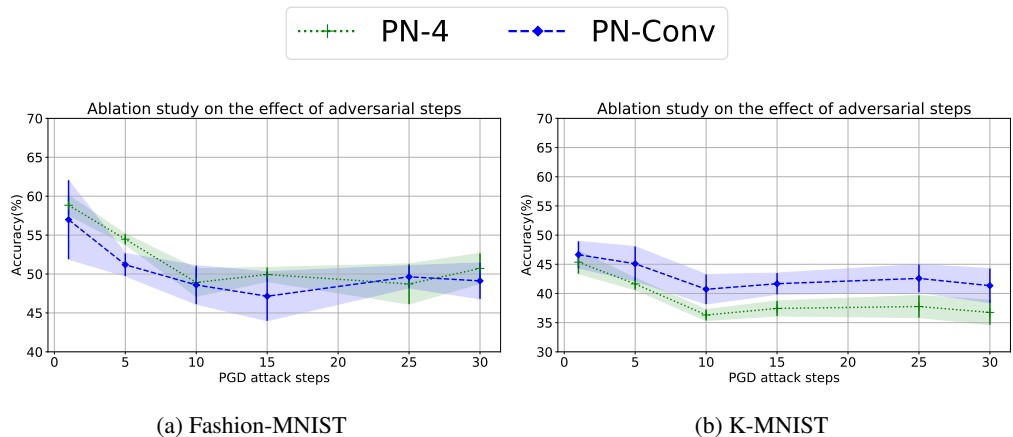

(a) Fashion-MNIST

(b) K-MNIST

Figure 7: Ablation study on the effect of adversarial steps in Fashion-MNIST and K-MNIST. All methods are run by considering a projection bound of 1.

### H.4 Evaluation of the accuracy of PNs

In this experiment, we evaluate the accuracy of PNs. We consider three networks, i.e., PN-4, PN-10 and PN-Conv, and train them under varying projection bounds using Algorithm 1. Each model is evaluated on the test set of (a) Fashion-MNIST and (b) E-MNIST.

The accuracy of each method is reported in Fig. 8, where the x-axis is plotted in log-scale (natural logarithm). The accuracy is better for bounds larger than 2 (in the log-axis) when compared to tighter bounds (i.e., values less than 0). Very tight bounds stifle the ability of the network to learn, which explains the decreased accuracy. Interestingly, PN-4 reaches similar accuracy to PN-10 and PN-Conv in Fashion-MNIST as the bound increases, while in E-MNIST it cannot reach the same performance as the bound increases. The best bounds for all three models are observed in the intermediate values, i.e., in the region of 1 in the log-axis for PN-4 and PN-10.

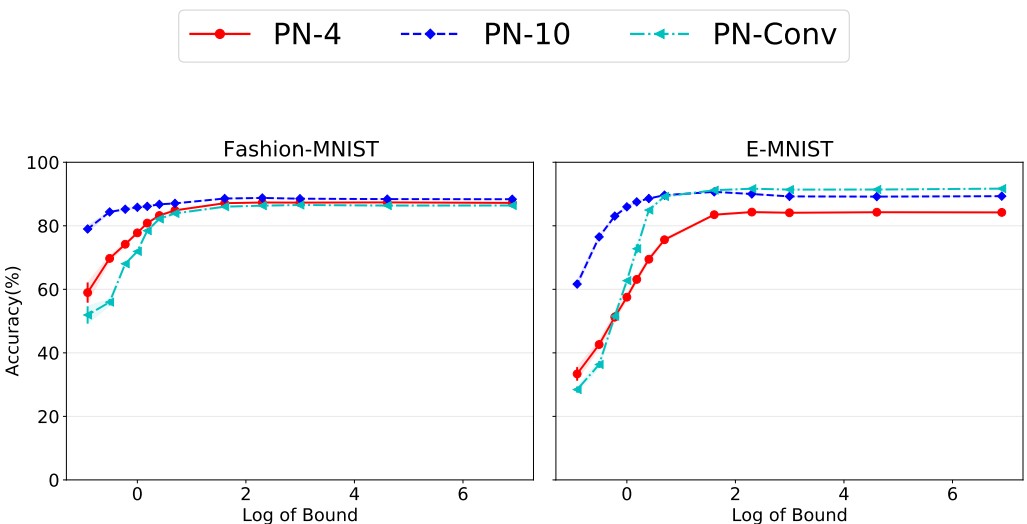

Figure 8: Accuracy of PN-4, PN-10 and PN-Conv under varying projection bounds (x-axis in log-scale) learned on (a) Fashion-MNIST, (b) E-MNIST. Notice that the performance increases for intermediate values, while it deteriorates when the bound is very tight.

We scrutinize further the projection bounds by training the same models only with cross-entropy loss (i.e., no bound regularization). In Table 10, we include the accuracy of the three networks with and without projection. Note that projection consistently improves the accuracy, particularly in the case of larger networks, i.e., PN-10.

| Method | PN-4 | PN-10 | PN-Conv |
|---|---|---|---|
| | | *Fashion-MNIST* | |
| No projection | $87.28 \pm 0.18\%$ | $88.48 \pm 0.17\%$ | $86.36 \pm 0.21\%$ |
| Projection | $87.32 \pm 0.14\%$ | $88.72 \pm 0.12\%$ | $86.38 \pm 0.26\%$ |
| | | *E-MNIST* | |
| No projection | $84.27 \pm 0.26\%$ | $89.31 \pm 0.09\%$ | $91.49 \pm 0.29\%$ |
| Projection | $84.34 \pm 0.31\%$ | $90.56 \pm 0.10\%$ | $91.57 \pm 0.19\%$ |

Table 10: The accuracy of different PN models on Fashion-MNIST (top) and E-MNIST (bottom) when trained only with SGD (first row) and when trained with projection (last row).

## H.5 Experimental results on additional datasets

To validate even further we experiment with additional datasets. We describe the datasets below and then present the robust accuracy in each case. The experimental setup remains the same as in Section 4.2 in the main paper. As a reminder, we are evaluating the robustness of the different models under adversarial noise.

**Dataset details:** There are six datasets used in this work:

1. *Fashion-MNIST* (Xiao et al., 2017) includes grayscale images of clothing. The training set consists of $60,000$ examples, and the test set of $10,000$ examples. The resolution of each image is $28 \times 28$, with each image belonging to one of the 10 classes.

2. *E-MNIST* (Cohen et al., 2017) includes handwritten character and digit images with a training set of $124,800$ examples, and a test set of $20,800$ examples. The resolution of each image is $28 \times 28$. E-MNIST includes 26 classes. We also use the variant *EMNIST-BY* that includes 62 classes with $697,932$ examples for training and $116,323$ examples for testing.

3. *K-MNIST* (Clanuwat et al., 2018) depicts grayscale images of Hiragana characters with a training set of $60,000$ examples, and a test set of $10,000$ examples. The resolution of each image is $28 \times 28$. K-MNIST has 10 classes.

4. *MNIST* (Lecun et al., 1998) includes handwritten digits images. MNIST has a training set of $60,000$ examples, and a test set of $10,000$ examples. The resolution of each image is $28 \times 28$.

5. *CIFAR-10* (Krizhevsky et al., 2014) depicts images of natural scenes. CIFAR-10 has a training set of $50,000$ examples, and a test set of $10,000$ examples. The resolution of each RGB image is $32 \times 32$.

6. *NSYNTH* (Engel et al., 2017) is an audio dataset containing $305,979$ musical notes, each with a unique pitch, timbre, and envelope.

We provide a visualization[3] of indicative samples from MNIST, Fashion-MNIST, K-MNIST and E-MNIST in Fig. 9.

We originally train PN-4, PN-10 and PN-Conv without projection bounds. The results are reported in Table 11 (columns titled 'No proj') for MNIST and K-MNIST, Table 13 (columns titled 'No proj') for E-MNIST-BY and Table 14 (columns titled 'No proj') for NSYNTH. Next, we consider the performance under varying projection bounds; the accuracy in each case is depicted in Fig. 10 for K-MNIST, MNIST and E-MNIST-BY and Fig. 11 for NSYNTH. The figures (and the tables) depict the same patterns that emerged in the two main experiments, i.e., the performance can be vastly improved for intermediate values of the projection bound. Similarly, we validate the performance when using adversarial training. The results in Table 12 demonstrate the benefits of using projection bounds even in the case of adversarial training.

## H.6 Experimental results of more types of attacks

To further verify the results of the main paper, we conduct experiments with three additional adversarial attacks: a) FGSM with $\epsilon = 0.01$, b) Projected Gradient Descent in Trades (TPGD) (Zhang et al., 2019), c) Targeted Auto-Projected Gradient Descent (APGDT) (Croce and Hein, 2020). In TPGD and APGDT, we use the default parameters for a one-step attack.

The quantitative results are reported in Table 15 for four datasets and the curves of Fashion-MNIST and E-MNIST are visualized in Fig. 12 and the curves of K-MNIST and MNIST are visualized in Fig. 13. The results in both cases remain similar to the attacks in the main paper, i.e., the proposed projection improves the performance *consistently* across attacks, types of networks and adversarial attacks.

## H.7 Experimental results in NCP model

To complement, the results of the CCP model, we conduct an experiment using the NCP model. That is, we use a $4^{\text{th}}$ degree polynomial expansion, called NCP-4, for our experiment. We conduct an

---

[3]The samples were found in https://www.tensorflow.org/datasets/catalog.

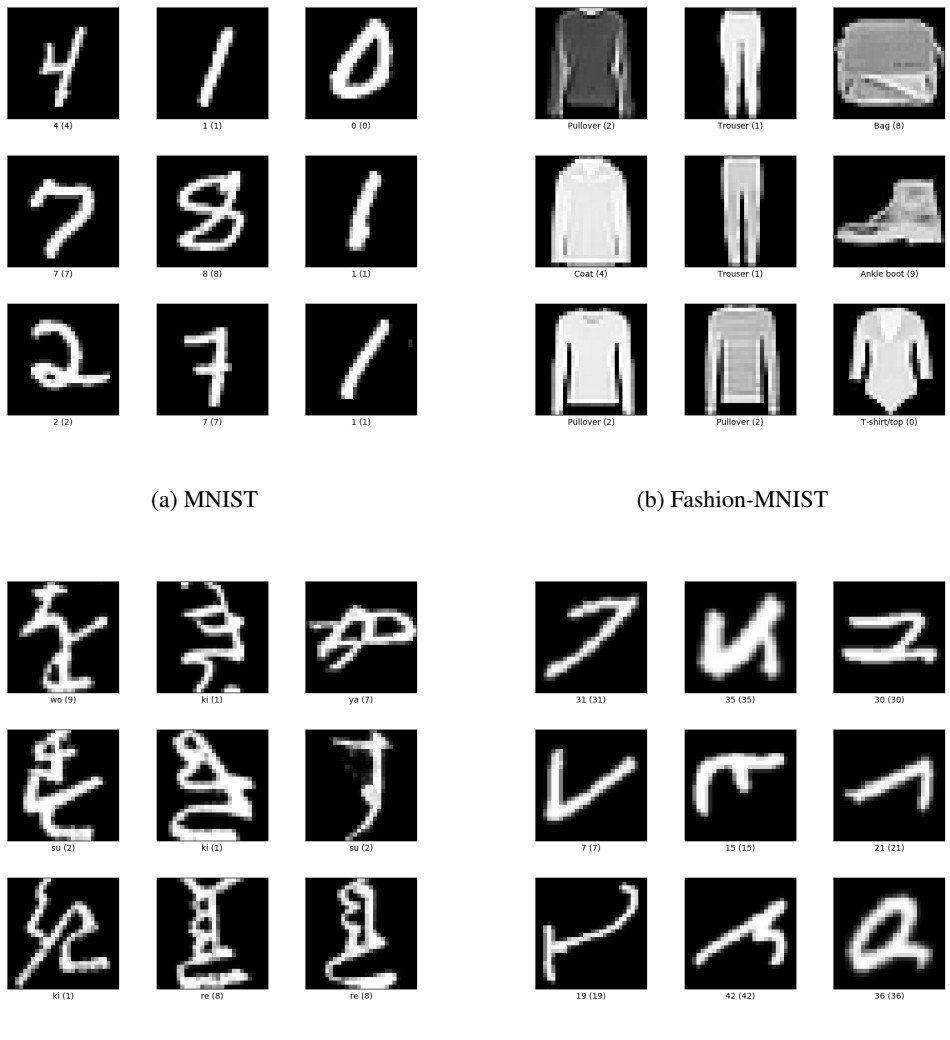

(a) MNIST  (b) Fashion-MNIST

(c) K-MNIST  (d) E-MNIST

Figure 9: Samples from the datasets used for the numerical evidence. Below each image, the class name and the class number are denoted.

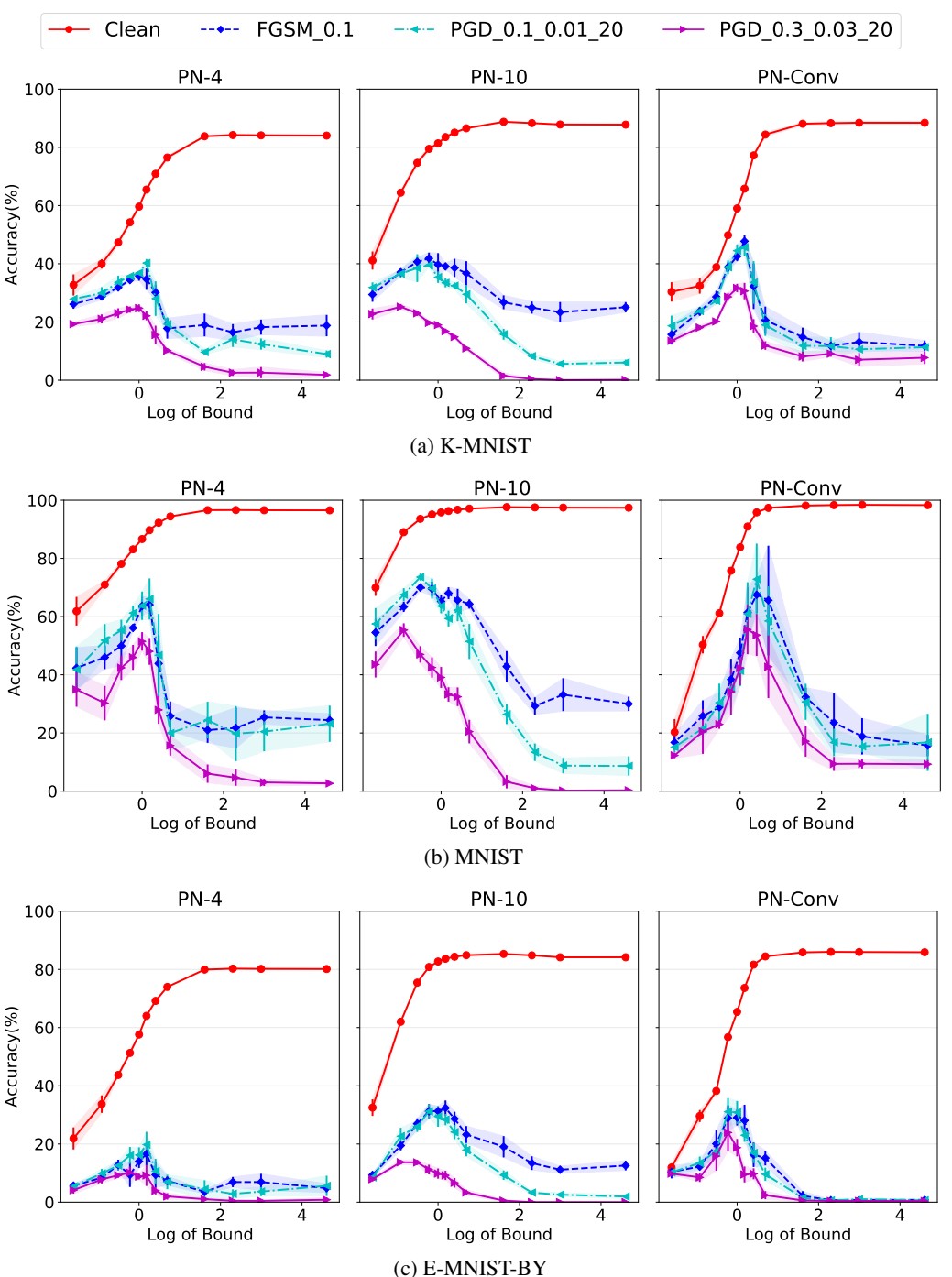

Figure 10: Adversarial attacks during testing on (a) K-MNIST (top), (b) MNIST (middle), (c) E-MNIST-BY (bottom) with the x-axis is plotted in log-scale. Note that intermediate values of projection bounds yield the highest accuracy. The patterns are consistent in all datasets and across adversarial attacks.

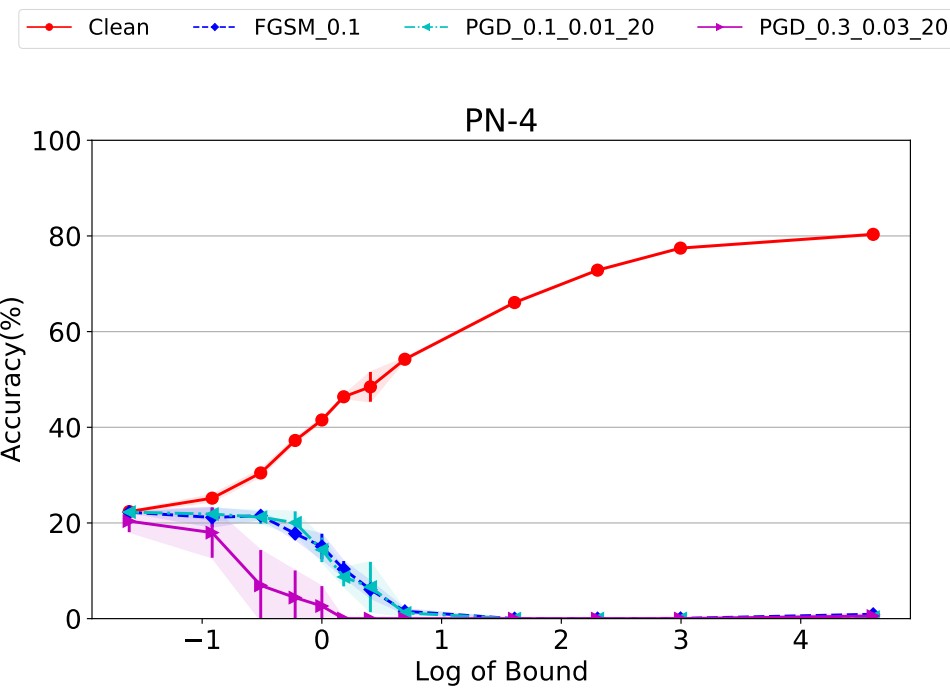

Figure 11: Adversarial attacks during testing on NSYNTH.

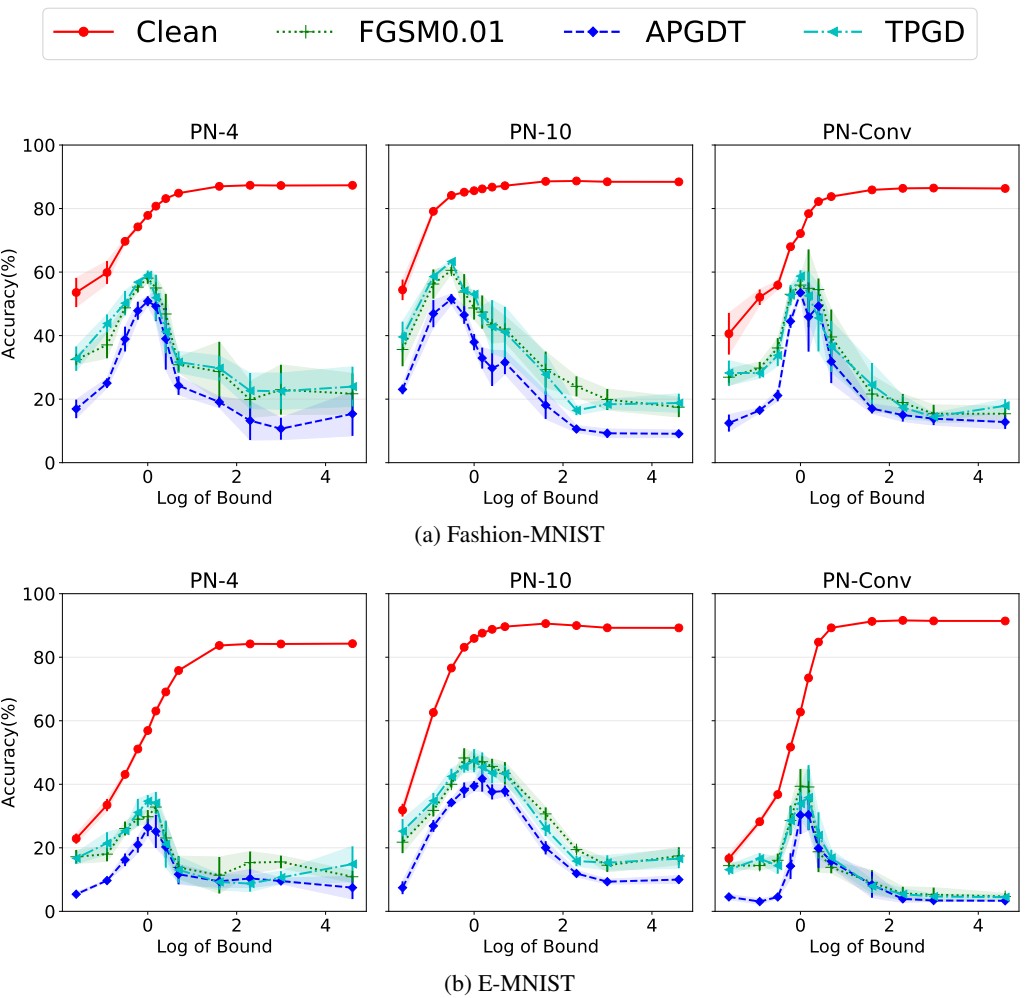

Figure 12: Three new adversarial attacks during testing on (a) Fashion-MNIST (top), (b) E-MNIST (bottom ) with the x-axis is plotted in log-scale. Note that intermediate values of projection bounds yield the highest accuracy. The patterns are consistent in both datasets and across adversarial attacks.

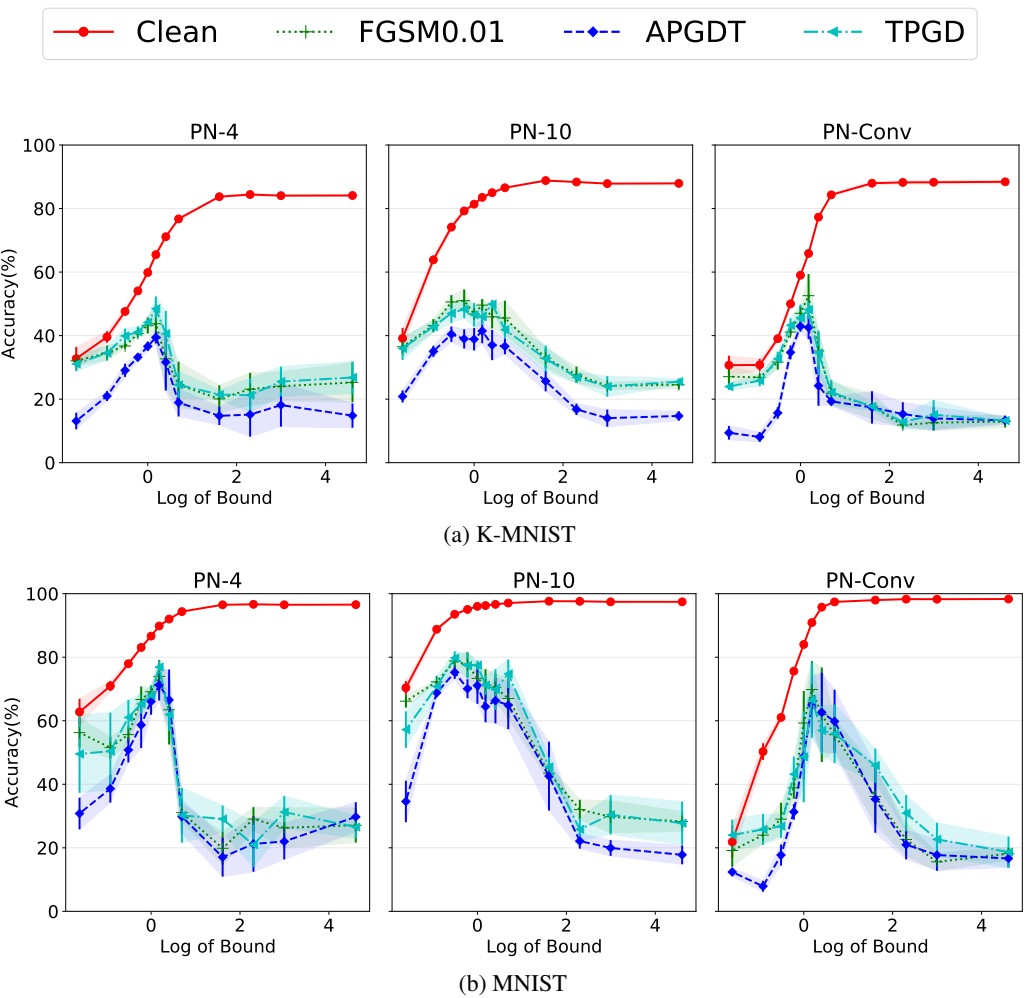

Figure 13: Three new adversarial attacks during testing on (a) K-MNIST (top), (b) MNIST (bottom) with the x-axis is plotted in log-scale. Note that intermediate values of projection bounds yield the highest accuracy. The patterns are consistent in both datasets and across adversarial attacks.

| | Method | No proj. | Our method | Jacobian | $L_2$ |
|---|---|---|---|---|---|
| | | | | | *K-MNIST* |
| PN-4 | Clean | $84.04 \pm 0.30\%$ | $\mathbf{84.23 \pm 0.30}\%$ | $83.16 \pm 0.31\%$ | $84.18 \pm 0.44\%$ |
| | FGSM-0.1 | $18.86 \pm 2.61\%$ | $\mathbf{35.84 \pm 1.67}\%$ | $22.61 \pm 1.30\%$ | $22.05 \pm 2.76\%$ |
| | PGD-(0.1, 20, 0.01) | $11.20 \pm 4.27\%$ | $\mathbf{40.26 \pm 1.36}\%$ | $16.00 \pm 5.63\%$ | $10.93 \pm 2.42\%$ |
| | PGD-(0.3, 20, 0.03) | $1.94 \pm 1.11\%$ | $\mathbf{24.75 \pm 1.32}\%$ | $4.46 \pm 2.59\%$ | $2.70 \pm 1.26\%$ |
| PN-10 | Clean | $87.90 \pm 0.24\%$ | $\mathbf{88.80 \pm 0.19}\%$ | $88.73 \pm 0.16\%$ | $87.93 \pm 0.18\%$ |
| | FGSM-0.1 | $24.52 \pm 1.44\%$ | $\mathbf{41.83 \pm 2.00}\%$ | $26.90 \pm 1.02\%$ | $26.62 \pm 1.59\%$ |
| | PGD-(0.1, 20, 0.01) | $7.54 \pm 0.79\%$ | $\mathbf{39.55 \pm 0.64}\%$ | $11.50 \pm 1.35\%$ | $5.09 \pm 0.68\%$ |
| | PGD-(0.3, 20, 0.03) | $0.05 \pm 0.04\%$ | $\mathbf{25.24 \pm 0.93}\%$ | $1.24 \pm 0.64\%$ | $0.19 \pm 0.12\%$ |
| PN-Conv | Clean | $88.41 \pm 0.37\%$ | $88.48 \pm 0.42\%$ | $86.57 \pm 0.46\%$ | $\mathbf{88.56 \pm 0.62}\%$ |
| | FGSM-0.1 | $13.34 \pm 2.01\%$ | $\mathbf{47.75 \pm 2.03}\%$ | $14.16 \pm 3.05\%$ | $12.43 \pm 2.58\%$ |
| | PGD-(0.1, 20, 0.01) | $10.81 \pm 1.25\%$ | $\mathbf{45.68 \pm 3.11}\%$ | $12.05 \pm 0.82\%$ | $11.05 \pm 0.85\%$ |
| | PGD-(0.3, 20, 0.03) | $6.91 \pm 2.04\%$ | $\mathbf{31.68 \pm 1.43}\%$ | $7.54 \pm 1.39\%$ | $6.28 \pm 2.37\%$ |
| | | | | | *MNIST* |
| PN-4 | Clean | $96.52 \pm 0.13\%$ | $\mathbf{96.62 \pm 0.17}\%$ | $95.88 \pm 0.16\%$ | $96.44 \pm 0.18\%$ |
| | FGSM-0.1 | $20.96 \pm 5.16\%$ | $\mathbf{64.09 \pm 2.41}\%$ | $33.59 \pm 8.46\%$ | $26.07 \pm 5.64\%$ |
| | PGD-(0.1, 20, 0.01) | $14.23 \pm 5.39\%$ | $\mathbf{66.05 \pm 7.06}\%$ | $20.83 \pm 5.64\%$ | $16.06 \pm 5.84\%$ |
| | PGD-(0.3, 20, 0.03) | $2.59 \pm 2.01\%$ | $\mathbf{51.47 \pm 3.17}\%$ | $4.92 \pm 1.18\%$ | $4.26 \pm 2.44\%$ |
| PN-10 | Clean | $97.46 \pm 0.11\%$ | $\mathbf{97.63 \pm 0.06}\%$ | $97.36 \pm 0.05\%$ | $97.53 \pm 0.10\%$ |
| | FGSM-0.1 | $30.12 \pm 4.58\%$ | $\mathbf{70.02 \pm 1.28}\%$ | $40.22 \pm 2.31\%$ | $28.77 \pm 2.41\%$ |
| | PGD-(0.1, 20, 0.01) | $9.70 \pm 2.11\%$ | $\mathbf{73.57 \pm 1.17}\%$ | $18.74 \pm 5.39\%$ | $10.91 \pm 2.32\%$ |
| | PGD-(0.3, 20, 0.03) | $0.47 \pm 0.53\%$ | $\mathbf{55.36 \pm 2.32}\%$ | $2.49 \pm 1.46\%$ | $0.44 \pm 0.29\%$ |
| PN-Conv | Clean | $98.32 \pm 0.12\%$ | $\mathbf{98.40 \pm 0.12}\%$ | $97.88 \pm 0.12\%$ | $98.32 \pm 0.11\%$ |
| | FGSM-0.1 | $18.98 \pm 2.99\%$ | $\mathbf{67.50 \pm 6.22}\%$ | $27.02 \pm 9.88\%$ | $23.77 \pm 5.58\%$ |
| | PGD-(0.1, 20, 0.01) | $12.57 \pm 2.81\%$ | $\mathbf{72.85 \pm 12.23}\%$ | $13.96 \pm 2.57\%$ | $13.84 \pm 3.18\%$ |
| | PGD-(0.3, 20, 0.03) | $10.57 \pm 4.08\%$ | $\mathbf{55.56 \pm 8.48}\%$ | $10.22 \pm 0.52\%$ | $9.10 \pm 3.62\%$ |

Table 11: Comparison of regularization techniques on K-MNIST (top) and MNIST (bottom). In each dataset, the base networks are PN-4, i.e., a $4^{\text{th}}$ degree polynomial, on the top four rows, PN-10, i.e., a $10^{\text{th}}$ degree polynomial, on the middle four rows and PN-Conv, i.e., a $4^{\text{th}}$ degree polynomial with convolutions, on the bottom four rows. Our projection method exhibits the best performance in all three attacks, with the difference on accuracy to stronger attacks being substantial.

| Method | AT | Our method + AT | Jacobian + AT | $L_2$ + AT |
|---|---|---|---|---|
| | Adversarial training (AT) with PN-10 on *K-MNIST* | | | |
| FGSM-0.1 | $70.93 \pm 0.46\%$ | $\mathbf{71.14 \pm 0.30}\%$ | $64.48 \pm 0.51\%$ | $70.90 \pm 0.57\%$ |
| PGD-(0.1, 20, 0.01) | $60.94 \pm 0.71\%$ | $61.20 \pm 0.39\%$ | $57.89 \pm 0.31\%$ | $\mathbf{61.47 \pm 0.44}\%$ |
| PGD-(0.3, 20, 0.03) | $30.77 \pm 0.26\%$ | $\mathbf{33.07 \pm 0.58}\%$ | $29.96 \pm 0.21\%$ | $30.35 \pm 0.42\%$ |
| | Adversarial training (AT) with PN-10 on *MNIST* | | | |
| FGSM-0.1 | $91.89 \pm 0.30\%$ | $91.94 \pm 0.17\%$ | $87.85 \pm 0.27\%$ | $\mathbf{92.22 \pm 0.30}\%$ |
| PGD-(0.1, 20, 0.01) | $87.36 \pm 0.29\%$ | $\mathbf{87.38 \pm 0.37}\%$ | $84.96 \pm 0.25\%$ | $87.26 \pm 0.49\%$ |
| PGD-(0.3, 20, 0.03) | $61.96 \pm 0.92\%$ | $\mathbf{63.96 \pm 1.02}\%$ | $62.24 \pm 0.24\%$ | $62.44 \pm 0.76\%$ |

Table 12: Comparison of regularization techniques on (a) K-MNIST (top) and (b) MNIST (bottom) along with adversarial training (AT). The base network is a PN-10, i.e., $10^{\text{th}}$ degree polynomial. Our projection method exhibits the best performance in all three attacks.

experiment in the K-MNIST dataset and present the result with varying bound in Fig. 14. Notice that the patterns remain similar to the CCP model, i.e., intermediate values of the projection bound can increase the performance significantly.

## H.8 LAYER-WISE BOUND

To assess the flexibility of the proposed method, we assess the performance of the layer-wise bound. In the previous sections, we have considered using a single bound for all the matrices, i.e., $\|\boldsymbol{U}_i\|_\infty \leq \lambda$, because the projection for a single matrix has efficient projection algorithms. However, Lemma 1 enables each matrix $\boldsymbol{U}_i$ to have a different bound $\lambda_i$. We assess the performance of having different bounds for each matrix $\boldsymbol{U}_i$.

| | Method | PN-4, PN-10 and PN-Conv on E-MNIST-BY | |
|---|---|---|---|
| | | No proj. | Our method |
| PN-4 | Clean | $80.18 \pm 0.19\%$ | $\mathbf{80.26 \pm 0.17}\%$ |
| | FGSM-0.1 | $3.65 \pm 0.76\%$ | $\mathbf{16.58 \pm 3.87}\%$ |
| | PGD-(0.1, 20, 0.01) | $4.57 \pm 1.98\%$ | $\mathbf{19.77 \pm 4.42}\%$ |
| | PGD-(0.3, 20, 0.03) | $0.59 \pm 0.40\%$ | $\mathbf{10.13 \pm 2.08}\%$ |
| PN-10 | Clean | $84.17 \pm 0.06\%$ | $\mathbf{85.32 \pm 0.04}\%$ |
| | FGSM-0.1 | $11.67 \pm 1.21\%$ | $\mathbf{32.37 \pm 2.58}\%$ |
| | PGD-(0.1, 20, 0.01) | $2.48 \pm 0.66\%$ | $\mathbf{31.22 \pm 2.32}\%$ |
| | PGD-(0.3, 20, 0.03) | $0.03 \pm 0.05\%$ | $\mathbf{13.74 \pm 0.77}\%$ |
| PN-Conv | Clean | $85.92 \pm 0.08\%$ | $\mathbf{86.03 \pm 0.08}\%$ |
| | FGSM-0.1 | $0.65 \pm 0.17\%$ | $\mathbf{29.07 \pm 2.72}\%$ |
| | PGD-(0.1, 20, 0.01) | $1.57 \pm 1.40\%$ | $\mathbf{31.06 \pm 4.70}\%$ |
| | PGD-(0.3, 20, 0.03) | $0.33 \pm 0.06\%$ | $\mathbf{23.93 \pm 6.32}\%$ |

Table 13: Comparison of regularization techniques on E-MNIST-BY. The base network are PN-4, i.e., $4^{\text{th}}$ degree polynomial, on the top four rows, PN-10, i.e., $10^{\text{th}}$ degree polynomial, on the middle four rows and PN-Conv, i.e., a $4^{\text{th}}$ degree polynomial with convolution, on the bottom four rows. Our projection method exhibits the best performance in all three attacks, with the difference on accuracy to stronger attacks being substantial.

| Model | PN-4 | |
|---|---|---|
| Projection | No-proj | Proj |
| Clean accuracy | $80.25 \pm 0.27\%$ | $\mathbf{80.33 \pm 0.26}\%$ |
| FGSM-0.1 | $0.91 \pm 0.14\%$ | $\mathbf{22.25 \pm 0.04}\%$ |
| PGD-(0.1, 20, 0.01) | $0.31 \pm 0.11\%$ | $\mathbf{22.27 \pm 0.00}\%$ |
| PGD-(0.3, 20, 0.03) | $0.46 \pm 0.29\%$ | $\mathbf{20.38 \pm 2.30}\%$ |

Table 14: Evaluation of the robustness of PN models on NSYNTH. Each line refers to a different adversarial attack. The projection offers an improvement in the accuracy in each case; in PGD attacks projection improves the accuracy by a remarkable margin.

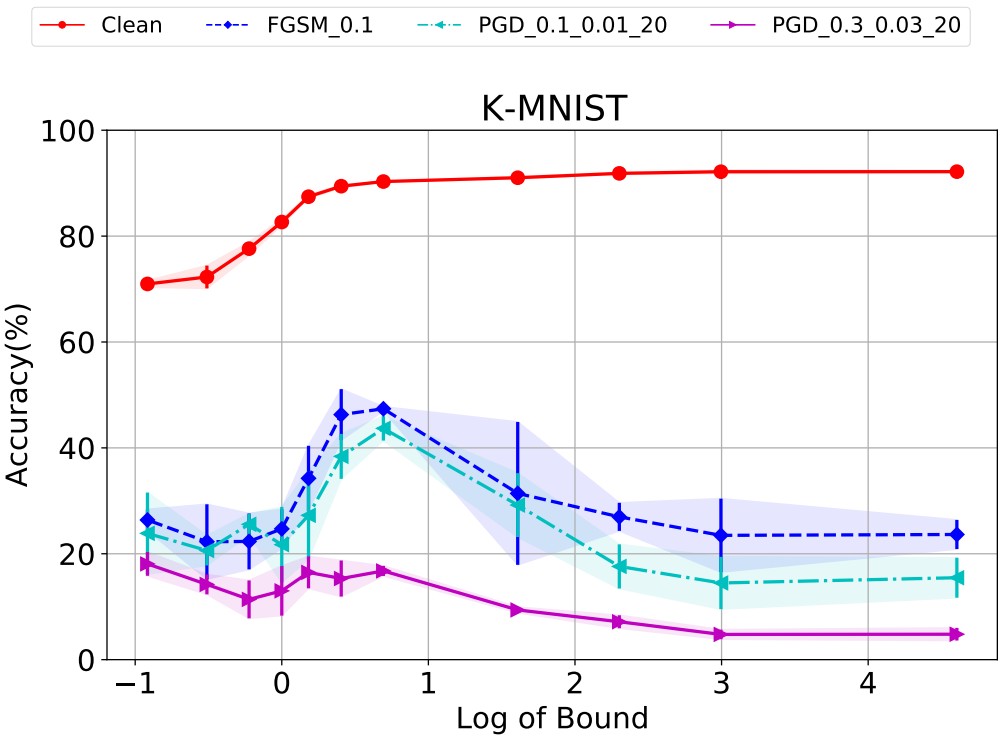

Figure 14: Experimental result of K-MNIST in NCP model.

| | Method | No proj. | Our method |
|---|---|---|---|
| | | | *Fashion-MNIST* |
| | FGSM-0.01 | $26.49 \pm 3.13\%$ | $\mathbf{58.09 \pm 1.63\%}$ |
| PN-4 | APGDT | $16.59 \pm 4.35\%$ | $\mathbf{50.83 \pm 1.55\%}$ |
| | TPGD | $26.88 \pm 6.78\%$ | $\mathbf{59.03 \pm 1.45\%}$ |
| | FGSM-0.01 | $18.59 \pm 1.82\%$ | $\mathbf{60.56 \pm 1.06\%}$ |
| PN-10 | APGDT | $8.76 \pm 1.14\%$ | $\mathbf{51.93 \pm 1.91\%}$ |
| | TPGD | $14.53 \pm 1.49\%$ | $\mathbf{63.33 \pm 0.51\%}$ |
| | FGSM-0.01 | $15.30 \pm 3.10\%$ | $\mathbf{55.90 \pm 2.60\%}$ |
| PN-Conv | APGDT | $11.88 \pm 1.33\%$ | $\mathbf{53.49 \pm 0.72\%}$ |
| | TPGD | $14.50 \pm 1.59\%$ | $\mathbf{58.72 \pm 1.87\%}$ |
| | | | *E-MNIST* |
| | FGSM-0.01 | $13.40 \pm 5.16\%$ | $\mathbf{32.83 \pm 2.08\%}$ |
| PN-4 | APGDT | $9.33 \pm 4.00\%$ | $\mathbf{26.38 \pm 2.70\%}$ |
| | TPGD | $17.40 \pm 3.11\%$ | $\mathbf{34.68 \pm 1.92\%}$ |
| | FGSM-0.01 | $14.47 \pm 1.80\%$ | $\mathbf{48.28 \pm 3.06\%}$ |
| PN-10 | APGDT | $10.13 \pm 0.93\%$ | $\mathbf{41.72 \pm 4.05\%}$ |
| | TPGD | $13.97 \pm 0.88\%$ | $\mathbf{47.44 \pm 3.62\%}$ |
| | FGSM-0.01 | $4.71 \pm 1.10\%$ | $\mathbf{39.37 \pm 5.43\%}$ |
| PN-Conv | APGDT | $3.58 \pm 0.66\%$ | $\mathbf{30.43 \pm 4.87\%}$ |
| | TPGD | $4.08 \pm 0.33\%$ | $\mathbf{35.85 \pm 10.20\%}$ |
| | | | *K-MNIST* |
| | FGSM-0.01 | $23.31 \pm 5.34\%$ | $\mathbf{43.74 \pm 5.97\%}$ |
| PN-4 | APGDT | $17.02 \pm 6.97\%$ | $\mathbf{39.43 \pm 1.89\%}$ |
| | TPGD | $23.45 \pm 7.67\%$ | $\mathbf{48.46 \pm 3.84\%}$ |
| | FGSM-0.01 | $26.87 \pm 2.14\%$ | $\mathbf{50.99 \pm 3.52\%}$ |
| PN-10 | APGDT | $16.23 \pm 1.32\%$ | $\mathbf{41.46 \pm 3.85\%}$ |
| | TPGD | $22.63 \pm 0.99\%$ | $\mathbf{49.91 \pm 1.37\%}$ |
| | FGSM-0.01 | $12.31 \pm 2.03\%$ | $\mathbf{52.58 \pm 6.80\%}$ |
| PN-Conv | APGDT | $13.47 \pm 2.19\%$ | $\mathbf{42.94 \pm 1.68\%}$ |
| | TPGD | $14.25 \pm 2.51\%$ | $\mathbf{48.19 \pm 3.02\%}$ |
| | | | *MNIST* |
| | FGSM-0.01 | $34.14 \pm 7.63\%$ | $\mathbf{73.95 \pm 5.18\%}$ |
| PN-4 | APGDT | $29.88 \pm 9.47\%$ | $\mathbf{71.26 \pm 4.88\%}$ |
| | TPGD | $27.01 \pm 9.77\%$ | $\mathbf{76.88 \pm 1.98\%}$ |
| | FGSM-0.01 | $32.34 \pm 4.67\%$ | $\mathbf{78.83 \pm 1.63\%}$ |
| PN-10 | APGDT | $19.55 \pm 1.72\%$ | $\mathbf{75.22 \pm 2.05\%}$ |
| | TPGD | $28.11 \pm 3.87\%$ | $\mathbf{79.74 \pm 2.07\%}$ |
| | FGSM-0.01 | $22.73 \pm 3.10\%$ | $\mathbf{69.83 \pm 8.91\%}$ |
| PN-Conv | APGDT | $17.95 \pm 3.39\%$ | $\mathbf{64.94 \pm 8.96\%}$ |
| | TPGD | $21.82 \pm 3.07\%$ | $\mathbf{66.47 \pm 11.83\%}$ |

Table 15: Evaluation of the robustness of PN models on four datasets with three new types of attacks. Each line refers to a different adversarial attack. The projection offers an improvement in the accuracy in each case.

We experiment on PN-4 that we set a different projection bound for each matrix $\boldsymbol{U}_i$. Specifically, we use five different candidate values for each $\lambda_i$ and then perform the grid search on the Fashion-MNIST FGSM-0.01 attack. The results on Fashion-MNIST in Table 16 exhibit how the layer-wise bounds outperform the previously used single bound[4]. The best performing values for PN-4 are $\lambda_1 = 1.5, \lambda_2 = 2, \lambda_3 = 1.5, \lambda_4 = 2, \mu = 0.8$. The values of $\lambda_i$ in the first few layers are larger, while the value in the output matrix $C$ is tighter.

To scrutinize the results even further, we evaluate whether the best performing $\lambda_i$ can improve the performance in different datasets and the FGSM-0.1 attack. In both cases, the best performing $\lambda_i$ can improve the performance of the single bound.

---

[4] The single bound is mentioned as 'Our method' in the previous tables. In this experiment both 'single bound' and 'layer-wise bound' are proposed.

| Method | No proj. | Jacobian | $L_2$ | Single bound | Layer-wise bound |
|---|---|---|---|---|---|
| | | | *Fashion-MNIST* | | |
| FGSM-0.01 | $26.49 \pm 3.13\%$ | $39.88 \pm 4.59\%$ | $24.36 \pm 1.95\%$ | $58.09 \pm 1.63\%$ | $\mathbf{63.95 \pm 1.26}\%$ |
| FGSM-0.1 | $12.92 \pm 2.74\%$ | $17.90 \pm 6.51\%$ | $13.80 \pm 3.65\%$ | $46.43 \pm 0.95\%$ | $\mathbf{55.14 \pm 3.65}\%$ |
| | | | *K-MNIST* | | |
| FGSM-0.01 | $23.31 \pm 5.34\%$ | $25.46 \pm 3.51\%$ | $27.85 \pm 7.62\%$ | $43.74 \pm 5.97\%$ | $\mathbf{49.61 \pm 1.44}\%$ |
| FGSM-0.1 | $18.86 \pm 2.61\%$ | $22.61 \pm 1.30\%$ | $22.05 \pm 2.76\%$ | $35.84 \pm 1.67\%$ | $\mathbf{47.54 \pm 3.74}\%$ |
| | | | *MNIST* | | |
| FGSM-0.01 | $34.14 \pm 7.63\%$ | $32.78 \pm 6.94\%$ | $29.31 \pm 3.95\%$ | $73.95 \pm 5.18\%$ | $\mathbf{79.23 \pm 3.65}\%$ |
| FGSM-0.1 | $20.96 \pm 5.16\%$ | $33.59 \pm 8.46\%$ | $26.07 \pm 5.64\%$ | $64.09 \pm 2.41\%$ | $\mathbf{74.97 \pm 5.60}\%$ |

Table 16: Evaluation of our layer-wise bound versus our single bound. To avoid confusion with previous results, note that 'single bound' corresponds to 'Our method' in the rest of the tables in this work. The different $\lambda_i$ values are optimized on Fashion-MNIST FGSM-0.01 attack. Then, the same $\lambda_i$ values are used for training the rest of the methods. The proposed layer-wise bound outperforms the single bound by a large margin, improving even further by baseline regularization schemes.

| Method | No proj. | Single bound | Layer-wise bound | Gaussian denoising | Median denoising | Guided denoising |
|---|---|---|---|---|---|---|
| | | | *Fashion-MNIST* | | | |
| FGSM-0.01 | $26.49 \pm 3.13\%$ | $58.09 \pm 1.63\%$ | $\mathbf{63.95 \pm 1.26}\%$ | $18.80 \pm 3.08\%$ | $19.68 \pm 3.20\%$ | $29.69 \pm 5.37\%$ |
| FGSM-0.1 | $12.92 \pm 2.74\%$ | $46.43 \pm 0.95\%$ | $\mathbf{55.14 \pm 3.65}\%$ | $14.14 \pm 2.77\%$ | $14.02 \pm 1.95\%$ | $22.94 \pm 5.65\%$ |

Table 17: Comparison of the proposed method against adversarial defense methods on feature denoising (Xie et al., 2019) and guided denoising (Liao et al., 2018). Notice that the single bound (cf. Appendix H.8 for details) already outperforms the proposed defense methods, while the layer-wise bounds further improves upon our single bound case.

## H.9 ADVERSARIAL DEFENSE METHOD

One frequent method used against adversarial perturbations are the so called adversarial defense methods. We assess the performance of adversarial defense methods on the PNs when compared with the proposed method.

We experiment on PN-4 in Fashion-MNIST. We chose three different methods: gaussian denoising, median denoising and guided denoising (Liao et al., 2018). Gaussian denoising and median denoising are the methods of using gaussian filter and median filter for feature denoising (Xie et al., 2019). The results in Table 17 show that in both attacks our method performs favourably to the adversarial defense methods.

