# OpenReview forum: "Controlling the Complexity and Lipschitz Constant improves Polynomial Nets"
_ICLR.cc/2022/Conference — ICLR 2022 Poster_

### Official Review · Reviewer_GLBZ · 2021-11-01

**Correctness:** 3
**Technical Novelty And Significance:** 3
**Empirical Novelty And Significance:** 2
**Recommendation:** 5
**Confidence:** 3

**Main Review:**

Strength: The paper contributes both theoretically and empirically. It provides the first known generalization error bounds and Lipschitz constant bounds. Given its theoretical contribution, regularization schemes are proposed to boost the performance of PNs.

Weakness: The theoretical contribution of the paper is useful but not significant. It finds ways to translate the complexity problem of CCP and NCP to functions of R(V) and uses Massart Lemma to complete the proofs, which is relatively less challenging. Most of my concern comes from the presentation of the paper: (list in order of the sections)
1. The C in Equation NCP is capitalized but not in the appendix.
2. The constraint on C in Equation 6 is not on infinity norm but should be on 1 norm.
3. The reference order is messy in the paper (e.g. Table 1 appears before Table 2), making it hard to read.
4. Figure 3 and Table 2 are never referred to (pls correct me if I am wrong), neither in the main text nor in the appendix.
5. Appendix G, bullet point 5, did not mention CIFAR10.
6. Appendix G.5, "from all four datasets in Fig. 9"? There are altogether 6 datasets as listed by the author right above that line.
7. Appendix G.5, Table 11 is about MNIST and K-MNIST, not MNIST and E-MNIST.
8. Although in the summary of main contributions, the author mentioned the sweet-spot for regularization parameter and accuracy robustness trade-off, there are little to no descriptions/explanations/justifications for them in the experiment section. The "Log of Bound" might be the regularization parameter that the authors were referring to, but still, the lack of sufficient descriptions of the data shown makes things hard to comprehend.

Besides the problem of paper presentation, I have concerns about the overall performance of the method on more challenging datasets. From Figure 3 and Table 2, the accuracy of CIFAR10 seems low and there is no comparison with other baselines. Since the method seems to be sensitive to the regularization parameter, can the authors give intuitions or guidance on its choice?

**Summary Of The Paper:**

The paper derives empirical Rademacher complexity bounds for two variants of PN, along with their Lipschitz constants. The derived bounds motivate practical regularization schemes that are implemented based on projected gradient descent. Empirical results demonstrate the usefulness of the proposed method.

**Summary Of The Review:**

Although the technical contribution of this paper is sound, the paper is not ready for publication due to the problems mentioned above.

---

> ### Author Response · Authors · 2021-11-18
> **Response to the Reviewer GLBZ (Part 2 of 2)**
>
>
> > **Q9.** I have concerns about the overall performance of the method on more challenging datasets. From Figure 3 and Table 2, the accuracy of CIFAR10 seems low and there is no comparison with other baselines.
>
> **A9.** We appreciate the remark from the reviewer. We have added the respective baselines in Table 2. The results indicate that the proposed method outperforms the baselines.
>
> |CIFAR-10 PN-Conv|No proj|Our method|Jacobian|L-2|
> |----------------|-------------------------------|-----------------------------|---------------|-------------------------------|
> |Clean accuracy|65.09±0.14%|**65.22±0.13%**|64.43±0.19%|65.11±0.08%|
> |FGSM0.1|6.00±0.53%|**15.13±0.81%**|3.34±0.40%|1.27±0.10%|
> |PGD-(0.1, 20, 0.01)|7.08±0.68%|**15.17±0.88%**|1.74±0.14%|1.05±0.05%|
> |PGD-(0.3, 20, 0.03)|0.41±0.09%|**11.71±1.11%**|0.04±0.02%|0.51±0.04%|
>
> _______
>
> > **Q10.**  Since the method seems to be sensitive to the regularization parameter, can the authors give intuitions or guidance on its choice?
>
> **A10.** There seems to be a pattern that is consistent in all our experiments that intermediate values of the bound result in the best accuracy. To be specific for adversarial training (AT), this region in the PN-4 and PN-Conv is around $\log(\lambda) = 0$, i.e. $\lambda = 1$, while in the case of the PN-10, this region is around $\lambda = 0.6$.
>
> In the revised paper, upon the request by the reviewer opuA we have also conducted a series of experiments with different layer-wise bounds (more details on the corresponding response). The insights from this large set of experiments is that a larger bound can be used for the first few layers (namely the layers closer to the input signal) and a tighter (smaller) bound can be used for the layers closer to the output.

---

> > ### Author Response · Authors · 2021-11-29
> > **Are the concerns of the presentation addressed?**
> >
> > Dear reviewer GLBZ,
> >
> > given that the discussion period is closing in the next few hours, we would appreciate it if the reviewer mentions any further concerns on the presentation or re-evaluates our improved submission. We appreciate the constructive feedback provided so far.

---

> > > ### Comment · Reviewer_GLBZ · 2021-11-30
> > > **Post rebuttal review**
> > >
> > > I would like to thank the authors for their response. The response did answer some of my concerns but I still think the paper is somewhat weak theoretically. The presentation of the findings can also benefit from a careful read. Therefore I am keeping my score.

---

> > > > ### Author Response · Authors · 2021-11-30
> > > > **Reply to the reviewer GLBZ**
> > > >
> > > > Dear reviewer GLBZ,
> > > >
> > > > we are thankful for the response of the reviewer, but we are unsure what the reviewer means the paper is weak theoretically.
> > > >
> > > > As a reminder, the [reviewer guidelines for ICLR](https://iclr.cc/Conferences/2022/ReviewerGuide) mention the following two proposals:
> > > > *  Submissions bring value to the ICLR community when they convincingly demonstrate new, relevant, impactful, or insightful knowledge.
> > > > * Don’t reject a paper just because you don’t find it “interesting”. This should not be a criterion at all for accepting/rejecting a paper. The research community is so big that somebody will find some value in the paper (maybe even a few years down the road), even if you don’t see it right now.
> > > >
> > > > The *original review* identified the *theoretical contribution as sound and useful*, and *highlighted the presentation issues* as the *major bottleneck*. The *presentation issues* identified *have been fixed*, which to our mind already improved the paper. The theoretical contribution, which is the *'first known generalization error bounds and Lipschitz constant bound'* (as the original review mentions), is utilised to obtain a regularization scheme, which provides substantial improvements. In addition, the empirical improvements have been improved on the rebuttal phase as can be seen from the general [response](https://openreview.net/forum?id=dQ7Cy_ndl1s&noteId=2PO9Jc1RP6).
> > > >
> > > > All in all, we provide sound and **useful** theoretical results (as mentioned in the original review) for a **new** class of functions that boost the performance of PNs (as mentioned in the original review).
> > > >
> > > > We would like to understand what additional result the reviewer would like to see in our submission to further improve their opinion.

---

> ### Author Response · Authors · 2021-11-18
> **Response to the Reviewer GLBZ (Part 1 of 2)**
>
> > **Q0.** The theoretical contribution of the paper is useful but not significant. It finds ways to translate the complexity problem of CCP and NCP to functions of R(V) and uses Massart Lemma to complete the proofs, which is relatively less challenging.
>
> **A0.** We appreciate that the reviewer finds our theoretical contribution to be useful. We remark that an important goal is to improve the performance of methods in downstream tasks. As our experimental section shows, we achieve practical improvements derived from the theory presented. Thus from a practical perspective we think the theoretical contribution is valuable.
>
> We would like to point out that the relatively simpler proof structure (compared to regular neural networks) is due to the mature state of Tensor Algebra identities/inequalities. In our opinion this is not necessarily something negative. The derivation of equivalent Rademacher Complexity bounds for Neural Networks with activation functions might require more complicated arguments, but we believe some researchers could regard the simpler proof structure for Polynomial Networks as an advantage.
> _______
> > **Q1&2.** The $C$ in Equation NCP is capitalized but not in the appendix. The constraint on $C$ in Equation 6 is not on the infinity norm but should be on 1 norm.
>
> **A1&2.** For the Rademacher Complexity, we consider a single output. That means that $C$ is a single row matrix for Rademacher Complexity, i.e. a vector which is denoted with lower-case letter in our work. Note that the infinity operator norm of matrix $C$ is equal to the maximum L1 norm of each row of $C$, hence, constraining its L-infinity operator norm corresponds to constraining the Rademacher Complexity of each output.
> _______
>
> > **Q3&4.** The reference order is messy in the paper (e.g. Table 1 appears before Table 2), making it hard to read. Figure 3 and Table 2 are never referred to (pls correct me if I am wrong), neither in the main text nor in the appendix.
>
> **A3&4.** Fixed; thanks for pointing this out.
> _______
>
> > **Q5.** Appendix G, bullet point 5, did not mention CIFAR10.
>
> **A5.** In Appendix G we mention the datasets that did not appear in the main paper. For this reason we do not mention CIFAR-10. We have clarified this in the updated manuscript.
>
> _______
>
> > **Q6.** Appendix G.5, "from all four datasets in Fig. 9"? There are altogether 6 datasets as listed by the author right above that line.
>
> **A6.** Fixed; thanks for pointing this out.
>
> _______
>
> > **Q7.** Appendix G.5, Table 11 is about MNIST and K-MNIST, not MNIST and E-MNIST.
>
> **A7.** Fixed; thanks for pointing this out.
>
> _______
>
> > **Q8.** Although in the summary of main contributions, the author mentioned the sweet-spot for regularization parameter and accuracy robustness trade-off, there are little to no descriptions/explanations/justifications for them in the experiment section. The "Log of Bound" might be the regularization parameter that the authors were referring to, but still, the lack of sufficient descriptions of the data shown makes things hard to comprehend.
>
> **A8.** Even though the original text did mention that the intermediate values provide a trade-off between the highest accuracy and the highest robust accuracy, we have clarified this in the revised manuscript to make it clearer to the reader. Specifically, we have inserted the following sentence: “Notice that in all cases, intermediate values of the bounds ($\lambda$) yield an increased performance in terms of the test-accuracy and the adversarial perturbations.”

---

### Official Review · Reviewer_7Q5M · 2021-11-02

**Correctness:** 4
**Technical Novelty And Significance:** 4
**Empirical Novelty And Significance:** 3
**Recommendation:** 8
**Confidence:** 3

**Main Review:**

## Strengths:
1. The main theoretical results are strong:
   - Novel - I am not aware of any prior works on the Rademacher Complexity and Lipschitz bounds of polynomial networks
   - Sound - I checked the proofs for Theorem 1 and 2 and they all look correct to me. The three other theorems (Theorem 3, 4 and 5) all look reasonable to me, but I admittedly did not check their proofs.
2. The empirical results are comprehensive and convincing. The authors showed that the proposed infinity projection approach is effective at making polynomial networks more robust than baselines on five image recognition datasets and one audio recognition dataset. Although the idea of constraining $L_\infty$ norm of weight matrices to defend against $L_\infty$ perturbation is not new in the adversarial robustness literature for standard feedforward networks, seeing its effectiveness in polynomial networks is still quite interesting.

## Weaknesses:
My biggest concern is the gap between the Rademacher Complexity upper-bound presented (in Theorem 1) and the practical bound that is used for regularizing the polynomial networks in the empirical section. Rather than controlling the norm of the face-splitting product $\\|U_1 \bullet \cdots \bullet U_k \\|_\infty$ suggested by the theory, the proposed technique individually controls the norm $\\|U_i\\|_\infty$. This seems to be quite a significant relaxation and it is unclear how loose the practical bound is. To help get a sense of how much is lost in this relaxation, I wonder if there is a way to numerically approximate the gap between these two bounds (for some fixed polynomial nets).

## Other comments:
- "Cifar10" should be capitalized (e.g., in the Figure 3 caption)
- Appendix C - “This re-parametrization creates equivalent models in terms of models” - It is unclear what the second “models” means here.
- Appendix D -
    - $V = \\{v_1, \cdots v_n\\}$: should it be $d^k$ instead of $n$?
    - Equation 34: should it be $d^k$ on top of max (since $l \in [d^k]$)?

**Summary Of The Paper:**

The paper introduces Rademacher Complexity bounds and Lipschitz constant bounds (both $\ell_2$ and $\ell_\infty$) for two families of polynomial networks. While prior theoretical results on complexity and Lipschitz constant bounds exist for standard neural networks that consist of linear layers and activation functions, these results do not directly apply to Polynomial networks due to their vastly different structures from standard neural networks.

On the practical side, the authors found that imposing $L_\infty$ norm constraint on the weight matrices of the polynomial networks is effective against adversarial perturbation. In particular, the paper focuses on robust image and audio recognition tasks and found out that imposing $L_\infty$ norm constraint via projected gradient descent is more effective than other commonly deployed regularization techniques (in standard networks) such as Jacobian regularization and weight decay.

**Summary Of The Review:**

The theoretical results in this paper are strong, and they serve as a significant first step in understanding the generalization and robustness properties of polynomial networks. The practical regularization technique motivated by the theoretical results is effective in making polynomial networks more robust evaluated by common gradient-based adversarial attacks on image and audio recognition tasks. The only weakness of the paper is that the proposed regularization technique can feel a bit disconnected from the main theoretical results (due to relaxation for tractability consideration), making it a bit less clear whether the practical regularization is actually controlling the Rademacher complexity effectively. Despite this weakness, I believe the paper still deserves acceptance for its great theoretical and practical contributions in polynomial networks.

---

> ### Author Response · Authors · 2021-11-18
> **Response to the Reviewer 7Q5M**
>
> > **Q1.** The gap between the norm of  the face-splitting product $|| U_1 \bullet \cdots \bullet U_k ||_{\infty}$
> >
> > and $\Pi_{i=1}^k || U_i ||_{\infty}$
>
> **A1.** We first remark that the reason that we use this relaxation is that, projection onto the ball $|| U_1 \bullet \cdots \bullet U_k ||_{\infty}$ is a non-convex problem for which there might be no efficient solution, to the best of our knowledge, there is no algorithm available to actually do the projection. This is a promising research direction but not the focus of our paper. Nevertheless, in order to understand how large the gap is, we use three different methods to detect the value of the gap (also included in sec. H.1):
>
> 1. All entries in each $U_i$ are random variables with normal distribution, and we calculate the value of gap. The results are in Figure 5 in the new version of paper.
> 2. We compute the value of the gap for the model trained without any regularization, and we plot how it changes during training. The results are in subfigures (b), (d) of Figure 4 in the new version of paper.
> 3. The ratio between the two quantities, when the model is trained with different regularization parameters. The results are in subfigures (a), (c) of Figure 4 in the new version of paper. The Fig. 4 can also be found here: https://imgur.com/a/XAHjJMb
>
> These experimental results exhibit that our relaxation is not a very loose relaxation.
>
> _______
>
> > **Q2.** "Cifar10" should be capitalized (e.g., in the Figure 3 caption).
>
> **A2.** Fixed; thanks for pointing this out.
>
> _______
>
> > **Q3.** Appendix C - “This re-parametrization creates equivalent models in terms of models” - It is unclear what the second “models” means here.
>
> **A3.** The second model here refers to the original form of CCP and NCP in the original polynomial network paper. In Appendix C, we do a re-parametrization to obtain an equivalent model. However, to make this explicit, we rephrase it to: 'This re-parametrization creates equivalent models, but enables us to [...]'.
>
> _______
>
> > **Q4.** Appendix D. V and Equation 34.
>
> **A4.** Fixed; thanks for pointing this out.

---

### Official Review · Reviewer_SK73 · 2021-11-02

**Correctness:** 4
**Technical Novelty And Significance:** 3
**Empirical Novelty And Significance:** 3
**Recommendation:** 6
**Confidence:** 3

**Main Review:**

Rademacher complexity bounds imply theoretical generalization guarantees. Polynomial networks have been recently proposed and have shown good empirical performance, so analysis of their theoretical properties is relevant. It is not clear to me how these bounds compare to known generalization bounds for regular neural networks.
The bounds are the first generalization bounds for polynomial networks, but do not use algorithmic properties like stability or regularization and are therefore potentially loose. Bounding network weight norms for regularization itself is not novel, but for polynomial networks small empirical improvements are observed in accuracy of using proposed regularization in non-adversarial settings.

L-infinity Lipschitz constant of the networks is related to robustness against L-infinity-bounded adversarial perturbations. Again the algorithmic insights of coupling regularization with adversarial training are not novel, but the idea is verified to be useful also in the context of polynomial networks.

In my opinion the theoretical contributions of the paper are the most interesting, and it would be useful if the authors include brief technical insights or proof sketches for the main theorems.

**Summary Of The Paper:**

The paper provides theoretical guarantees for polynomial neural networks, specifically bounds on Empirical Rademacher Complexity (which implies generalization guarantees) and on the Lipschitz constant of the networks (implies robustness). Polynomial nets have shown state-of-the-art results for several machine learning tasks. Specifically CCP and NCP models of Chrysos et al. (2020) are considered, and the bounds assume bounded l-infinity-norm which is meaningful for image inputs. Further an algorithm is proposed which adds regularization to polynomial networks using the terms in the theoretical upper bounds (corresponding to weight norms) which leads to gain in accuracy, especially in adversarial settings (where the regularization is combined with adversarial training).

**Summary Of The Review:**

The paper provides first generalization bounds for polynomial networks, which have been found to be empirically effective in prior work. Also an upper bound is provided for the Lipschitz constant of polynomial networks. Formal guarantees of low error and high robustness for empirically used techniques are an important contribution of the paper. The algorithmic insights from the bounds can potentially guide regularization and robustness techniques used with these networks. My main complaint is that very little space is allocated to providing proof insights, and practical insights of regularizing network weights and combining with adversarial training have limited novelty.

---

> ### Author Response · Authors · 2021-11-18
> **Response to the Reviewer SK73**
>
> We are thankful to the reviewer for their feedback. We respond below to the remarks:
>
> >**Q1.** The bounds are the first generalization bounds for polynomial networks, but do not use algorithmic properties like stability or regularization and are therefore potentially loose.
>
> **A1.** Obtaining generalization bounds through stability arguments is an interesting research direction. We are currently not aware of any stability bounds for Polynomial Networks, so we do not know if they are tighter. We remark that even though our bounds could be loose, the goal is to provide regularization schemes that improve the performance on downstream classification tasks. This is indeed achieved, as our numerical experiments show. In the revised manuscript, we point towards the stability bounds as a research direction: 'A future direction of research is to obtain generalization bounds for this class of functions using stability notions.'
> _______
>
> >**Q2.** In my opinion the theoretical contributions of the paper are the most interesting, and it would be useful if the authors include brief technical insights or proof sketches for the main theorems.
>
> **A2.** We appreciate the remark from the reviewer. We have *included the proof sketch with the insights to Theorem 1*. The other theorems share similar insights, which due to space limitations are not included in the main paper. If more space is allowed in the camera-ready version, we will include those insights.

---

> > ### Author Response · Authors · 2021-11-30
> > **Any remaining concerns of the reviewer?**
> >
> > Dear Reviewer SK73,
> >
> > given that the discussion window is closing soon, we would like to confirm whether the concern of the reviewer has been addressed.
> >
> > One of the main limitations mentioned was the lack of insights, which have been added. The regularization schemes (i.e., layer-wise bounds have been added during the rebuttal), which are designed based on the insights of the bounds, can be used to improve the critical robustness. Given the theoretical guarantees and the thorough empirical evidence, we believe that our work presents a complete story.
> >
> > If there is any additional comment or question, we would be happy to answer to the reviewer.

---

### Official Review · Reviewer_opuA · 2021-11-04

**Correctness:** 4
**Technical Novelty And Significance:** 3
**Empirical Novelty And Significance:** 2
**Recommendation:** 5
**Confidence:** 3

**Main Review:**

Pros:
- The Rademacher complexity bounds of the CCP and the NCP are derived.
- The Lipschitz constants for the CCP and the NCP are derived.

Cons:

1. Lemma 1 requires projecting $\prod_{i=1}^k || U_i||_{\infty}$, all $U_i$ jointly, in a ball. But in Eq. (9), the authors project each $U_i$ independently to the same ball, which makes me less excited. Do you set the same $\lambda$ for all $U_i$? Different $U_i$ may need different $\lambda$, right?
If we take a squre of the constraints of $U_i$ in Eq. (9) on both sides, i.e. $|| U_i ||^2 \leq \lambda^2$, and add it to the objective using Lagrange multiplier, it is equivalent to the L2 regularization (the weight decay) for each $U_i$.

2. In Fig. 2, decreasing the bounds could hurt the classification performance on clean data. Is there a bound that can keep the performance of clean data the highest and the accuracy for the adversarial data also the highest?

3. Results in Table 1 show that the proposed method on clean data does not have obvious superiority over the Jacobian regularization and the L2 regularization methods. But from the title of this paper, I would expect that the proposed method can also significantly outperform other regularization methods on clean data.

4. Since the main advantage of this method is adversarial defense and attack. Why not compare with some work that is proposed for adversarial defense/attack? For example, “Feature Denoising for Improving Adversarial Robustness, CVPR 2019”.

Minor:

- Should Table 1 appear before Table 2?

- Figure 3 is not referenced.

**Summary Of The Paper:**

This paper analyzed the empirical Rademacher complexity of polynomial nets, the Coupled CP-Decomposition (CCP), and Nested Coupled CP-decomposition(NCP) nets. The corresponding Lipschitz constants for both models are derived. The Projected SGD and the Projected SGD + Adversarial Training algorithms are used to optimize the CCP and the NCP with the Lipschitz constants. Experiments on classification and adversarial attack tasks are performed to evaluate the proposed method. Results show that the proposed projection regularization is better than the Jacobian regularization and the L2 regularization methods in adversarial attack tasks.


**Summary Of The Review:**

- Optimizing the projection of $\prod_{i=1}^k || U_i||_{\infty}$ is not good enough.
- I think the authors should also compare with some works that are for adversarial defense/attack tasks.

---

> ### Author Response · Authors · 2021-11-18
> **Response to the Reviewer opuA (part 2 of 2)**
>
> >**Q3.** If we take a square of the constraints of $U_i$ in Eq. (9) on both sides, i.e. $||U_i|| \leq \lambda$ and add it to the objective using Lagrange multiplier, it is equivalent to the L2 regularization (the weight decay) for each Ui.
>
> **A3.**  In the initial version there was a typo where in eq. (9) we missed the subscript and it should read $||U_i||_\infty$ instead of $||U_i||$. This has been fixed, and now it corresponds to eq. (11). Our proposed choice of norm is the $\ell_\infty$-operator norm, which leads to a different regularization than L2 regularization (weight decay).
> _______
>
> >**Q4.** Is there a bound that can keep the performance of clean data the highest and the accuracy for the adversarial data also the highest?
>
> **A4.** There is a trade-off between the robustness and accuracy of neural networks that has been well documented in prior work (references [B], [C]), and has been derived formally in simple settings like linear regression (reference [A]). Indeed, our experiments do not indicate the existence of such a bound. So, the short answer is that there is a trade-off in the obtained results, which is consistent with trade-offs studied theoretically. However, there are values of the bounds that simultaneously improve both the clean and the robust accuracy.
> _______
>
>
>
> >**Q5.** Results in Table 1 show that the proposed method on clean data does not have obvious superiority over the Jacobian regularization and the L2 regularization methods. But from the title of this paper, I would expect that the proposed method can also significantly outperform other regularization methods on clean data.
>
> **A5.** Results in Table 1 show that our method outperforms L2 and Jacobian Regularization consistently in various datasets. It is true that the percentage is not as large as in adversarial attacks, but the consistency across different datasets is indicative.
>
> _______
>
> >**Q6.** Since the main advantage of this method is adversarial defense and attack. Why not compare with some work that is proposed for adversarial defense/attack? For example, “Feature Denoising for Improving Adversarial Robustness, CVPR 2019”.
>
> **A6.** We appreciate the remark by the reviewer. We have opted in to compare with the following three methods: gaussian denoising[D], median denoising[D] and guided denoising[E]. We use PN-4 on Fashion-MNIST for this experiment. The results in the following table show that our method has the best results compared with these three methods.
>
> |Fashion-MNIST|FGSM0.01|FGSM0.1|
> |----------------|-------------------------------|-----------------------------|
> |No proj|26.49±3.13%|12.92±2.74%|
> |Our method|58.09±1.63%|46.43±0.95%|
> |Layer-wise bound|**63.95±1.26%**|**55.14±3.65%**|
> |Gaussian denoising|18.80±3.08%|14.14±2.77%|
> |Median denoising|19.68±3.20%|14.02±1.95%|
> |Guided denoising|29.69±5.37%|22.94±5.65%|
>
> Moreover, because the suggested methods in [D] and [E] are denoising procedures for the input, they are not a direct replacement to regularization. Hence, in practice they could be combined with our method.
> _______
>
> >**Minor.** Should Table 1 appear before Table 2? Figure 3 is not referenced.
> >
> **A.** Fixed; thanks for pointing this out.
>
>
> ### References
> [A] Precise Tradeoffs in Adversarial Training for Linear Regression. Adel Javanmard, Mahdi Soltanolkotabi, Hamed Hassani. Proceedings of the Thirty Third Conference on Learning Theory (COLT) 2020.
>
> [B] Edgar Dobriban, Hamed Hassani, David Hong, and Alexander Robey. Provable tradeoffs in adversarially robust classification. arXiv preprint arXiv:2006.05161, 2020.
>
> [C] Dimitris Tsipras, Shibani Santurkar, Logan Engstrom, Alexander Turner, and Aleksander Madry. Robustness may be at odds with accuracy. arXiv preprint arXiv:1805.12152, 2018.
>
> [D] F. Liao, M. Liang, Y. Dong, T. Pang, X. Hu, and J. Zhu. Defense against adversarial attacks using high-level representation guided denoiser. InProceedings of the IEEE Conference on ComputerVision and Pattern Recognition, 2018.
>
> [E] C. Xie, Y. Wu, L. v. d. Maaten, A. L. Yuille, and K. He.  Feature denoising for improving adversarial robustness. InProceedings of the IEEE/CVF Conference on Computer Vision and Pattern Recognition, 2019.
>
> [F] Yuichi Yoshida, Takeru Miyato. Spectral Norm Regularization for Improving the Generalizability of Deep Learning. https://arxiv.org/abs/1705.10941

---

> ### Author Response · Authors · 2021-11-18
> **Response to the Reviewer opuA (part 1 of 2)**
>
> > **Q1.** Do you set the same $\lambda$ for all  $U_i$ ? Different $U_i$ may need different $\lambda$ right?
>
> **A1.** We appreciate the suggestion from the reviewer. We have included experiments with different $\lambda$ for each matrix $U_i$ in the revised manuscript. We briefly describe below the obtained results (also included in sec. H.8 in the paper).
>
> We conduct an experiment on PN-4 where we set a different projection bound for each matrix $U_i$. Specifically, we use five different candidate values for each $\lambda_i$ (i.e., bound that corresponds to $U_i$), and then perform the grid search on the  Fashion-MNIST FGSM-0.01 attack. The results in the table below indicate that the improvement over the previous uniform bounds is statistically significant, confirming the intuition that different $U_i$ might need different $\lambda_i$.
>
> |PN-4|No proj|Jacobian|L-2|Single bound|Layer-wise bound|
> |----------------|-------------------------------|-----------------------------|---------------|-------------------------------|-----------------------------|
> |Fashion-MNIST FGSM0.01|26.49±3.13%|39.88±4.59%|24.36±1.95%|58.09±1.63%|**63.95±1.26%**|
>
> One reasonable question is whether the obtained $\lambda_i$ can be transferred to other datasets as well. This would reduce the amount of computation needed for the hyper-parameter search. That is, we use the same layer-wise bounds on K-MNIST and MNIST and present the results below:
>
> |PN-4|No proj|Jacobian|L-2|Single bound|Layer-wise bound|
> |----------------|-------------------------------|-----------------------------|---------------|-------------------------------|-----------------------------|
> |K-MNIST FGSM0.01|23.31±5.34%|25.46±3.51%|27.85±7.62%|43.74±5.97%|**49.61±1.44%**|
> |MNIST FGSM0.01|34.14±7.63%|32.78±6.94%|29.31±3.95%|73.95±5.18%|**79.23±3.65%**|
>
> One further extension is to assess whether the same layer-wise bounds can be transferred to the stronger FGSM-0.1 attack, and we confirm this in the table below:
>
> |PN-4|No proj|Jacobian|L-2|Single bound|Layer-wise bound|
> |----------------|-------------------------------|-----------------------------|---------------|-------------------------------|-----------------------------|
> |Fashion-MNIST FGSM0.1 |12.92±2.74%|17.90±6.51%|13.80±3.65%|46.43±0.95%|**55.14±3.65%**|
> |K-MNIST FGSM0.1|18.86±2.61%|22.61±1.30%|22.05±2.76%|35.84±1.67%|**47.54±3.74%**|
> |MNIST FGSM0.1|20.96±5.16%|33.59±8.46%|26.07±5.64%|64.09±2.41%|**74.97±5.60%**|
>
> We should note though, that choosing a different $\lambda_i$ becomes impractical as the degree of the polynomial grows since there are more $U_i$’s and the hyper-parameter grid grows exponentially. This is also an issue in regularization schemes such as weight decay and spectral normalization when one wants to use different values per layer. The interested practitioner can choose whether to use a single $\lambda$ or to further perform search for the layer-wise bounds, according to their computational budget.
>
> To sum up, our updated experiments exhibit how the proposed method can improve even further the robust accuracy by selecting a layer-wise bound. We are thankful to the reviewer for the remark.
>
> >**Q2.** Lemma 1 requires projecting $\Pi ||U_i||_\infty$ for all $U_i$ jointly, in a ball. But in Eq. (9), the authors project each $U_i$ independently to the same ball, which makes me less excited.
>
> **A2.** You are right that Lemma 1 requires a projection over the ball $\Pi ||U_i||_\infty \leq \lambda $,
>
> or as Theorem 1 suggests, projection over the ball $||\bullet_{i=1}^k U_i|| \leq \lambda$. However, the corresponding optimization problem is a non-convex non-smooth problem for which an efficient algorithm to solve it does not exist, after conducting an extensive search in the literature.
> In general, such type of problems are NP-hard so it might be hard to find an efficient algorithm. Constraining each matrix separately, as we do, is a way to control the product in an efficient way, because the projection for a single matrix has efficient projection algorithms. This also issue also appears in related regularization methods like Spectral Normalization for Neural Networks[F] and they use the same solution as we do i.e., project each layer independently.
>
> Because of the hardness of the problem, we believe that it is a promising avenue of research. We will remark this in the revised version. In our opinion, this is not crucial to demonstrate the merits of this work, as we already show empirical improvements even without access to such projection oracle.

---

> > ### Comment · Reviewer_opuA · 2021-11-28
> > **Response**
> >
> > I would thank the authors for their response. I'm glad to see that the layer-wise bound significantly and consistently improves the results achieved by the single bound. However, grid search is used to find the best $\lambda_i$ for each layer and the results are only achieved on small datasets MNIST, K-MNIST, FASHION-MNIST. I would consider the grid-search is not scalable for large-scale datasets and deeper networks. The main reason is that the important theoretical problem, the projection of $\prod_{i=1}^k || U_i||_{\infty}$, is not properly solved in this paper. Also, as I mentioned in my initial review. the results of the proposed regularization have no obvious advantage on clean data. Therefore, I would like to keep my score.

---

> > > ### Author Response · Authors · 2021-11-28
> > > **Second response to reviewer opuA (2/2)**
> > >
> > > ____
> > >
> > >
> > > >The results of the proposed regularization have no obvious advantage on clean data
> > >
> > > **A.** Improving accuracy on clean data is a difficult task using regularization alone, however our experiments show that on clean data we either slightly outperform or are at the same level as other regularization schemes, while performing much better on adversarial data of different perturbation sizes. This shows that our method is indeed closer to the Pareto-Optimal frontier, i.e. it has a better accuracy-robustness trade-off than the other methods. In this sense, our method is better. The vast literature on the robustness issues in Deep Learning (that is also published in ICLR) has demonstrated that evaluating accuracy alone is not enough for a new regularization method.
> > >
> > > Our experiments show that our method is strong compared to baselines, we would like to remind the reviewer that in the ICLR guidelines, it is stated that empirical performance alone is not a reason for rejection. See bottom Q&A section in https://iclr.cc/Conferences/2021/ReviewerGuide  (copied below for convenience)
> > >
> > > >Q: If a submission does not achieve state-of-the-art results, is that grounds for rejection?
> > > A: No, a lack of state-of-the-art results does not by itself constitute grounds for rejection. Submissions bring value to the ICLR community when they convincingly demonstrate new, relevant, impactful knowledge. Submissions can achieve this without achieving state-of-the-art results.
> > >
> > > ___
> > >
> > > Now onto the final issue raised by the reviewer
> > >
> > > >The main reason is that the important theoretical problem, the projection of $\Pi_{i=1}^k \|U_i\|_\infty$, is not properly solved in this paper
> > >
> > > We try to convince the reviewer again, that this is a hard problem which might not have an efficient solution, i.e. NP-hard, and also that it is not uncommon to relax the problem to a simpler one that can be solved efficiently, as we do. We remark that there are two different problems:
> > > 1. Projection onto $\Pi_{i=1}^k ||U_i|||_\infty \leq \lambda$ : Note that this is the same bound that appears for example in the Spectral Normalization [MKKY18] and spectral regularization frameworks [YM17], which are motivated by the spectral bounds of Bartlett [BFT17]. None of such methods or any follow-up work that we know of, tackles the problem of the projection onto such a set. This gives evidence that it is a hard problem that might remain without answer, as these works have been known for many years. The spectral normalization and spectral regularization papers **workaround this issue in the same way as we do**, by relaxing the problem and doing projection/regularization for each layer independently. However, if the reviewer has any suggestion on the topic, we would be happy to study it.
> > > 2. Projection onto $|| \bullet_{k=1}^k U_i ||_\infty \leq \lambda$. The term in this projection has multiplications between variables of different layers. The only regularizer that we know of similar structure is the path-norm of neural networks [NTS15]. The 1-path-norm would be a regularizer that is also non-smooth and non-convex as ours. However, to the best of our knowledge, only the proximal mapping (a generalization of projection) is known for the 1-path-norm in the case of shallow neural networks [LRNC20]. The derivation of such mapping is already highly non-trivial and the authors of [LRNC20] mention how *their arguments do not extend to more layers*, a problem that appears out of reach. For this reason we do not believe withholding our results, which show that the relaxed problem already shows improvement, makes sense.
> > >
> > >
> > > **References**
> > >
> > > [LRNC20] Efficient Proximal Mapping of the 1-path-norm of Shallow Networks
> > > Fabian Latorre, Paul Rolland, Nadav Hallak, Volkan Cevher, ICML 2020.
> > > http://proceedings.mlr.press/v119/latorre20a.html
> > >
> > > [NTS15] Norm-Based Capacity Control in Neural Networks
> > > Behnam Neyshabur, Ryota Tomioka, Nathan Srebro. COLT 2015.
> > > http://proceedings.mlr.press/v40/Neyshabur15
> > >
> > > [MKKY18] Spectral Normalization for Generative Adversarial Networks,
> > > Takeru Miyato and Toshiki Kataoka and Masanori Koyama and Yuichi Yoshida
> > > https://openreview.net/forum?id=B1QRgziT-
> > >
> > > [YM17] Spectral Norm Regularization for Improving the Generalizability of Deep Learning
> > > Yuichi Yoshida, Takeru Miyato https://arxiv.org/abs/1705.10941
> > >
> > > [BFT17] Spectrally-normalized margin bounds for neural networks
> > > Peter L. Bartlett. Dylan J. Foster. Matus Telgarsky. NeurIPS 2017
> > > https://proceedings.neurips.cc/paper/2017/file/b22b257ad0519d4500539da3c8bcf4dd-Paper.pdf

---

> > > > ### Author Response · Authors · 2021-11-30
> > > > **Have the concerns of the reviewer been addressed?**
> > > >
> > > > Dear reviewer opuA,
> > > >
> > > > since the discussion window is closing soon, we would like to check whether there are any remaining concerns or whether our responses have addressed them.
> > > >
> > > > As a kind reminder to the reviewer, the [reviewer guidelines for ICLR](https://iclr.cc/Conferences/2022/ReviewerGuide) mention the following two proposals:
> > > > *  Submissions bring value to the ICLR community when they convincingly demonstrate new, relevant, impactful, or insightful knowledge.
> > > > * Don’t reject a paper just because you don’t find it “interesting”. This should not be a criterion at all for accepting/rejecting a paper. The research community is so big that somebody will find some value in the paper (maybe even a few years down the road), even if you don’t see it right now.
> > > >
> > > > In our rebuttal we have addressed the first and last point of the original review (layer-wise bound, and comparison with adversarial defense methods), while we have answered in the remaining two points raised.
> > > >
> > > > Lastly, as the other reviewers highlight, we provide sound and **useful** theoretical results (first known generalization error bounds and Lipschitz constant bounds) for a **new** class of functions that boost the performance of PNs.
> > > >
> > > > We truly believe the suggestions of the reviewer have improved the paper, and we hope our responses have addressed the concerns of the reviewer.

---

> > > ### Author Response · Authors · 2021-11-28
> > > **Second response to reviewer opuA (1/2)**
> > >
> > > We are thankful to the reviewer for responding. The reviewer raises few points that do not necessarily agree with theoretical/empirical/practical evidence and we address them below.
> > >
> > > **tl;dr:** a) the layer-wise bound results generalize beyond image recognition in audio recognition (which is also higher-dimensional than MNIST experiments) using the **same layer-wise bounds** optimized for  Fashion-MNIST FGSM0.01, b) the original submission had already a **single bound that works** well without the need for grid-search on $\lambda_i$, c) many influential papers in the literature relax the non-convex projection in the same way as we do.
> > >
> > >
> > >
> > > Detailed responses:
> > > ====
> > >
> > >
> > > >*results are only achieved on small MNIST-like datasets*:
> > >
> > > **A.** We have extended the results to **NSYNTH** (in an audio recognition experiment; NSYNTH has $4\times$ the dimensionality of the MNIST datasets), and using the **same** $\lambda_i$ values as Fashion-MNIST FGSM-0.01 we obtain the following results in FGSM-0.1 attack: 39.97% the layer-wise bound, versus 22.25% the single-bound (0.91% without projection).
> > > To avoid any confusion of the reader, there are experiments on CIFAR10 (with the single bound) on the paper that demonstrate a clear improvement over the baselines.
> > >
> > > ___
> > >
> > >
> > > >grid-search is not scalable
> > >
> > > **A.** This is a well known issue with grid-search, however we would like to argue why we believe this is not a reason for rejection:
> > >
> > > 1. With the same argument, there exists no layer-wise regularization method that is scalable (because one would have to do a grid-search). Nevertheless, there have been a myriad of layer-wise methods in the context of neural networks that have proven to be useful for performance, as well as impactful as measured by number of citations. For example the well known *drop-out* method as well as *weight decay, $\ell_1$-regularization, max-norm regularization, spectral regularization*, among many others, can be defined using a different parameter or constraint for each layer. This has not prevented these methods to be embraced by researchers and practitioners alike.
> > > 2. Simply put, if one has additional computational power grid-search is an **optional, but not mandatory procedure** that can improve the performance even further. Moreover, grid-search is **trivially parallelizable**. We show that the simplest and most efficient way of regularizing, which is using a single parameter for all layers, already improves the performance compared to the baselines in our experiments.
> > > 3. Nowadays, there exist many algorithms and efficiently implemented software tools for hyperparameter optimization that, to some degree, can help to alleviate or solve such a problem. To name a few tools: **HYPEROPT** (https://github.com/hyperopt/hyperopt) and **OPTUNA** (https://optuna.org). Thus we strongly believe this does not pose any barrier for practitioners.
> > > 4. There are many heuristics that also help alleviate the problem, and we already show one such method. In our experiments we find optimal hyperparameters using a **single dataset and adversarial attack** (Fashion-MNIST FGSM0.01), and we reuse the same $\lambda_i$ choices, at no extra computational cost, for other datasets. As our experiments show, such choice improves over using a single $\lambda$ so this simple heuristic is good. This can only mean that state-of-the-art methods for hyperparameter tuning could be even more efficient and perform **even better**.
> > > 5. Finally, we remark that the layer-wise bound was a proposal of the reviewer, which we implemented, but it is not our main claim. Actually, in the paper it first appears in the last section of the supplementary (sec. H.8).

---

### Author Response · Authors · 2021-11-18
**Revisions to the paper**

We appreciate the effort of the reviewers and their feedback. A revised version of our work has been uploaded. We have extended a number of sections:

1. We have conducted experiments with variable layer-wise constraints (suggested by reviewer opuA). The results with layer-wise constraints improve even further the robustness, outperforming by a large margin the baselines (sec. H.8).
2. We have conducted additional experiments with the requested adversarial defense methods (suggested by reviewer opuA) (sec. H.9).
3. We have included the baseline results on CIFAR-10 as requested (suggested by reviewer GLBZ)(sec. 4.2).
4. We have added the sketch proof for the interested reader in the main paper (suggested by reviewer SK73) (sec. 2.1).
5. We have numerically approximated the ratio between the theoretical bound and the algorithmic bound (as requested by reviewer 7Q5M) (sec. H.1).



These changes have improved the experimental evaluation and presentation of our paper, and the layer-wise constraints proposed by reviewer opuA show that even larger improvements can be obtained, increasing the value of our proposed method. Again, we thank all the reviewers for helping us improve our work. Other particular questions and issues raised by reviewers are answered in their respective threads. The changes in the updated manuscript are denoted in *red color* for visibility.

We kindly ask all reviewers to consider raising their scores, if their opinion of our paper has improved after the revision.

---

### Decision · Program_Chairs · 2022-01-20

**Decision:**

Accept (Poster)

**Comment:**

This paper considers generalization of polynomial networks. It gives a characterization of the Rademacher complexity as well as Lipschitz constants for polynomial nets. Inspired by the theoretical results, the paper also proposed regularization schemes that empirically improves accuracy and robustness. Most reviewers found the theoretical results to be interesting (but there are some concerns about the mismatch between the upperbound in theory and used in practice, which was partially addressed in the response). There are some more concerns about the experiments but many of them are addressed in the new version. Overall although polynomial networks are not popular in practice, this paper provides some interesting theoretical results.